# MiR-205-driven downregulation of cholesterol biosynthesis through SQLE-inhibition identifies therapeutic vulnerability in aggressive prostate cancer

C. Kalogirou[1,2], J. Linxweiler[3], P. Schmucker [2], M. T. Snaebjornsson[4], W. Schmitz [1], S. Wach[5], M. Krebs[2], E. Hartmann[6], M. Puhr[7], A. Müller [8], M. Spahn[9], A. K. Seitz[2], T. Frank[2], H. Marouf[1], G. Büchel [1,10], M. Eckstein [11], H. Kübler[2], M. Eilers [1], M. Saar[3], K. Junker[3], F. Röhrig[1], B. Kneitz[2], M. T. Rosenfeldt [6] & A. Schulze [1,4✉]

Prostate cancer (PCa) shows strong dependence on the androgen receptor (AR) pathway. Here, we show that squalene epoxidase (SQLE), an enzyme of the cholesterol biosynthesis pathway, is overexpressed in advanced PCa and its expression correlates with poor survival. SQLE expression is controlled by micro-RNA 205 (miR-205), which is significantly down-regulated in advanced PCa. Restoration of miR-205 expression or competitive inhibition of SQLE led to inhibition of de novo cholesterol biosynthesis. Furthermore, SQLE was essential for proliferation of AR-positive PCa cell lines, including abiraterone or enzalutamide resistant derivatives, and blocked transactivation of the AR pathway. Inhibition of SQLE with the FDA approved antifungal drug terbinafine also efficiently blocked orthotopic tumour growth in mice. Finally, terbinafine reduced levels of prostate specific antigen (PSA) in three out of four late-stage PCa patients. These results highlight SQLE as a therapeutic target for the treatment of advanced PCa.

[1] Department of Biochemistry and Molecular Biology, Theodor-Boveri-Institute, Biocenter, Würzburg, Germany. [2] Department of Urology and Paediatric Urology, University Hospital Würzburg, Würzburg, Germany. [3] Department of Urology, Saarland University, Homburg/Saar, Germany. [4] German Cancer Research Center, Division of Tumor Metabolism and Microenvironment, Heidelberg, Germany. [5] Department of Urology, University Hospital Erlangen, Friedrich-Alexander University of Erlangen-Nuremberg, Erlangen, Germany. [6] Institute of Pathology, Julius Maximilians University and Comprehensive Cancer Center (CCC) Mainfranken, Würzburg, Germany. [7] Department of Urology, Medical University of Innsbruck, Innsbruck, Austria. [8] Clinic for Diagnostic and Interventional Radiology, Saarland University, Homburg/Saar, Germany. [9] Center for Urology, Hirslanden Private Hospital Group, Zurich, Switzerland. [10] Mildred Scheel Early Career Center, University Hospital Würzburg, Würzburg, Germany. [11] Institute of Pathology, University Hospital Erlangen, Friedrich-Alexander University of Erlangen-Nuremberg, Erlangen, Germany. ✉email: almut.schulze@dkfz-heidelberg.de

Prostate cancer (PCa) is a very heterogeneous disease regarding its clinical behaviour and likelihood of progression, and therefore poses a global therapeutic challenge for clinicians[1]. After initial response to androgen-deprivation therapy (ADT), advanced PCa can sustain growth even after suppression of exogenous testosterone supply through biochemical castration (castration-resistant prostate cancer, CRPC)[2]. One mechanism of ADT resistance involves the intratumoral synthesis of steroid hormones, either from adrenal precursors or from cholesterol produced by PCa cells[3–5]. Multiple studies have shown that CRPC still relies on the provision of steroid hormones and that intratumoral steroidogenesis can contribute to disease progression[3, 6, 7]. In addition, AR mutations observed in CRPC can result in relaxed ligand specificity, allowing other steroids, including glucocorticoids, to activate AR signalling and promote tumour growth[8, 9]. Importantly, steroid-hormone synthesis requires cholesterol as a substrate and PCa is among the solid cancer entities with the highest amount of cholesterol uptake, storage, and usage[10].

Intratumoral steroidogenesis and ligand promiscuity provided the rationale for the development of therapeutic strategies that interfere with the androgen-receptor pathway in PCa. Inhibition of cytochrome P450 family 17 subfamily A member 1 (CYP17A1), an enzyme in the testosterone-synthesis pathway, by abiraterone acetate or the use of the nonsteroidal AR inhibitor enzalutamide have been evaluated for their efficacy in CRPC and are in clinical use[11, 12]. Although both agents have significantly improved prognosis for CRPC patients, the 5-year overall survival (OS) in CRPC is still below 20–30%. Moreover, the emergence of additional resistance mechanisms contributes to clinical limitations[13]. Therefore, developing additional treatment strategies for CRPC remains a high clinical need.

In this study, we have investigated the role of squalene epoxidase (SQLE, also known as squalene monooxygenase)[14], a rate-limiting enzyme of cholesterol synthesis and known cancer-related gene in PCa, whose expression is correlated with poor prognosis and survival in this entity[15, 16]. We confirm that high SQLE protein expression is correlated with poor outcome in PCa. We also show that SQLE is a target of micro-RNA-205 (miR-205), a micro-RNA that is downregulated in PCa[17–22] via epigenetic silencing[23] or via reduced expression of its upstream regulator p63[24]. Moreover, we show that blocking SQLE through miR-205 overexpression, genetic deletion, or competitive pharmacological inhibition significantly impaired growth in a panel of PCa cell lines, including abiraterone- and enzalutamide-resistant derivatives. We prove that inhibition of cell proliferation upon SQLE blockade is mediated by reduced cholesterol synthesis, leading to diminished activity of the androgen receptor. Moreover, SQLE inhibition efficiently reduced orthotopic tumour growth without affecting systemic cholesterol metabolism in mice. Moreover, in a small case series of late-stage, heavily pre-treated PCa patients, we found that orally administered terbinafine led to substantial reduction in PSA levels in three out of four patients. In summary, our study reveals that elevated cholesterol biosynthesis through enhanced SQLE expression contributes to PCa tumour growth and suggests that pharmacological blockade of cholesterol biosynthesis via SQLE inhibition could be a therapeutic strategy for the treatment of advanced PCa.

## Results

**Elevated SQLE expression is correlated with aggressive PCa.** To address the role of SQLE in PCa, we first confirmed that high SQLE expression is significantly associated with shorter biochemical relapse (BCR)-free survival in a TCGA dataset for primary PCa (PRAD primary; $n = 430$), as well as with higher preoperative PSA levels, overall Gleason scores, advanced tumour stages, and nodal-positive disease (Fig. 1a). SQLE gene amplification was more frequent in metastatic lesions (24%) compared with primary tumours (7%) (Fig. S1a) and gene amplification was associated with enhanced expression in both primary and metastatic PCa (Fig. S1b), indicating that elevated SQLE expression might drive disease progression. In addition, 3-hydroxy-3-methylglutaryl-CoA synthase 1 (HMGCS1) and Farnesyl-Diphosphate Farnesyltransferase 1 (FDFT1), two other enzymes of the cholesterol biosynthesis pathway, showed significant coexpression with SQLE in metastatic PCa (Fig. S1c), suggesting a general upregulation of cholesterol biosynthesis in metastatic PCa. We also analysed a tissue microarray (TMA) containing tumour samples from 453 PCa patients (TMA1, Table 1) to correlate SQLE protein expression with established clinical end-points. Representative tissue cores with high and low Gleason scores are shown in Fig. 1b. Analysis of all cores using an SQLE expression score revealed that SQLE protein levels were increased in tumours with high Gleason score (≤7a, ISUP2), advanced tumour stage and positive nodal status (Fig. 1c). Moreover, SQLE levels showed positive correlation with preoperative serum PSA (Fig. 1d). High SQLE protein levels also predicted significantly worse outcome for relapse-free survival (RFS) and OS in this cohort (Fig. 1e, f).

In addition, we analysed SQLE protein levels in a TMA from a second PCa cohort comprising 179 patients (TMA2, Table 1)[25]. To account for intratumoral heterogeneity, we compared tumour areas with high Gleason score (Gleason ≥ 7b, ISUP ≥ 3) to those with low Gleason score (Gleason 6 and 7a, ISUP1/2) of the same tumour across the entire cohort. Representative tissue cores are shown in Fig. S1d. Grouped analysis showed significantly higher SQLE expression in regions with high Gleason score (Fig. S1e), while no significant association with Gleason score, tumour stage, nodal status and resection status, or correlation with preoperative serum PSA was found in this patient cohort (Fig. S1f, g). Nevertheless, SQLE expression score determined in the high Gleason core from each patient reliably predicted worse outcome for RFS but not OS (Fig. S1h, i). Cox regression and multivariate analysis also indicated that SQLE either as a continuous or categorical variable is independently associated with RFS (Supplementary Table 1).

While residual confounding factors may still limit the value of SQLE as a prognostic tumour marker, our findings indicate that high expression of SQLE mRNA and protein is associated with advanced PCa.

**Micro-RNA 205 negatively regulates SQLE in PCa.** PCa progression is associated with alterations in micro-RNA expression[26]. We therefore used in silico alignment tools to identify potential micro-RNAs that could regulate the expression of SQLE. Interestingly, several of the most highly ranked micro-RNAs also showed reduced expression in PCa in publicly available datasets (Gene Expression Omnibus), with the top-scoring miR-205 showing the strongest regulation (Fig. 2a). Based on this information, we identified a conserved sequence in the 3′-UTR of the SQLE mRNA that matches the seed region for miR-205 (Fig. 2b). Applying a dual-luciferase assay and transient transfection of a miR-205 expression vector in LNCaP cells, we confirmed that miR-205 indeed regulates SQLE expression and that deletion of the micro-RNA-binding site within the 3′-UTR of the SQLE mRNA abolishes this regulation (Fig. 2c). We next generated a doxycycline-inducible overexpression model for miR-205 in a panel of PCa cell lines (Fig. 2d). Induction of miR-205 levels significantly diminished SQLE expression in all cell lines, including the hormone-ablation resistant (LNCaP-Abl), as well as

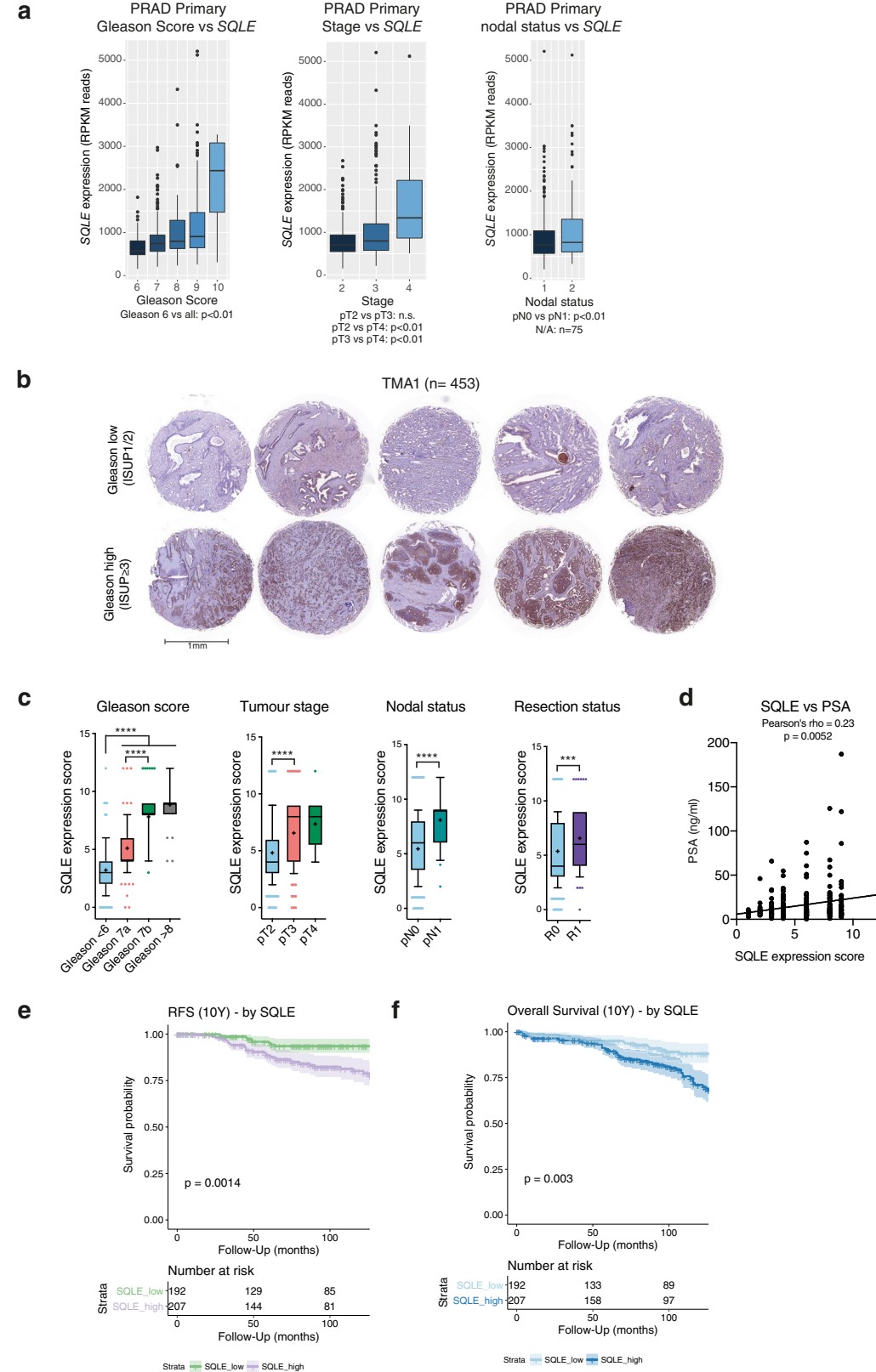

abiraterone (LNCaP-Abl-Abi) and enzalutamide-resistant (LNCaP-Abl-Enza) derivatives of the LNCaP cell line (Fig. 2e).

To assess the correlation between miR-205 and *SQLE* expression, we analysed an in-house collection of formalin-fixed, paraffin-embedded tissue samples consisting of primary PCa ($N = 59$), lymph-node metastasis ($N = 22$), and benign prostate tissues ($N = 10$). This revealed that *SQLE* mRNA levels were inversely correlated with miR-205 expression across the entire cohort and along the line of progression from benign to primary tumour and to lymph-node metastasis (Fig. 2f). Moreover, elevated *SQLE* mRNA expression was significantly elevated in tumours with advanced stage, Gleason score and resection status

**Fig. 1 SQLE expression is correlated with metastasis formation and lethal disease in prostate cancer. a** SQLE expression in relation to Gleason score (left), pathological tumour stage (pT, middle), and nodal status (pN, right) in the TCGA prostate adenocarcinoma (PRAD) dataset (PanCancer Atlas, $N = 494$) as assessed by RNASeq2 on reads per kilo base per million mapped reads (RPKM). Data are presented as box and whiskers plots (box 25–75th percentile, whiskers 1.5 × interquartile range, line at median, outliers plotted individually). Differences between Gleason scores and pathological tumour stages were calculated with standard forward ANOVA. Differences between nodal statuses were calculated with unpaired two-tailed Student's t-test. **b** SQLE expression in representative tissue cores from the PCa cohort of tumour microarray 1 (TMA1) ($N = 453$). Tumour tissue samples displaying low Gleason score (ISUP1/2, top) were compared to samples displaying high Gleason score (ISUP ≥ 3, bottom). **c** Box-and-whiskers plots (box 25–75th percentile, whiskers 10–90th percentile, line at median, + at mean) of SQLE histoscores (SQLE expression score) in tumour tissue samples compared by Gleason score, tumour stage, nodal status and resection status from the patient cohort represented in TMA1 ($N = 453$). (***$p \leq 0.0001$, ****$p \leq 0.0001$, two-sided Mann–Whitney test). **d** Pearson correlation (one-sided) of SQLE expression score with preoperative prostate specific antigen (PSA) levels in patients from the same cohort as in (**c**). **e**, **f** Ten-year Kaplan–Meier plot of recurrence-free (RFS) (**e**) and overall survival (OS) (**f**) in the PCa cohort of TMA1 ($N = 453$) dichotomised into high ("SQLE_high") and low ("SQLE_low") based on median SQLE expression score. Survival differences were calculated with the log-rank test.

**Table 1 Clinical characteristics of PCa patient cohorts used in the study.**

|  | TMA1 | TMA2 | FFPE collection PCa | FFPE collection BPH |
|---|---|---|---|---|
| Origin | a) | b) | c) | c) |
| Number of patients | 453 | 179 | 59 | 10 |
| Time of diagnosis | 1999–2010 | 1987–2005 | 2009–2012 | 2009–2012 |
| Age at surgery | 65 (45–83) | 66 (43–81) | 66,5 (52–81) | 71,2 (67–81) |
| Preoperative PSA | 14 (0,4–187) | 47,6 (20–160) | 13,5 (1,9–122,1) | 3,6 (2,1–8,0) |
| Follow-Up (months) | 98 (3–246) | 78,3 (6–154) | 21,2 (3–79) |  |
| *Overall Gleason-Score (specimen)* |  |  |  |  |
| Gleason 6 | 139 (30,7%) | 82 (45,8%) | 15 (25,4%) |  |
| Gleason 7 | 239 (52,8%) | 41 (22,9%) | 37 (62,7%) |  |
| Gleason 8 | 42 (9,3%) | 23 (12,8%) | 2 (3,4%) |  |
| Gleason 9 | 32 (7,2%) | 14 (7,8%) | 5 (8,5%) |  |
| Gleason 10 | 0 (0%) | 1 (0,6%) | 0 (0%) |  |
| N/A | 1 (0,02%) | 18 (10,1%) | 0 (0%) |  |
| *Tumour Stage* |  |  |  |  |
| pT2 | 266 (58,7%) | 30 (16,8%) | 37 (62,7%) |  |
| pT3 | 156 (34,4%) | 116 (64,8%) | 21 (35,6%) |  |
| pT4 | 31 (6,9%) | 33 (18,4%) | 1 (1,7%) |  |
| N/A | 0 (0%) | 0 (0%) | 0 (0%) |  |
| *Nodal stage* |  |  |  |  |
| pN0 | 417 (92,1%) | 104 (58,1%) | 50 (84,7%) |  |
| pN+ | 26 (5,7%) | 75 (41,9%) | 9 (15,3%) |  |
| pNX | 1 (0,02%) | 0 (0%) | 0 (0%) |  |
| N/A | 9 (2%) |  |  |  |
| *Resection status* |  |  |  |  |
| R0 | 353 (77,9%) | 61 (34,1%) | 41 (70%) |  |
| R+ | 87 (19,2%) | 86 (48%) | 18 (30%) |  |
| RX | 3 (0,7%) | 32 (17,9%) | 0 (0%) |  |
| N/A | 10 (2,2%) | 0 (0%) | 0 (0%) |  |

a) Dept. of Pathology, University Hospital Erlangen, Germany. b) Community Hospital of Karlsruhe, Germany and University Hospital Leuven, Belgium. c) Dept. of Urology and Paediatric Urology, UKW Würzburg, Germany.

in this patient cohort (Supplementary Fig. 2a). Additionally, by comparing paired primary tumour and lymph-node metastasis samples of nine patients, we demonstrated that SQLE expression was significantly elevated in metastases compared with primary tumours, similar to the results obtained for kallikrein-related peptidase 3 (*KLK3*), the gene coding for PSA (Fig. 2g). These findings substantiate the conclusion that SQLE mRNA and protein expression is correlated with poor outcome and metastatic disease in PCa.

To confirm the regulation of SQLE by miR-205, we performed a knockout of miR-205 using a CRISPR/Cas9 strategy (Fig. 2h). This efficiently ablated expression of miR-205 (Fig. 2i) without affecting the expression of its host gene (*MIR205HG*), a long noncoding RNA with functions during development[27] (Fig. S2b). Expression of SQLE mRNA and protein was strongly increased following miR-205 knock-out in PC-3 (Fig. 2i, j) and LNCaP cells (Supplementary Fig. 2c). As control, we also confirmed that miR-

205 knockout increased expression of ZEB1, a previously confirmed miR-205 target[28], and promoted epithelial-to-mesenchymal transition (EMT), indicated by increased expression of vimentin (*VIM*) and decreased expression of E-cadherin (*ECAD*) mRNA (Supplementary Fig. 2d, e), as well as enhanced cell migration (Supplementary Fig. 2f).

Together, these results confirm that miR-205 controls SQLE and suggest that downregulation of miR-205 could be responsible for the enhanced expression of SQLE found in advanced PCa.

**MiR-205 and SQLE regulate cholesterol synthesis in PCa cells.** SQLE is one of the rate-limiting enzymes of the cholesterol biosynthesis pathway. To evaluate the impact of SQLE and miR-205 on cholesterol biosynthesis in PCa cells, we performed tracing experiments using stable isotope-labelled subtrates, [U$^{13}$C]-glucose, [U$^{13}$C]-glutamine, and [U$^{13}$C]-acetate, that feed into

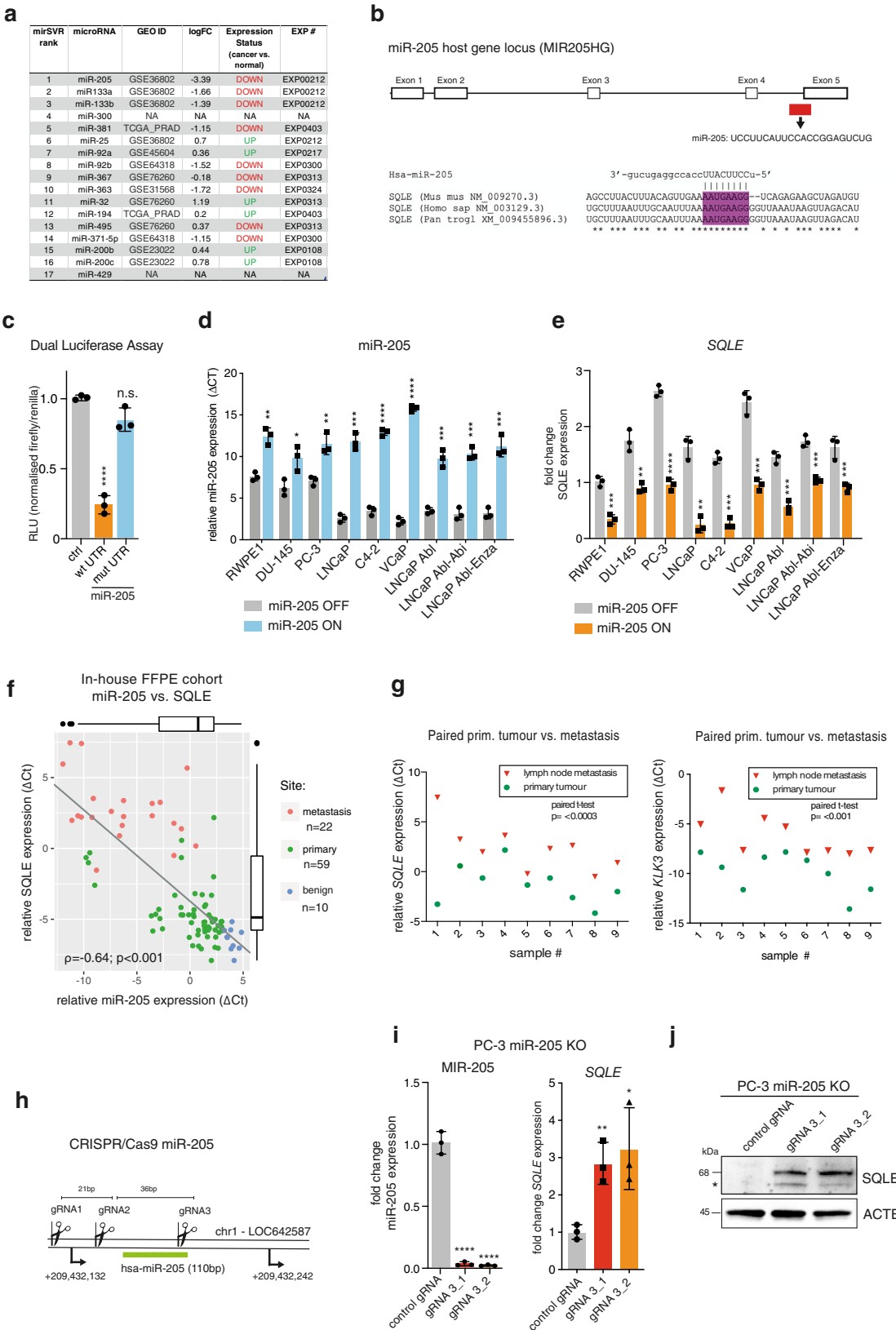

cholesterol synthesis via acetyl-CoA. We cultured C4-2 and LNCaP cells in the presence of these compounds and determined the incorporation of $^{13}$C atoms into free cholesterol and cholesteryl esters using LC–MS. Activation of miR-205 expression efficiently blocked de novo cholesterol synthesis, indicated by the reduction in labelled isotopologues and significantly reduced total

labelling of free cholesterol (Fig. 3a). A similar inhibition of cholesterol synthesis was also observed when cells were treated with the antimicrobial agent terbinafine, a competitive inhibitor of the SQLE enzyme[29]. These results were also confirmed using LNCaP cells labelled with [U$^{13}$C]-glucose (Supplementary Fig. 3a). In addition, overexpression of SQLE increased the

**Fig. 2 miR-205 negatively regulates *SQLE* in PCa. a** In silico assessment of microRNAs predicted to regulate *SQLE* (http://www.microrna.org). MicroRNAs are ranked by mirSVR score and their expression in PCa is shown as log2FC (dbDEMC 2.0). **b** Diagram showing the location of miR-205 within its host gene (*MIR205HG*) and the alignment to the 3′UTR of the *SQLE* gene from human, mouse, and chimpanzee. **c** A pMIR-REPORT plasmid containing a wild-type or mutated 3′-UTR of the *SQLE* mRNA was cotransfected with precursor miR-205 in LNCaP cells. Firefly luciferase activity, in relative light units (RLU), was analysed 48 h after transfection and normalised to renilla luciferase (pRL-TK). Reporter activities of cells cotransfected with miR-precursor negative control were used as control. Data are presented as mean +/− SD of three independent experiments (*$p < 0.05$, unpaired two-tailed Student's *t*-test). **d, e** A panel of PCa cell lines was transduced with a doxycycline-inducible vector expressing miR-205 and treated with 1 µg/mL of doxycycline (miR-205 ON) or vehicle control (miR-205 OFF) for 48 h. Induction of miR-205 (**d**) and *SQLE* expression (**e**) were analysed by qRT-PCR. Data are presented as mean +/− SD of six independent biological replicates in three independent experiments (*$p < 0.05$, **$p < 0.01$, ***$p < 0.001$, ****$p < 0.0001$, unpaired two-tailed Student's *t*-test). **f** Spearman rank correlation of relative miR-205 and *SQLE* expression determined by qRT-PCR in benign prostate tissues ($N = 10$, blue), primary prostate cancer tissues ($N = 59$, green), and lymph-node metastases ($N = 22$, red). RNA was extracted from formalin-fixed paraffin-embedded (FFPE) tissues from an in-house collection. **g** Comparison of *SQLE* (left) and *KLK3* (right) expression in nine paired samples of primary tumour tissue and lymph-node metastases from the same cohort as in (**f**). Statistical differences were calculated with standard forward ANOVA. **h** CRISPR/Cas9 knockout strategy for miR-205. The location of the three guide RNAs is displayed on the miR-205 locus on chromosome 1. The mature miR-205 sequence is marked in green. **i** Expression of miR-205 and *SQLE* in pools of PC-3 cells after deletion of miR-205 using two combinations of gRNAs. Cells expressing Cas9 and a nontargeting guide RNA were used as controls. Data are presented as mean +/− SD of three independent experiments (*$p < 0.05$, unpaired two-tailed Student's *t*-test). **j** WB showing expression of SQLE protein in parallel samples to (*i*). Actin B (ACTB) is shown as loading control. A representative image of two independent experiments is shown.

labelling of free cholesterol in C4-2 cells (Supplementary Fig. 3b), indicating that SQLE expression is rate-limiting for cholesterol synthesis in PCa cells.

Inhibition of cholesterol synthesis could result in a compensatory increase in cholesterol uptake, for example, via the scavenger receptor B1 (SR-B1)[30]. We therefore determined the effect of miR-205 expression or SQLE inhibition on cholesterol uptake by culturing the cells in the presence of cholesterol-2,3,4-$^{13}C_3$ and detecting the fraction of cholesterol-$^{13}C_3$ within the total cholesterol pool. This analysis showed that miR-205 induction or inhibition of SQLE not only reduced the fraction of unlabelled cholesterol (resulting from de novo synthesis) but also the fraction of cholesterol-$^{13}C_3$ (resulting from uptake) in C4-2 and LNCaP cells (Fig. 3b and Supplementary Fig. 3c), suggesting that cholesterol uptake cannot compensate for the loss of SQLE activity. Interestingly, we also observed that de novo synthesis of esterified cholesterol was strongly affected by miR-205 induction or terbinafine treatment (Fig. 3c). Synthesis of cholesteryl esters requires the activity of sterol O-acyltransferase 1 (SOAT1)[31], which attaches an acyl chain onto cholesterol via an ester bond. Interestingly, terbinafine treatment resulted in a strong reduction in SOAT1 mRNA expression (Fig. 3d), indicating a potential feedback regulation between cholesterol synthesis and the esterification pathway. We next compared the effect of terbinafine on cholesterol synthesis to that of a second SQLE inhibitor, NB-598, which was recently identified as a specific inhibitor of this enzyme in neuroendocrine cancers[32]. In line with the higher binding affinity of this compound[33], we observed that NB-598 more efficiently reduced cholesterol synthesis compared with terbinafine, with 1 µM NB-598 already achieving almost complete inhibition (Fig. 3e–g).

These results show that blocking SQLE activity, either by induction of miR-205 or by chemical inhibition, efficiently ablates the synthesis of cholesterol and cholesteryl esters in PCa cells without compensatory upregulation of cholesterol uptake.

**SQLE inhibition blocks cell proliferation and overcomes resistance to second-generation AR inhibitors.** To determine the functional role of SQLE in vital cellular processes in PCa, we first established doxycycline-inducible expression systems for two independent shRNA sequences targeting *SQLE* in PC-3, C4-2, and LNCaP cells. Treatment with doxycycline efficiently reduced SQLE mRNA and protein expression in all three cell lines (Supplementary Fig. 4a, b). Importantly, silencing of SQLE also reduced proliferation, which was restored by the addition of cell-

permeable cholesterol (Fig. 4a–c). Conversely, overexpression of SQLE increased proliferation and migration in C4-2 cells (Fig. 4d–f). Using a doxycycline-inducible system, we observed that induction of miR-205 reduced SQLE protein expression in both PC-3 and LNCaP cells (Supplementary Fig. 4c) and lowered cell proliferation, which was also restored by cell-permeable cholesterol (Fig. 4g). Interestingly, addition of the synthetic androgen R1881 fully relieved proliferation inhibition caused by miR-205 overexpression in the AR-positive LNCaP cells (Fig. 4g). In contrast, inhibition of proliferation by miR-205 in AR-negative PC-3 cells was not rescued by R1881 (Fig. 4g). This suggests that the miR-205/*SQLE* axis supports PCa proliferation through at least two mechanisms: it provides cholesterol, most likely for membrane synthesis, but also mediates AR activation in hormone-dependent PCa cells.

We next applied chemical inhibition of SQLE in a panel of PCa cells using terbinafine and NB-598. While terbinafine reduced proliferation to some extent in all cell lines, the AR-positive cell lines (C4-2, VCaP and LNCaP) were significantly more sensitive to the compound compared with the AR-negative cell lines (PC-3 and DU145) (Fig. 5a). Importantly, C4-2 cells, an androgen-independent derivative of the LNCaP cell line[34], as well as the CRPC-derived VCaP cell line[35], both showed reduced proliferation after terbinafine treatment (Fig. 5a), indicating that these cells still depend on SQLE activity, despite being independent of exogenous androgens. Terbinafine treatment also resulted in the induction of apoptosis only in AR-positive PCa cell lines, as demonstrated by Annexin V positivity and PARP cleavage (Supplementary Fig. 5a, b). Similar results were also obtained with NB-598, as C4-2, VCaP, and LNCaP cells were more sensitive toward treatment with this compound (Fig. 5b). Surprisingly, although NB-598 was very effective in blocking cholesterol synthesis (Fig. 3e), we noticed that it was less efficient in reducing cell viability compared with terbinafine, with 50–60% of cells remaining viable after 96 h of treatment with 25 µM of NB-598 (Fig. 5b). It has been suggested that terbinafine, in addition to its function as SQLE inhibitor, also induces autophagy in liver cancer cells through an unknown mechanism[36]. To explore this possibility, we investigated whether terbinafine or NB-598 alter the autophagy machinery in PCa cells. Interestingly, terbinafine, but not NB-598, increased the expression of LC3 (MAP1LC3A) in LNCaP cells (Fig. 5c), the cell line that showed the highest sensitivity toward this compound (Fig. 5a). This suggests that induction of autophagy could contribute to the toxicity of this drug at least in some PCa cell lines.

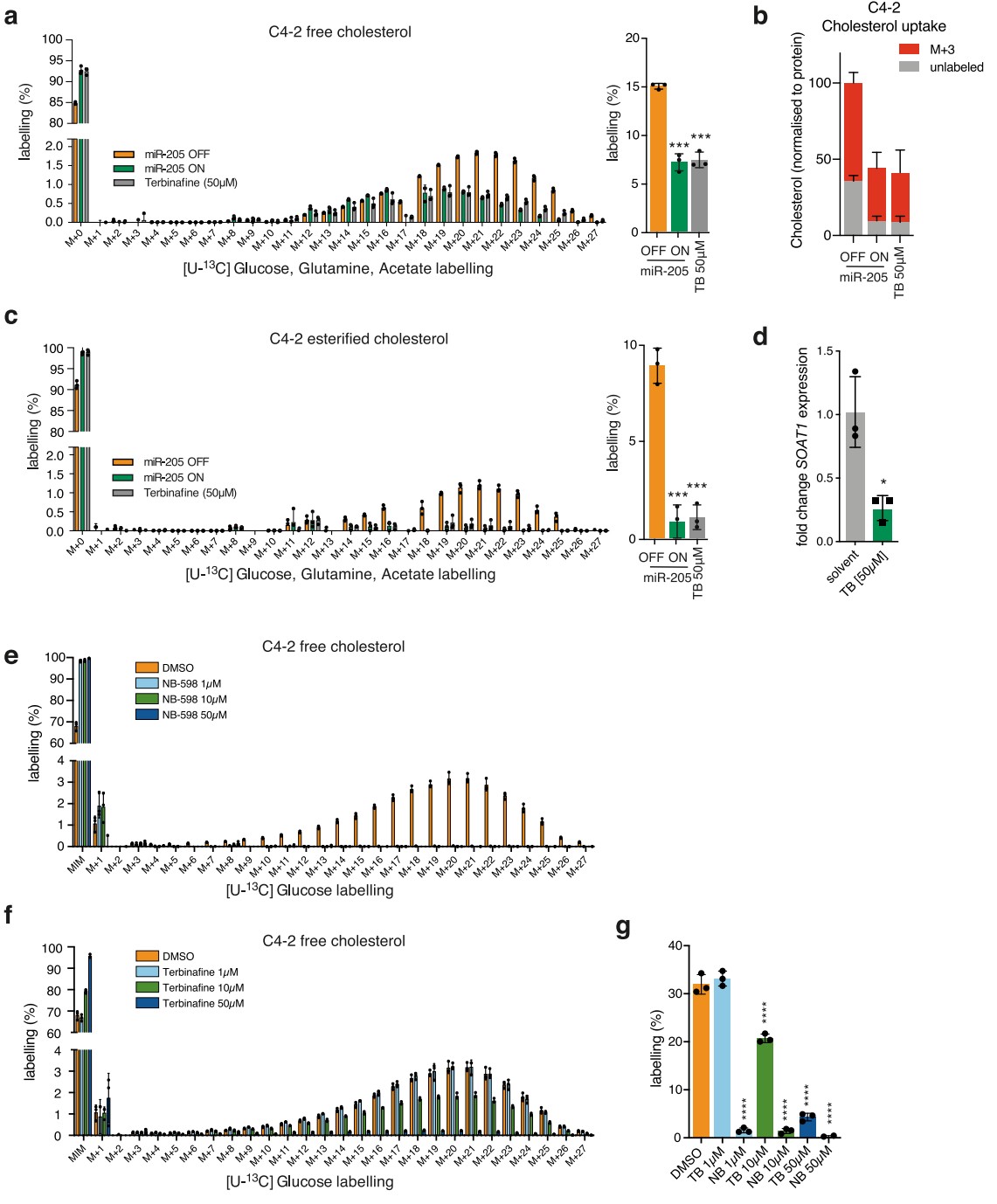

**Fig. 3 *SQLE* regulation by miR-205 controls cholesterol biosynthesis in PCa cells. a** C4-2 cells expressing doxycycline-inducible miR-205 were treated with 1 μg/mL of doxycycline for 72 h (miR-205 ON) or solvent (miR-205 OFF), or with 50 μM of terbinafine (TB) for 48 h, and labelled with [U13C]-glucose, [U13C]-glutamine and [U13C]-acetate. Cells were extracted and analysed by LC-MS. Graphs show relative peak intensities of individual isotopologues (left graph) and overall labelled fractions (right graph) for free cholesterol. Data are presented as mean +/− SD of three independent biological replicates (*$p < 0.05$, unpaired two-tailed Student's *t*-test). **b** C4-2 cells were treated as in (**a**) and labelled with cholesterol-2,3,4-13C₃ for 48 h. Cells were extracted and analysed by LC-MS. Graphs show the relative peak area normalised to protein for free cholesterol. The M + 3 fraction represents cholesterol-2,3,4-13C₃ taken up from the medium. Data are presented as mean +/− SD of three independent biological replicates. **c** Relative peak intensities of individual isotopologues (left graph) and overall labelled fractions (right graph) for esterified cholesterol in the same experimental conditions as in (a). **d** Expression of *SOAT1* mRNA in C4-2 cells treated with 50 μM of terbinafine for 48 h. Data show mean +/− SD of three independent biological replicates (**$p \leq 0.01$, unpaired two-tailed Student's t-test). **e–g** C4-2 cells were treated with increasing amounts of NB-598 (**e**) or terbinafine (**f**) for 72 h and cultured in [U13C]-glucose containing media for the last 48 h. Metabolites were extracted and analysed by LC-MS. Graphs show relative peak intensities of individual isotopologues (**e** and **f**) and overall labelled fractions (**g**) for free cholesterol. Data are presented as mean +/− SD of three independent biological replicates (****$p < 0.0001$, unpaired two-tailed Student's *t*-test).

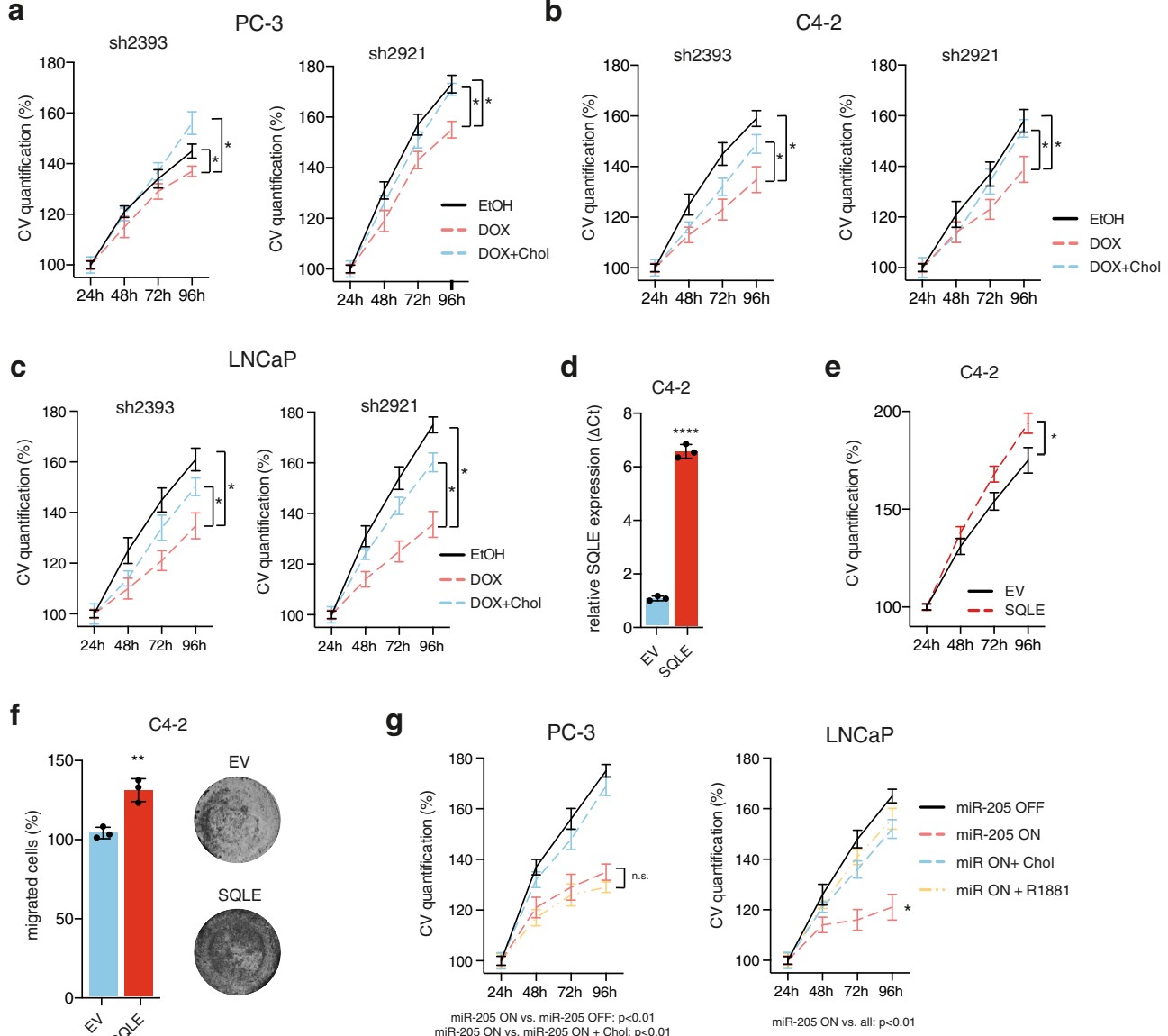

**Fig. 4 SQLE supports proliferation in PCa cells. a–c** Cell viability assays measured by crystal violet (CV) staining of PC-3 (a), C4-2 (b), and LNCaP (c) cells expressing two independent shRNAs targeting *SQLE* (sh2393 or sh2921) and treated with 1 µg/mL doxycycline (DOX) or solvent (EtOH). Cell counts were normalised to cell number at 24 h. Rescue experiments were performed by addition of 0.125 µg/mL cell-permeable cholesterol (DOX + Chol). Data are presented as mean +/− SD of three independent samples (*$p < 0.05$, unpaired two-tailed Student's *t*-test on the final time points). **d** Constitutive overexpression of SQLE in C4-2 cells using a retroviral vector (pBabe) as determined by qRT-PCR. Empty vector cells (EV) were used as control. Data are presented as mean +/− SD of three independent samples (****$p < 0.0001$, unpaired two-tailed Student's *t*-test). **e** Cell viability assays measured by crystal violet staining of C4-2 cells overexpressing SQLE (SQLE) or empty vector (EV) at the indicated time points. Cell count was normalised to initial cell number at 24 h. Data are presented as mean +/− SD of three independent samples (*$p < 0.05$, unpaired two-tailed Student's *t*-test on the final time points). **f** Transwell migration assay of SQLE overexpressing (SQLE) and empty vector (EV) C4-2 cells. Data are presented as mean +/− SD of three independent experiments (**$p < 0.01$, unpaired two-tailed Student's *t*-test). **g** Cell viability assays measured by crystal violet staining of PC-3 and LNCaP cells expressing doxycycline-inducible miR-205 treated with 1 µg/mL doxycycline (miR-205 ON) or ethanol (MIR-205 OFF) for the indicated times. Rescue experiments were performed by addition of 0.125 µg/mL cell-permeable cholesterol (miR ON + Chol) or 1 nM R1881 (miR ON + R1881). Cell counts were normalised to cell number at 24 h. Data are presented as mean +/− SD of three independent samples (*$p < 0.05$, unpaired two-tailed Student's *t*-test on the final time point; n.s. = not significant).

It has been reported that SQLE is downregulated in cholesterol auxotrophic lymphoma and that the resulting accumulation of squalene protects cancer cells from oxidative stress[37]. In contrast, SQLE has been identified as a selective vulnerability in neuroendocrine tumours, where its inhibition leads to toxic squalene accumulation[32]. To determine the mechanism underlying the essential role of SQLE in PCa, we compared the effect of different inhibitors and metabolites of the cholesterol- and

testosterone- biosynthesis pathway (Fig. 5d) on cell viability in two AR-positive PCa cell lines (C4-2 and LNCaP). This analysis revealed that inhibition of proliferation by the FDFT1 inhibitor YM53601, similar to the effect of terbinafine, was also restored by the addition of either cholesterol or the synthetic androgen R1881 (Fig. 5e). In contrast, only R1881 was able to rescue viability after CYP17A1 inhibition with abiraterone (Fig. 5e). These results strongly suggest that cholesterol biosynthesis is rate-limiting for

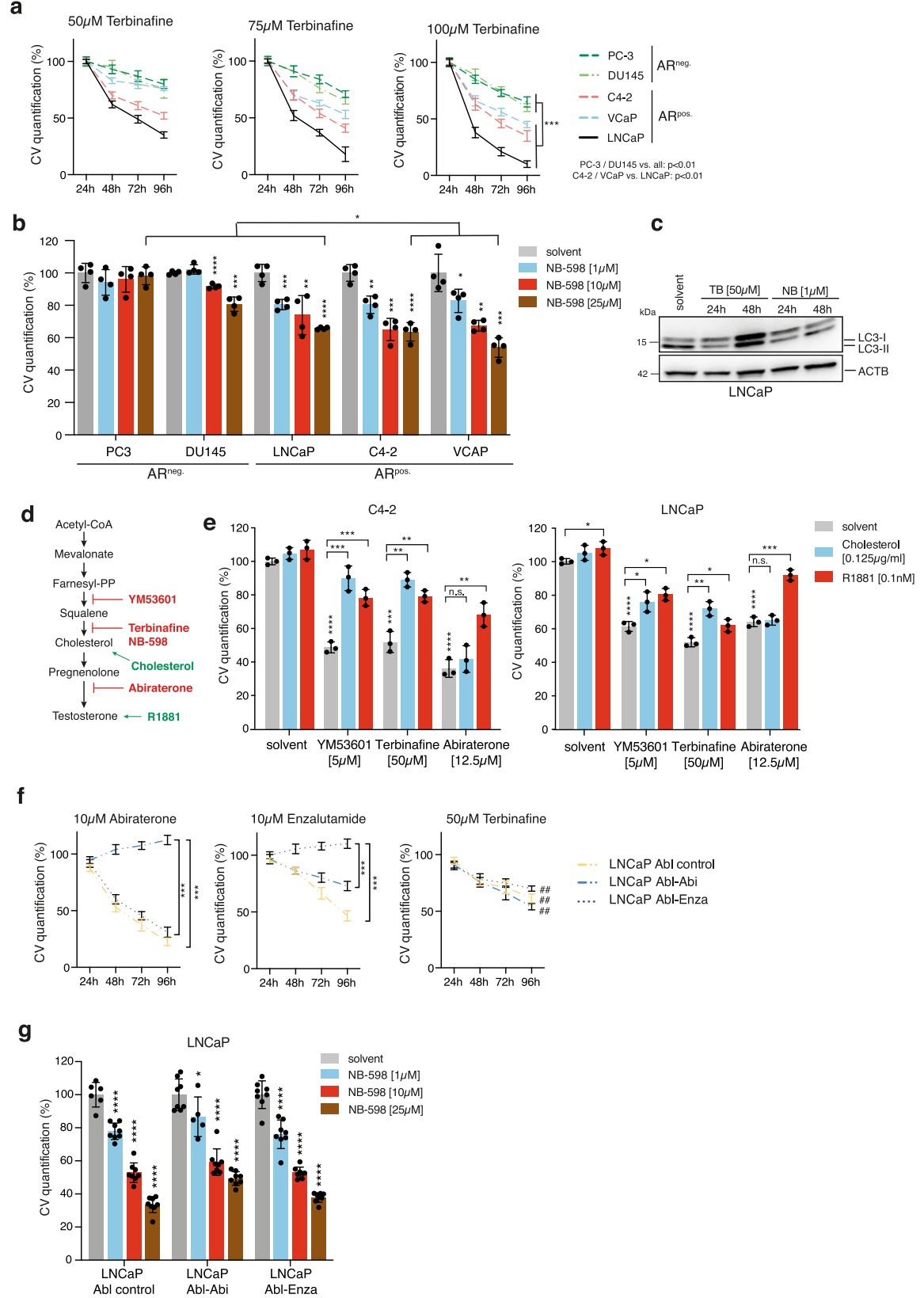

viability in AR-positive PCa cells and that inhibition of SQLE limits viability by blocking cholesterol synthesis and potentially androgen production.

To investigate whether inhibition of SQLE also targets treatment-resistant disease states in PCa, we used a series of derivatives of the LNCaP cell line that were established after long-term culture in androgen-depleted medium[8]. Compared with the parental LNCaP-Abl cell line, LNCaP-Abl-Abi or LNCaP-Abl-Enza cells were fully resistant to treatment with abiraterone or enzalutamide, respectively (Fig. 5f, left and middle graph). However, all cell lines still responded to terbinafine treatment with a 30–50% reduction in cell number (Fig. 5f). Moreover, an

**Fig. 5 Inhibition of SQLE targets AR-positive PCa cells and overcomes resistance to second-generation AR inhibitors. a** Cell viability assays measured by crystal violet (CV) staining of a panel of androgen-receptor (AR)-positive (AR^pos.) and AR-negative (AR^neg.) PCa cell lines after treatment with increasing doses of terbinafine for the indicated times. Data are presented as mean $+/-$ SD of three independent samples. Cell count was normalised to initial cell number at 24 h. Differences between the curves were calculated using unpaired two-tailed Student's t-test for the indicated time points. **b** Cell viability assays measured by crystal violet staining of the same cell line panel as in (a) after treatment with increasing doses of NB-598 for 96 h. Data are presented as mean $+/-$ SD of three independent samples (*$p < 0.05$, **$p < 0.01$, ***$p < 0.001$, ****$p < 0.0001$, unpaired two-tailed Student's t-test). **c** LNCaP cells were treated with 50 μM terbinafine (TB), 1 μM NB-598 (NB), or solvent (DMSO) for 24 or 48 h. LC3 was detected by WB. Actin B is shown as loading control. A representative image of two independent experiments is shown. **d** Diagram of the cholesterol pathway showing inhibitors and rescue compounds used in (e). **e** Cell viability assays measured by crystal violet staining in C4-2 and LNCaP cells after 72 h of treatment with the indicated drugs (5 μM YM53601, 50 μM terbinafine or 12.5 μM abiraterone acetate). Rescue experiments were carried out by addition of 0.125 μg/mL cell-permeable cholesterol or 0.1 nM R1881. Data are presented as mean $+/-$ SD of three independent experiments (*$p < 0.05$, unpaired two-tailed Student's t-test). **f** Cell viability assays measured by crystal violet staining in a panel of LNCaP derived cell lines resistant toward abiraterone (LNCaP Abl-Abi), enzalutamide (LNCaP Abl-Enza), and the parental line (LNCaP Abl control) after treatment with 10 μM abiraterone (left graph), 10 μM enzalutamide (middle graph), or 50 μM terbinafine (right graph) for the indicated times. Cell count was normalised to initial cell number at 24 h. Data are presented as mean $+/-$ SD of three independent replicates (***$p < 0.001$, unpaired two-tailed Student's t-test between Abl-Abi or Abl-Enza and control and on the final time point, ##$p < 0.01$, unpaired two-tailed Student's t-test of cell number at 96 h compared with 24 h). **g** Cell viability assays measured by crystal violet staining in the same cell lines used in (f) after treatment with increasing doses of NB-598 for 96 h. Data are presented as mean $+/-$ SD of three independent replicates (*$p < 0.05$, **$p < 0.01$, ***$p < 0.001$, ****$p < 0.0001$, unpaired two-tailed Student's t-test).

even stronger reduction in cell number was observed after treatment with increasing doses of NB-598 in all cell lines (Fig. 5g). This indicates that SQLE inhibition is also effective in models of advanced PCa, including CRPC as well as enzalutamide or abiraterone resistance, which could have clinical implications for treatment-resistant PCa.

**SQLE regulation by miR-205 controls AR transactivation.** AR belongs to the family of nuclear receptors and is activated by the binding of androgenic hormones, including testosterone and dihydrotestosterone (DHT), leading to its dimerisation, phosphorylation, translocation and DNA binding (Fig. 6a)[2]. While androgens are usually generated by the testes or the endocrine cells of the adrenal gland, progression to CRPC can also involve intratumoral steroidogenesis. This is supported by the observation that enhanced levels of enzymes and metabolic intermediates of testosterone synthesis can be detected in CRPC cells and tumour tissues[3–5]. And that conversion of $^3$H-labelled cholesterol to testosterone has been demonstrated in androgen-independent C81 cells[7]. We therefore hypothesised that regulation of SQLE-dependent cholesterol synthesis by miR-205 could modulate androgen production and AR transactivation in PCa.

To address this question, we used an AR-responsive reporter construct in which a sequence of six copies of the androgen-response element (ARE) had been cloned in front of the luciferase gene[38]. We first confirmed that treatment with dihydrotestosterone (DHT) resulted in a robust activation of this reporter construct in LNCaP and C4-2 cells (Supplementary Fig. 6a). Using this system, we could show that overexpression of mir-205 or inhibition of SQLE by terbinafine resulted in a significant reduction in AR reporter activity in C4-2 cells (Fig. 6b). As the AR gene has been previously identified as a target of miR-205 in some PCa cell lines[21], we also investigated expression of AR mRNA upon miR-205 induction in C4-2 and LNCaP cells. In contrast to the results previously reported, we did not find any evidence for substantial downregulation of AR expression by miR-205 in the cell lines used here (Supplementary Fig. 6b). We next determined the effect of SQLE silencing on AR transactivation and observed that silencing of SQLE using two independent shRNAs efficiently reduced the activity of the AR reporter (Fig. 6c). Conversely, overexpression of SQLE in C4-2 cells enhanced AR reporter activity (Fig. 6d), while AR-driven expression of KLK3 and secretion of PSA into the medium was significantly decreased by SQLE silencing and induced upon SQLE overexpression (Fig. 6e, f). Similar results were also

obtained for transmembrane serine protease 2 (TMPRSS2), another AR-dependent gene (Fig. 6f). Moreover, inhibition of SQLE using terbinafine or NB-598 efficiently reduced KLK3 mRNA levels in C4-2 and LNCaP cells (Fig. 6g and Supplementary Fig. 6c). Importantly, expression of KLK3 and TMPRSS2 mRNA, as well as PSA secretion after terbinafine treatment or miR-205 expression, could be partially rescued by the addition of cell-permeable cholesterol (Fig. 6h and Supplementary Fig. 6d), confirming that cholesterol synthesis downstream of SQLE is rate-limiting for AR transactivation. These findings were confirmed in VCaP cells (Supplementary Fig. 6e).

Taken together, these results demonstrate that modulation of SQLE, through expression of miR-205, genetic targeting, or inhibition of its enzymatic activity, controls AR transactivation in PCa cells.

**SQLE inhibition suppresses tumour growth in an orthotopic PCa mouse model.** We next evaluated the effect of SQLE inhibition in vivo in an established murine orthotopic PCa model. In this system, VCaP cells are orthotopically implanted into the prostate glands of male CB17-SCID mice. After confirmation of orthotopic tumours by 3D ultrasound imaging, mice were randomly allocated to control and treatment cohorts and treated with either carrier or 50 mg/kg of terbinafine every 48 h with intraperitoneal injections. Tumour growth was monitored by consecutive serum PSA levels and by 3D ultrasound and CT/MRI measurements (Fig. 7a). We observed that terbinafine significantly inhibited the progression of serum PSA, with levels reaching an average of 541 ng/mL (±50 ng/mL) in vehicle controls and 230.4 ng/mL (±46 ng/mL) in the terbinafine-treated cohort after six weeks of exposure ($p < 0.01$) (Fig. 7b). In addition, 3D ultrasound imaging revealed reduced tumour growth, with significant differences observed after six weeks of treatment (average 1304 mm$^3$ ± 142 mm$^3$ in vehicle controls vs. 626 mm$^3$ ± 85 mm$^3$ in terbinafine-treated cohort, $p < 0.01$) (Fig. 7c and d). Interestingly, histological analysis of tumour tissue at the endpoint revealed that proliferation associated Ki67 labelling was not significantly reduced in terbinafine-treated tumours compared with DMSO controls (Supplementary Fig. 7a). However, we observed enhanced levels of cleaved caspase 3 and increased TUNEL positivity (Fig. 7e, f), indicating that the tumour reduction following terbinafine treatment was mainly caused by induction of apoptosis rather than inhibition of proliferation. We also observed enhanced formation of LC3 puncta in terbinafine-treated tumours (Fig. 7g and Supplementary Fig. 7b), suggesting

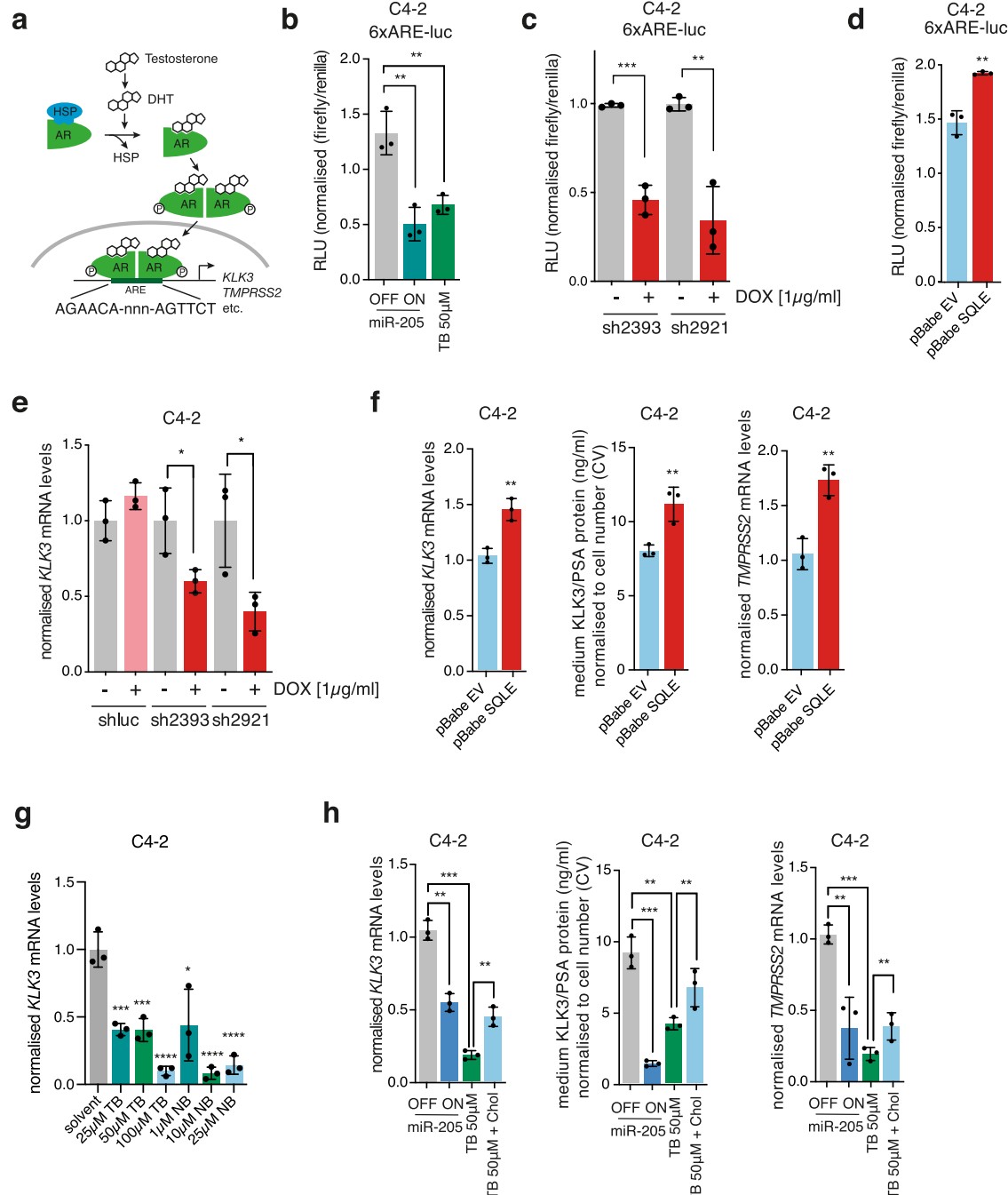

that autophagy could also contribute to the reduction in tumour growth. Furthermore, by subjecting mice to MRI imaging, we were able to demonstrate that tumours of terbinafine-treated mice showed lacunae and other signs of diffusion disturbances in diffusion-weighted ADC mapping (Supplementary Fig. 7c), both indicators of tumour necrosis. This was confirmed by the significant reduction in vital tumour area in terbinafine-treated animals, determined by quantifying gross sections of H&E-stained tumour tissue (Fig. 7h and Supplementary Fig. 7d).

To evaluate whether reduced tumour growth was linked to inhibition of AR transactivation, we also determined the levels of nuclear AR in tumour sections from treated and untreated mice. This analysis clearly showed that terbinafine treatment reduced the abundance of nuclear AR (Fig. 7i), suggesting that terbinafine interferes with AR activation also in vivo. We also investigated whether terbinafine caused systemic effects on circulating

cholesterol levels. However, despite the efficient inhibition of tumour growth, we did not find any significant difference in free cholesterol levels in the serum of treated and control animals (Fig. 7j). This suggests that the anti-tumour effect of terbinafine was not caused by a systemic inhibition of cholesterol biosynthesis that would reduce the availability of circulating cholesterol for steroid hormone synthesis. Instead, our data are consistent with a specific effect of terbinafine on intratumoral cholesterol synthesis that would reduce the availability of substrates for androgen synthesis. Importantly, terbinafine treatment had no overt adverse effects, as serum aspartate-aminotransferase (AST) was unaltered and no morphological alterations in liver or kidney tissue were observed (Supplementary Fig. 7e–g).

Collectively, these results demonstrate that the SQLE inhibitor terbinafine efficiently reduces the growth of PCa orthografts without major systemic toxicity and that inhibition of tumour

**Fig. 6 miR205 and SQLE control AR transactivation in PCa cells. a** Diagram outlining the mechanism of AR transactivation. In the presence of dihydrotestosterone (DHT), AR dimerises and translocates to the nucleus, where it binds to the androgen-response elements (ARE) in its target genes. **b** Activity of the 6xARE-luc reporter in C4-2 cells expressing doxycycline-inducible miR-205 and treated with 1 μg/mL doxycycline (miR-205 ON) or ethanol (miR-205 OFF), or with 50 μM terbinafine for 48 h. Relative light units (RLU) are presented as mean +/− SD of three independent biological replicates (**$p < 0.01$, ***$p < 0.001$, unpaired two-tailed Student's $t$-test). **c** Activity of the 6xARE-luc reporter in C4-2 cells expressing doxycycline-inducible shRNA targeting SQLE (sh2392 or sh2921) and treated with 1 μg/mL doxycycline (DOX) or ethanol for 48 h. Data are presented as mean +/− SD of three independent biological replicates (**$p < 0.01$, ***$p < 0.001$, unpaired two-tailed Student's $t$-test). **d** Activity of the 6xARE-luc reporter in SQLE-overexpressing (SQLE) and empty vector (EV) C4-2 cells. Data are presented as mean +/− SD of three independent biological replicates (**$p < 0.01$, unpaired two-tailed Student's $t$-test). **e** RNA from C4-2 cells treated as in (**c**) was analysed for expression of *KLK3* by qRT-PCR. Data are presented as mean +/− SD of three independent biological replicates (*$p < 0.05$, unpaired two-tailed Student's $t$-test). **f** Expression of KLK3 (left graph), levels of PSA in medium supernatants (middle graph), and expression of *TMPRSS2* (right graph) in C4-2 cells expressing SQLE or empty vector (EV). Data are presented as mean +/− SD of three independent biological replicates (**$p < 0.01$, unpaired two-tailed Student's $t$-test). **g** C4-2 cells were treated with the indicated doses of terbinafine (TB) or NB-598 (NB) for 48 h. Expression of KLK3 mRNA was quantified by qRT-PCR. Data are presented as mean +/− SD of three independent biological replicates (*$p < 0.05$, **$p < 0.01$, ***$p < 0.001$, ****$p < 0.0001$, unpaired two-tailed Student's $t$-test). **h** C4-2 cells expressing doxycycline-inducible miR-205 were treated with 1 μg/mL doxycycline (miR-205 ON) or solvent (miR-205 OFF), or with 50 μM terbinafine (TB 50 μM) for 48 h. Rescue experiments were carried out using 0.125 μg/mL of cell-permeable cholesterol (TB 50 μM + Chol). Expression of *KLK3* mRNA (left), levels of PSA in medium supernatants (middle), and expression of *TMPRSS2* mRNA were determined. Data are presented as mean +/− SD of three independent biological replicates (*$p < 0.05$, **$p < 0.01$, ***$p < 0.001$, unpaired two-tailed Student's $t$-test).

growth by terbinafine is associated with the induction of apoptosis and reduced AR activity.

**Terbinafine treatment reduces PSA levels in patients with advanced CRPC.** In order to investigate the effect of SQLE inhibition in human PCa, we analysed a case series of four late-stage, heavily pretreated PCa patients receiving orally administered terbinafine (Lamisil) as off-label, individual clinical interventions. Clinical characteristics of patients are provided in Supplementary Table 2 and diagnostic imaging for patients #1 and #2 is displayed in Supplementary Fig. 8a–d. All patients had a long medical history regarding their PCa, having received an average of four preceding therapies (excluding ADT and radiation) before being offered this individual clinical intervention and presented with a rapidly progressive disease, with an average PSA velocity of over 115 ng/mL/month and an average PSA-doubling time of 2.1 months (Supplementary Table 2).

Three of the four patients received the standard dose of 500 mg of terbinafine daily, while one patient received a reduced dose of 250 mg a day due to liver impairment. This application of terbinafine represents the maximum dose and duration for the treatment of fungal infections[39]. During the short course of treatment, we did not observe any adverse effects ≥ grade III according to the common terminology criteria for adverse events (CTCAE). After two weeks of treatment, we were able to document a PSA decline in three of the four patients (Fig. 8a). One patient showed an initial moderate increase (12.2% over the baseline) in PSA levels after two weeks of terbinafine treatment. However, after sustained treatment, no further PSA incline was noted in this patient. A detailed account of treatment history and PSA course for two patients is provided in Fig. 8b and Supplementary Fig. 8e. Ultimately, due to the advanced disease stage, three of four patients died of their disease at an average of 2.7 months after the end of terbinafine treatment, whereas one patient died due to an unrelated cause two weeks after terbinafine treatment had ceased. Case descriptions of all patients are provided in the Supplementary Information.

We also investigated expression of SQLE in primary tumour material (obtained prior to ADT) and hepatic metastases (obtained during second-line cabazitaxel therapy) in patient #1 by IHC. Here, we were able to confirm SQLE expression in the primary tumour tissue, which was sustained in the liver metastases (Fig. 8c). Metastases were also positive for PSA expression, indicating active AR signalling (Fig. 8c). Similarly, sustained SQLE expression was confirmed in tumour tissue

obtained by palliative desobstructive surgery through transurethral resection of the prostate (TUR-P) from patient #2 (Supplementary Fig. 8f).

## Discussion

Compared with other solid cancer entities, PCa relies heavily on androgens to support AR signalling and sustain unlimited growth and this dependency is often maintained even after progressing to the CRPC stage[13]. Recent studies have revealed distinct subtypes of CRPC that differ in their expression of AR and AR-responsive genes[40–42]. The molecular mechanisms that support AR activity in CRPC include the emergence of constitutively active splice variants of AR, with AR-V7 being the most frequent[43], or AR mutations that allow the binding of ligands other than testosterone, including corticoids[44] and interleukin-6[45]. Another mechanism of CRPC progression involves the intratumoral synthesis of testosterone from cholesterol or other steroid precursors[3]. This is evidenced by the observation that critical enzymes of the androgen synthesis pathway, including cytochrome P450 family 11 subfamily A member 1 (CYP11A1) that converts cholesterol to pregnenolone, are overexpressed in androgen-independent PCa and that metabolic intermediates of androgen synthesis can be detected in these tissues[4–7]. Furthermore, it has been proposed that prostate cancers that progress to lethal disease show increased expression of enzymes involved in de novo cholesterol biosynthesis[16]. Indeed SQLE, the second rate-limiting enzyme of the cholesterol biosynthesis pathway, was found to be overexpressed in advanced PCa and its expression predicts poor patient survival[15, 16]. SQLE is an important control point in the cholesterol biosynthesis pathway[14] and its protein stability is controlled via a cholesterol-dependent mechanism[46]. Interestingly, SQLE has already been studied in other cancer entities, including breast-, hepatocellular-, and neuroendocrine carcinomas, confirming its cancer-related role and possible druggability[32, 36, 47]. However, cancer-specific downregulation of SQLE has also been linked to cell survival through the antioxidant functions of squalene in cholesterol auxotrophic lymphomas[37].

Here, we show that increased expression of SQLE mRNA and protein is indicative of poor overall and disease-specific survival in a large unselected cohort of PCa patients. Moreover, high levels of SQLE protein detected in tumour areas with high Gleason scores were predictive of poor BCR-free and cancer-specific survival in a stratified cohort of high-risk PCa patients. We also demonstrate that *SQLE* is a target gene of miR-205, a micro-RNA

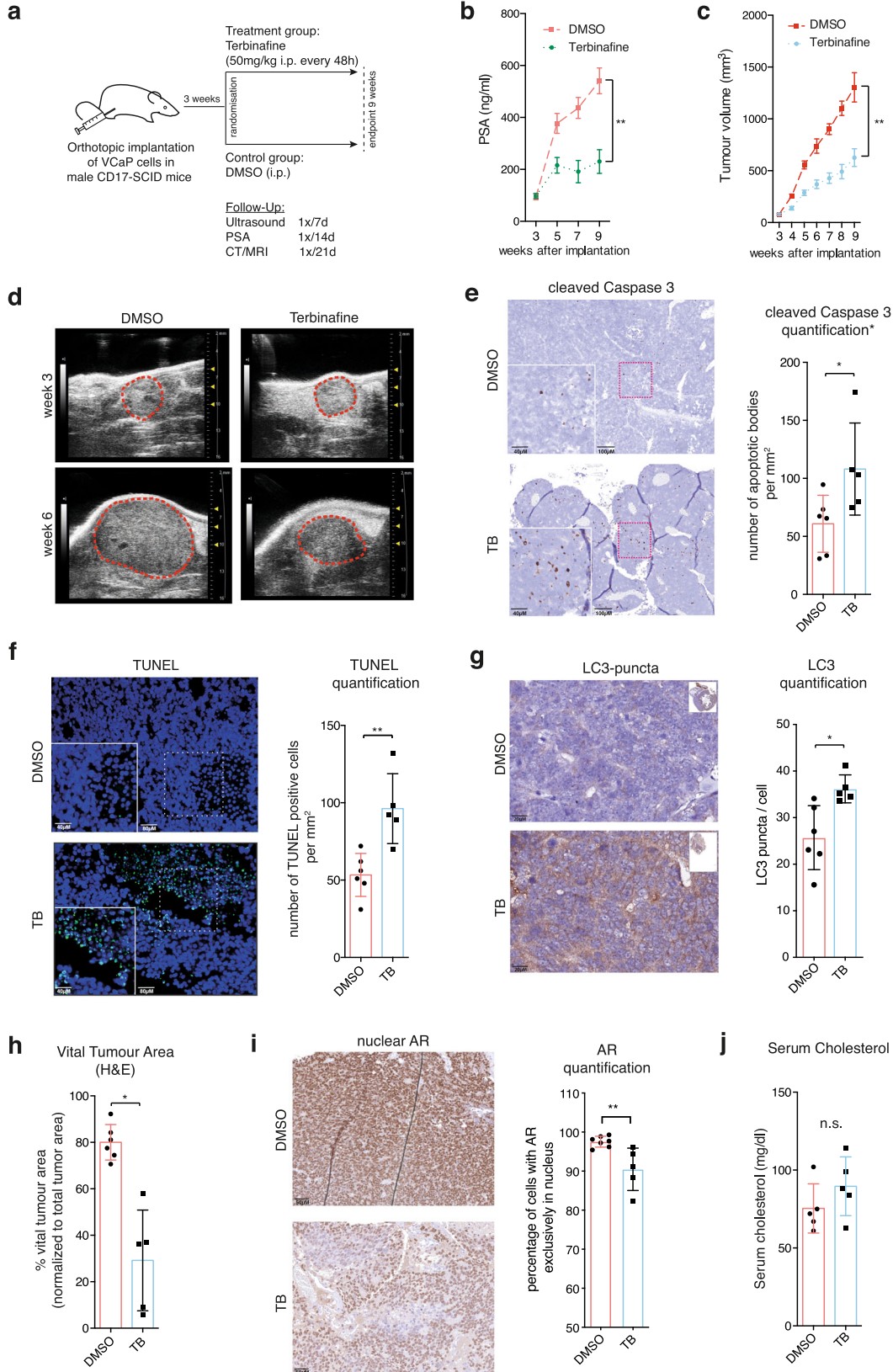

that is strongly downregulated in advanced PCa[17, 20], either through epigenetic silencing[23] or through loss of p63[24]. MiR-205 has previously been linked to transcriptional activation of tumour suppressor genes[22], downregulation of anti-apoptotic factors[19], and repression of genes linked to EMT[18] in PCa. Moreover, miR-205 was also shown to negatively regulate androgen signalling in

some PCa cell lines by binding to the 3′-UTR of the *AR* gene[21]. In this study, we identify *SQLE* as *a bona fide* target gene of miR-205. This is supported by the inhibition of a dual-luciferase reporter construct containing a conserved region from the 3′-UTR of the *SQLE* mRNA and the profound downregulation of *SQLE* mRNA levels following overexpression of miR-205 in a

**Fig. 7 SQLE inhibition suppresses tumour growth in an orthotopic mouse model of PCa. a** Study design of the orthotopic xenograft model used. **b** Time course of serum prostate-specific antigen (PSA) levels in vehicle- (DMSO, red line) or terbinafine (TB, blue line) treated mice at the indicated time points after orthotopic xenografting of VCaP cells. Data are presented as mean $+/-$ SD of DMSO ($N = 8$) and TB ($N = 7$) cohorts. ($**p < 0.01$, unpaired two-tailed Student's $t$-test). **c** Tumour volume in vehicle- (DMSO, red line) or terbinafine- (TB, blue line) treated mice at the indicated time points after orthotopic xenografting. Tumour-volume determination was carried out with high-resolution 3D ultrasonography. Data are presented as mean $+/-$ SD of DMSO ($N = 8$) and TB ($N = 7$) cohorts ($**p < 0.01$, unpaired two-tailed Student's $t$-test). **d** Representative ultrasonography images of vehicle- or terbinafine-treated mice at the indicated time points after orthotopic xenografting. **e** Left: representative IHC sections stained for the apoptosis marker cleaved Caspase 3 in vehicle- (DMSO) or terbinafine- (TB) treated mice. Right: quantitative analysis of cleaved Caspase 3-positive cells (=apoptotic bodies/mm$^2$) using automated image analysis. Data are presented as mean $+/-$ SD of DMSO ($N = 6$) and TB ($N = 5$) cohorts ($*p < 0.05$, unpaired two-tailed Student's $t$-test). **f** Left: Representative images for TUNEL staining of tumour tissue from vehicle- (DMSO) or terbinafine- (TB) treated mice. Green fluorescence indicates TUNEL-positive, apoptotic cells. Cellular nuclei were counterstained with DAPI. Right: quantitative analysis of TUNEL signal. Data are presented as mean $+/-$ SD of DMSO ($n = 6$) and TB ($n = 5$) cohorts. ($**p < 0.01$, unpaired two-tailed Student's $t$-test). **g** Left: Representative images for LC3 puncta staining of tumour tissue from vehicle- (DMSO) or terbinafine- (TB) treated mice. Right: quantitative analysis of LC3 puncta per cell using automated image analysis. Data are presented as mean $+/-$ SD of DMSO ($N = 6$) and TB ($N = 5$) cohorts ($*p < 0.05$, unpaired two-tailed Student's $t$-test). **h** Percent vital tumour area determined in hematoxylin- and eosin- (H&E) stained sections of tumours from vehicle- (DMSO) or terbinafine- (TB) treated mice using automated image analysis. Data are presented as mean $+/-$ SD of DMSO ($N = 6$) and TB ($N = 5$) cohorts. ($*p < 0.05$, unpaired two-tailed Student's $t$-test). **i** Left: Representative images for nuclear androgen receptor (AR) staining of tumour tissue from vehicle (DMSO) or terbinafine (TB) treated mice. Right: quantitative analysis of cells with AR exclusively in the nucleus using automated image analysis. Data are presented as mean $+/-$ SD of DMSO ($N = 6$) and TB ($N = 5$) cohorts ($**p < 0.01$, unpaired two-tailed Student's $t$-test). **j** Serum cholesterol levels determined at the endpoint in vehicle- (DMSO) or terbinafine- (TB) treated mice (n.s. = not significant, unpaired two-tailed Student's $t$-test, DMSO $n = 6$, TB $n = 5$).

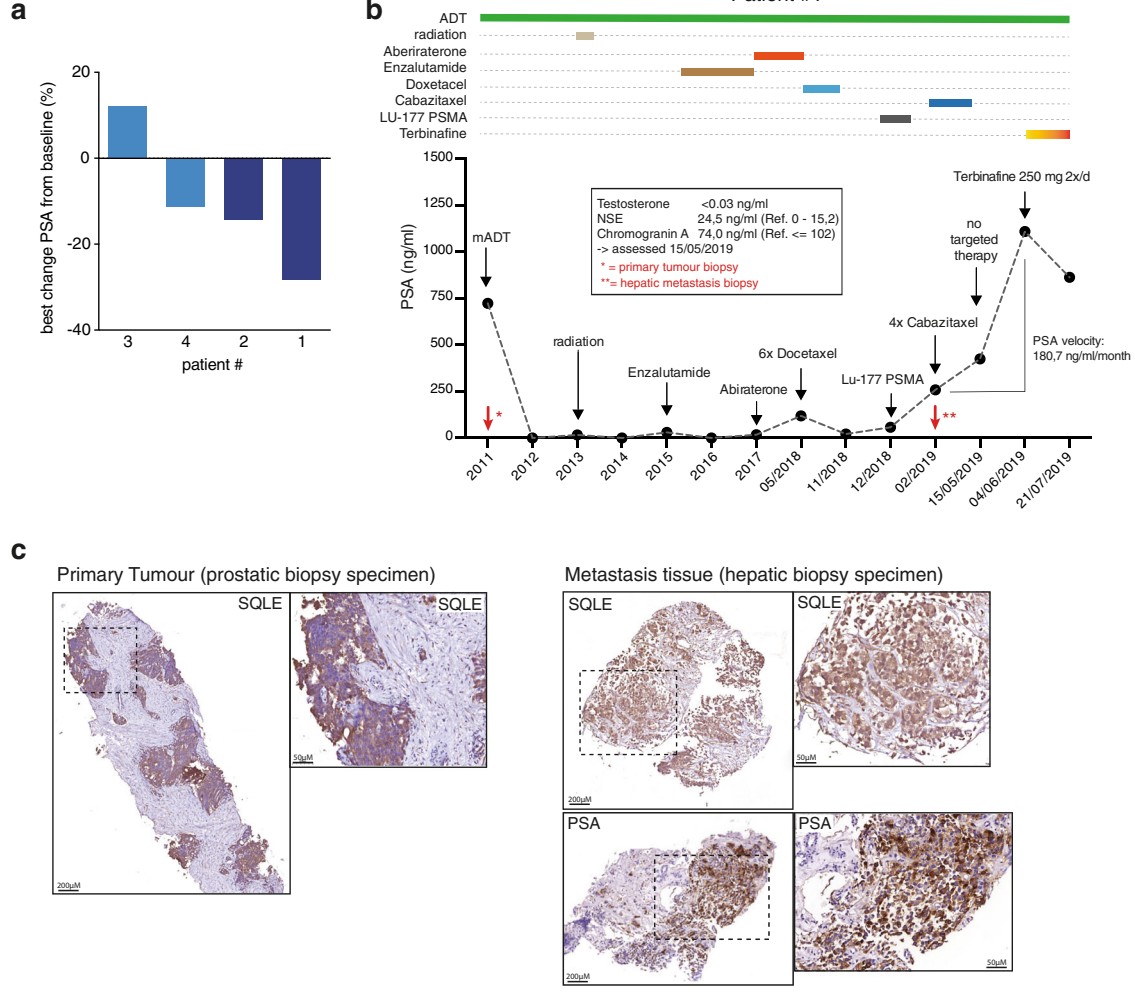

**Fig. 8 SQLE inhibition by standard-dose terbinafine delays biochemical progression in patients with metastatic PCa. a** Waterfall plot showing PSA percent change from baseline for four patients treated with either 500 mg/day (patients #1 and #4) or 250 mg/day (patient #2) for two weeks or 500 mg/day for six weeks (patient #3). **b** Serum PSA levels of patient #1 over the course of the disease. Respective treatment regimens are indicated above the graph. Inset: additional patient information. Red arrows indicate timepoints of tissue resection. **c** Representative images of IHC sections of tumour tissue stained for SQLE or PSA. Left: initial prostate biopsy specimen from 2011. Right: liver metastasis biopsy from May 2019.

panel of PCa cell lines. Moreover, we demonstrate a strong inverse correlation between miR-205 and *SQLE* expression in a cohort of aggressive PCa patients and enhanced expression of *SQLE* in a panel of primary PCa tumour samples and paired lymph-node metastases. In addition, we found that CRISPR/Cas9-mediated knockout of miR-205 resulted in robust over-expression of *SQLE* mRNA and protein, which was similar to the induction of *ZEB1*, a previously identified miR-205 target[28].

Using stable isotope-tracing experiments, we could demonstrate that over-expression of miR-205 or treatment with terbinafine reduce the synthesis of free cholesterol and cholesteryl esters, while overexpression of SQLE resulted in enhanced cholesterol synthesis. Moreover, treatment with chemical inhibitors of SQLE, terbinafine or NB-598, also efficiently blocked de novo cholesterol synthesis in PCa cells. Expression of cholesterol synthesis enzymes is controlled via regulated intramembrane processing of the sterol regulatory element-binding protein 2 (SREBP2)[48]. Thus, blocking cholesterol synthesis through miR-205 overexpression or SQLE inhibition could result in increased cholesterol uptake by increasing the expression of the low-density lipoprotein receptor (LDLR) or the cholesterol transporter SR-B1 Indeed, it has been suggested that cholesterol uptake via SR-B1 promotes cholesterol metabolism and steroid-hormone production in androgen-dependent PCa cells[30]. Interestingly, inhibition of cholesterol synthesis following miR-205 expression or SQLE inhibition did not cause a compensatory increase in cholesterol uptake in the cell lines used here, suggesting that targeting SQLE is an effective method to lower intracellular cholesterol availability. We also found that SQLE inhibition resulted in a dramatic decrease in newly synthesised cholesterol esters, most likely caused by a reduction in *SOAT1* expression. As SOAT1 is the target of mitotane, a drug used for the treatment of adrenocortical carcinoma, it would be interesting to investigate whether SOAT1 inhibition could also interfere with cholesterol metabolism in PCa cells[49].

We also demonstrate that silencing of *SQLE* resulted in a solid inhibition of proliferation in PCa cells. Importantly, proliferation was restored by addition of cell-permeable cholesterol, confirming that cholesterol is a rate-limiting metabolite for PCa proliferation. A similar inhibition of proliferation was also observed following inhibition of SQLE by terbinafine or NB-598. Indeed, inhibition of proliferation following SQLE inhibition was stronger in the AR-positive PCa cell lines used in our study. Moreover, addition of the synthetic androgen R1881 restored proliferation following miR-205 overexpression or SQLE inhibition only in AR-positive PCa cells, suggesting that the miR-205/*SQLE* axis controls androgen signalling by regulating cholesterol synthesis. Notably, cell line models of resistance towards the nonsteroidal antiandrogen enzalutamide or the CYP17A1 inhibitor abiraterone were still sensitive to terbinafine treatment. Since one of the major resistance mechanisms toward these drugs involves alternative activation of the glucocorticoid receptor (GR), this finding indicates that interfering with cholesterol synthesis still presents a vital option in treatment-resistant PCa. However, we also recognise that some forms of CRPC, most notably neuroendocrine or "double-negative" metastatic PCa, may be insensitive to SQLE inhibition due to the activation of alternative pathways[40, 42].

We could also show that overexpression of miR-205, silencing of SQLE, or treatment with SQLE inhibitors attenuated AR-dependent transcriptional activity and reduced the expression of AR target genes in androgen-dependent and androgen-sensitive PCa cell lines. We did not observe consistent changes in *AR* mRNA levels in response to miR-205 induction in the C4-2 or LNCaP cells used in our study, making it unlikely that the observed effects on AR reporter activity and AR target gene

expression were due to a direct effect of miR-205 on the *AR* transcript[21]. Moreover, transactivation of the AR and expression of AR target genes following SQLE inhibition was partially restored by addition of cell-permeable cholesterol, confirming that SQLE blockade interferes with AR activity by reducing cholesterol availability in PCa cells. Interestingly, *SQLE* itself has been identified as an AR target gene[50], suggesting a feed-forward loop that promotes androgen signalling in PCa.

Encouraged by these findings, we sought to confirm the importance of the miR-205/SQLE axis in an in vivo model of PCa. For this, we chose an established orthotopic implantation model using the VCaP cell line[51], as these cells maintain high AR expression and activity[52, 53]. Using this model, we observed that terbinafine treatment efficiently lowered serum PSA and reduced tumour growth, with tumours from treated mice showing evidence of enhanced apoptosis. It has been suggested that terbinafine, in addition to its function as an inhibitor of SQLE, also induces autophagy in hepatocellular carcinoma through an unknown mechanism[36]. We therefore investigated whether terbinafine treatment also affects the autophagy machinery in PCa tumours. Indeed, we observed an increased number of LC-3 puncta in terbinafine-treated tumours, indicative of increased autophagy. Further investigation will be required to define the contribution of autophagy in the inhibition of tumour growth by this drug. We also found that terbinafine reduced the abundance of tumour cells with nuclear AR, suggesting that inhibition of SQLE interferes with AR signalling also in vivo. Importantly, terbinafine treatment did not alter the levels of circulating cholesterol, suggesting that the inhibition of tumour growth is caused by the specific reduction of intratumoral cholesterol synthesis rather than a systemic effect on cholesterol homoeostasis. Moreover, terbinafine did not induce any adverse side effects, evidenced by unchanged serum AST and morphological integrity of liver and kidney tissue.

We also report here on a first-in-man off-label use of terbinafine in four patients with rapidly progressive end-stage metastatic PCa after multiple prior treatment modalities, including abiraterone and enzalutamide. Despite the very short course of treatment and the lack of possibility to restage the patient with sectional imaging, we observed a drop in PSA levels in three of the four patients, suggesting that inhibition of cholesterol synthesis by SQLE blockade can reduce biochemical markers of disease progression in PCa. This is remarkable, considering the rapid PSA increase before terbinafine treatment in all patients. It should also be noted that one patient had been on continuous treatment for several years with atorvastatin, an inhibitor of HMG-CoA reductase, but still showed response toward terbinafine, indicating that targeting SQLE could still be effective after cholesterol-lowering drug treatment. Histological analysis of primary tumour tissue and metastases from two of the patients prior to treatment revealed high levels of SQLE expression, confirming the presence of the drug target. Furthermore, SQLE expression was maintained in the metastases together with signs of AR activity, indicated by PSA staining.

Taken together, our findings reveal that *SQLE* is a target of miR-205 in PCa and a critical driver of disease progression and malignancy by the regulating cholesterol biosynthesis and AR activation. We also demonstrate that competitive inhibition of SQLE with the FDA- and EMA-approved drug terbinafine reduces tumour growth in a murine PCa model and provide preliminary evidence on the efficacy of the drug in PCa patients. Interestingly, a recent population-based cohort study reported an up to 47% reduction in the risk of disease-specific death for PCa patients receiving oral terbinafine treatment[54]. Based on these data, we believe that SQLE could be a target for the treatment of

advanced PCa and that inhibitors of SQLE should be evaluated in clinical trials.

## Methods

**Tissue culture and reagents.** The following human cell lines were obtained from ATCC (American Type Culture Collection, Chicago, IL, USA) and used at low passage: LNCaP (ATCC cat.# 1740), C4-2 (ATCC cat.# 3314), PC-3 (ATCC cat.# 1435), DU145 (ATCC cat.# HTB-81), RWPE-1 (ATCC cat.# 11609) and HEK293T (ATCC cat.#3216). VCaP cells (ATCC cat# 2876) were kindly provided by Johannes Linxweiler (Medical University Homburg/Saar, Germany). LNCaP Abl cell lines and their Abiraterone- (LNCaP Abl/Abi) and Enzalutamide-resistant (LNCaP Abl/Enza) clones were previously published and kindly provided by Martin Puhr (Medical University of Innsbruck, Austria)[8]. Cell lines were cultured in DMEM (VCaP) or RPMI 1640 (all others) supplemented with 10% foetal calf serum (FCS), 1x nonessential amino acids (NEAA), 2 mmol/L glutamine, and 100 U/mL Penicillin/Streptomycin (all PAA, Pasching, Austria). LNCaP-Abl cells and derivatives were cultured in the same medium but with charcoal-stripped serum (PAA, Pasching Austria) and additionally supplemented with 2.5 mg/mL D-glucose and 8 μM of the respective drugs (abiraterone or enzalutamide). C4-2 and VCaP cells were also cultured in charcoal stripped serum. For lenti- and retroviral packaging, HEK293T (ATCC cat.# 3216) and Phoenix Ampho (ATCC cat.# 3213) cells were used. These cells were cultured in DMEM supplemented with 10% FCS, 2 mmol/L glutamine and 100 U/mL Penicillin/Streptomycin. All cells were grown at 37 °C in a humidified incubator at 5% $CO_2$ and regularly tested for the absence of mycoplasma. Terbinafine, abiraterone acetate, YM-53601, 5α-dehydrotestosterone (DHT), R1881, and cell-permeable cholesterol were all from Sigma, NB-598 and enzalutamide from Selleckchem.

**Analysis of cancer genome atlas data and code availability.** Raw gene expression data of primary PCa specimens were extracted from TCGA (PRAD primary, PanCancer Atlas) datasets using cBioPortal (http://cbioportal.org). For correlation of *SQLE* expression and clinical characteristics, R 3.10 (http://www.r-project.org) and the packages TCGA2STAT (https://cran.r-project.org/web/packages/TCGA2STAT/index.html) and plyr (https://cran.r-project.org/web/packages/plyr/index.html) were used to extract RPKM-normalised RNASeq2 data and clinical variables from the PRAD primary tumour collective. Data processing and visualisation were done using a personalised R-Script (see Supplementary Software 1). Copy number and gene expression data of primary and metastatic PCa were extracted from TCGA (PRAD primary, PanCancer Atlas and PRAD metastasis, SU2C Dream Team). Z score (RPKM) gene expression for *HMGCS1* and *FDFT1* was correlated to the expression of *SQLE* and displayed in dot plots using Spearman correlation assuming nonnormally distributed data. FPKM: fragments per kilo-base per million mapped reads.

**Human tissue samples and in-house patient cohorts.** For tissue analyses, we used two independent tissue microarrays (TMAs) of curative-intent radical prostatectomy specimen. The first tissue microarray (TMA1) comprised consecutively collected, formalin-fixed, and paraffin-embedded tumour samples of 453 PCa patients diagnosed in the Department of Pathology, University Hospital Erlangen from 1999 to 2010. Clinical recurrence was defined either as histologically proven local recurrence or distant metastasis confirmed by computed tomography or bone scan. The assembly of this TMA is based on the approval of the Ethics Committee of the University Hospital Erlangen (No. 3755). The second TMA (TMA2) comprised 179 patients who had undergone radical prostatectomy between 1987 and 2005 at the Community Hospital of Karlsruhe, Germany and the University Hospital Leuven, Belgium[55], and were considered high-risk disease according to the D'Amico risk-classification system[56]. The assembly of this second TMA was approved by the respective local Ethics Committees (No. 59/04 and B322201214832). Patients in both cohorts were followed every three months for the first two years after surgery, every six months in the following three years, and annually thereafter. Clinical failure was defined either as histologically proven local recurrence or distant metastasis confirmed by computed tomography, bone scan, or PET/CT scan. For both cohorts, recurrence-free survival (RFS) was defined as time from radical prostatectomy to clinical failure, whereas OS was defined as time from radical prostatectomy to death of any cause. For qPCR analysis of miR-205 and *SQLE* expression, a collection of FFPE samples of radical prostatectomy specimen ($N = 59$), lymph-node metastases ($N = 22$), and benign prostatic tissues ($N = 10$) obtained from TUR-Ps at the Department of Urology and Paediatric Urology of the Julius-Maximilians-University Medical Centre Würzburg, Germany, between 2009 and 2012, were included in the study. The study was approved by the Ethical Review Board of the Julius-Maximilians-University Würzburg (no. 136/2008) and written informed consent was obtained from all patients who participated in the study. Clinicopathological and outcome data for all cohorts are available in Table 1.

**Immunohistochemistry (IHC) and automated detection.** Paraffin-embedded sections of human and murine samples were cut into 4-μm sections with a microtome. TMAs were provided in 2-μm sections. After deparaffinization and rehydration, antigen retrieval was performed with citrate buffer (pH 6.0) in a microwave oven for 6 min. Sections were stained with anti-SQLE (Proteintech cat# 12544-1-AP), anti-Ki67 (SP6, Thermo Fisher cat# RM-9106), anti-cleaved Caspase 3 (CST cat# 9664), anti-AR (clone AR441, Dako M3562), anti-PSA (Agilent/Dako cat# M3620) or anti-LC3 (clone 5F10, Nanotools 0231-100) in 1% BSA in PBS at 4 °C and biotinylated secondary antibody (Vector Laboratories). Dilution for all antibodies was 1:100. Slides were developed with 3'3-diaminobenzidine (CST) and counterstained with Gilmore 3 hematoxylin. The specificity of the SQLE antibody was established using a panel of human tissue samples. Slides were scanned in 40x resolution using a Pannoramic SLIDE II slide scanner and analysed using Case Viewer software 3.2 (3DHistech). SQLE staining in TMAs was independently scored by several researchers (MR, CK, and AS) blinded to clinicopathological and outcome data. Potentially discordant scores were resolved by joint review. The percentage of cell immunoreactivity (0–100%, divided into 4 groups: 0–25%, 25–50%, 50–75%, and 75–100%) and the staining intensity (score 0–3, with 0 indicating negative; 1, weak; 2, moderate; and 3, strong) were recorded for each. SQLE expression scores were calculated as the product of the percentage immunoreactivity in the staining intensity, resulting in scores between 0 and 12. Automated image analysis was performed with QuPath (https://qupath.github.io/)[57]. Images were imported and tumour tissue was selected using annotation gating. For Ki-67 and cleaved Caspase 3, analysis was carried out in positive cell detection mode using optical density sum with a requested pixel size of 0.5 μm. Default parameters were used for nuclei identification except for minimum area, which was set to 15 μm². Intensity-parameter threshold was set to 0.05 for Ki-67 and 0.1 for cleaved caspase 3. Score compartment was "nucleus: DAB mean" and thresholds 1–3 were 0.15, 0.4, and 0.6 for Ki-67 staining and 0.1, 0.4, and 0.6 for cleaved caspase-3 staining, respectively. The mean number of processed nuclei per sample was $2.08 \times 10^5$ (SEM $0.52 \times 10^5$) for Ki-67 and $2.03 \times 10^5$ (SEM $0.54 \times 10^5$) for cleaved caspase 3. For androgen-receptor nuclear localisation analysis, the intensity-threshold mode "nucleus: DAB OD max" with a threshold of 0.2 was used. The mean number of processed nuclei per sample was $1.35 \times 10^5$ (SEM $0.17 \times 10^5$). For LC-3 puncta, the subcellular detection function was used. Cells were detected in standard mode in 10 random 10x high-power fields of all tumours (standard settings in ODS mode with a requested pixel size of 0.5 μm, a threshold of 0.1, and cell expansion setting of 5 μm with a minimum of $1 \times 10^4$ detected cells per high-power field). Next, subcellular detection of LC-3 spots was carried out with a detection threshold of 0.1, splitting data by intensity without smoothing with an expected spot size of 0.8 μm² (range: minimal spot size 0.2 μm² and maximal spot size 2 μm²). The detected spots per high-power field were then normalised to the detected cell number in the same field to determine the average spot per cell ratio.

**Dual luciferase assay.** 3'-UTR-binding sites of SQLE mRNA and miR-205 were identified using online tools (mircrorna.org, targetscan, miRanda) and cloned into the XbaI site of the pMIR-REPORT vector (Ambion) after annealing and phosphorylation with T4 PNK (NEB Biolabs, Ipswich, MA, USA). Sequences were as follows: SQLE_XBAI_for 5'-CAATTTGCAATTTAAAATGAAGGGGTTAAATA AGTTAGACAT-3' and SQLE_XBAI_rev 5'-CTAGATGTCTAACTTATTTAACC CCTTCATTTTAAATTGCAAATTGAGCT-3'. The reverse oligo for mutagenesis was SQLEmut_XBAI_rev 5-CTAGATGTCTAACTTATTTAACCTTCGTGCTTA AATTGCAAATTGAGCT-3'.

About 250 ng of the resulting vector was then cotransfected with 50 nmol/l miR-205 mimic or control miR mimic (both Ambion) into LNCaP cells using Lipofectamine 2000. About 50 ng of cotransfected pRL-TK renilla luciferase plasmid was used as control. Forty-eight hours after transfection, luminescence of firefly and renilla luciferase was assessed using a TECAN SPARK luminometer. Data were presented as relative light units (RLU) by dividing the luminescence of firefly luciferase by that of renilla luciferase.

**Generation of miRNA-205 overexpression and SQLE shRNA cell lines.** The miR-205 core sequence was amplified by PCR with the following custom primers: miR-205_fw (5'- TACAATACTCGAGAAGGTATATAAAGATCCTCAGACAAT CC-3') and miR-205_rev (5'- TTAGATGAATTCTAGCCCCTTGAAGTGTCA GCTCCATGCCTC-3') using Q5 High-Fidelity Polymerase (NEB Biolabs, Ipswich, MA, USA) and cloned into the lentiviral LT3GEPIR vector (Addgene#11177) using EcoRI and XhoI restriction sites. Two independent shRNA sequences targeting SQLE (http://splashrna.mskcc.org) were also cloned into LT3GEPIR using the EcoRI and XhoI restriction sites. Sequences were as follows:

SQLEsh2393 5'- TGCTGTTGACAGTGAGCGCGCCTGTTTTCTTTATTTCA AATAGTGAAGCCACAGATGTATTTGAAATAAAGAAAACAGGCTTGCCTA CTGCCTCGGA-3' and SQLEsh2921 5'- TGCTGTTGACAGTGAGCGCACAGT TTTTCTTTTGAATTTATAGTGAAGCCACAGATGTATAAATTCAAAAGAA AAACTGTACGCTACTGCCTCGGA-3'. Common primers for amplification of these shRNA oligos were EcoRI_fw 5'- TACAATACTCGAGAAGGTATATTGC TGTTGACAGTGAGCG-3' and XhoI_rev 5'-ATTCATCACAATTGTCCGCGTC GATCCTAGG-3'. All constructs were confirmed by sequencing.

Lentiviruses were produced by cotransfecting HEK293T cells with shRNA plasmids and the packaging plasmids psPAX.2 (Addgene Cat#12260) and pMD.G2 (Addgene Cat#12259). Supernatants containing lentiviruses were collected 48 h after transfection, mixed with polybrene (8 μg/mL) (Sigma), and used for infection.

Fresh medium containing puromycin (2 μg/mL) (Sigma) was added after 24 h and cells were selected as stable pools before use.

**RNA extraction and quantitative real-time PCR (qRT-PCR)**. RNA extraction from FFPE tissues was performed using the total RNA extraction kit (Qiagen) and cDNA was produced with the IM-PROM-II reverse transcription kit (Promega). RNA extraction from cells was performed using TRIzol (Thermo Fisher). MiR-205 and *MIR205HG* lnc-RNA expression was assayed using TaqMan miR assay kits on an Applied Biosystems 7900HT system (Life technologies, Carlsbad, MN, USA). snRNA RNU6b and SNORA74A, respectively, were used for normalisation by the comparative ΔCt method. Fold changes in micro-RNA expression between samples and controls were determined by the $2^{-\Delta\Delta Ct}$ method. mRNA levels of *SQLE*, *KLK3*, *ZEB1*, *VIM*, and *CDH1* were determined by qRT-PCR using validated $RT^2$ qPCR primer assays (Qiagen). Actin beta (*ACTB*) was used for normalisation. For SOAT1 qPCR, a TAQMAN assay (cat# 4351372, Thermo Fisher) was used. QRT-PCR reactions were performed in triplicate on three biological independent samples. Primer information is provided in Supplementary Table 3.

**CRISPR/Cas9 knockout of miR-205**. We used lentiCRISPRv2 (Addgene #52961) and pLKO.5.sgRNA.EFS.GFP (Addgene #57822) to transduce catalytically active Cas9 and two separate sgRNAs each into the used cell lines. sgRNA sequences were designed using DESKGEN cloud (http://deskgen.com), phosphorylated with T4 PNK, and annealed. Sequences were as follows:

gRNA1_fw 5′-CACCGAGGATCTTTTTCTCAGAGGG-3′
gRNA1_rev 5′-AAACCCCTCTGAGAAAAAGATCCTC-3′
gRNA2_fw 5′-CACCGGAAGCACATGGATTGTCTG-3′
gRNA2_rev 5′-AAACCAGACAATCCATGTGCTTCTC-3′
gRNA3_fw 5′-CACCGGGGGTATGAGACAGACTCCGG-3′
gRNA3_rev 5′-AAACCCGGAGTCTGTCTCATACCCC-3′

The pLKO.5.sgRNA.EFS.GFP vector was cut with BsmBI and the annealed oligos were ligated as described (http://genome-engeneering.org). Lentiviral particles were produced in HEK293T cells as described above. PC-3 and LNCaP cells were infected using combinations of two gRNA expression vectors. Stable clonal pools were selected using puromycin (2 μg/mL) and sorted for top 25% GFP expression (using a BD FACSAria III cell sorter).

**Establishment of SQLE-overexpressing PCa cells**. SQLE cDNA including the Kozak sequence was amplified from LNCaP cDNA using the following primers: SQLE_fw 5′-ACCTTAAACCCCACTCGAGCA -3′ and SQLE rev 5′-CACAAAT GGTTCCCCTTTAAGC-3′. The amplicon was then used as template for restriction-free cloning into pBabe puro (Addgene #1764) using 50 ng of the SQLE template and the following primers: SQLE_mega_fw 5′- TGTGGTGGTACG TAGGAATTCGCGGAACCATGTGGACTTTTCTG-3′ and SQLE_mega_rev 5′-A TTCCACAGGGTCGACCACTGTTTAATGAACCATATACTTCATTTCT-3′. Amplification was accomplished with Q5 Taq polymerase (NEB) using the following protocol: annealing 58 °C, 20 sec, extension 72 °C, 45 sec, and final extension 5 min, 34 cycles. About 60 ng of pBabe puro plasmid was then mixed with 426 ng of gel-purified megaprimer and the following protocol was carried out: annealing 60 °C, 20 sec, extension 72 °C, 2:20 min, and final extension 5 min, 34 cycles. The PCR product was then digested using DpnI and transformed into competent bacteria. All plasmids were validated by sequencing. Retroviral particles were produced using Phoenix-ampho cells and used to infect C4-2 cells. Stable pools were selected with puromycin (2 μg/mL).

**Western blotting**. Cells were lysed in lysis buffer (1% Triton X100, 50 mM Tris pH 7.5, 300 mM NaCl, 1 mM EGTA, 1 mM DTT, and 1 mM NaVO4) with protease inhibitors for 30, cleared by centrifugation, and quantified using BCA assay (Biovision). For detection of LC3, cells were washed with PBS and lysed in Phosphosafe (Novagen) for 5 min, then scraped, and cleared by centrifugation. Proteins were separated on SDS-PAGE and blotted onto PVDF membrane (Immobilon), blocked with blocking solution (LI-COR), and incubated with primary and secondary antibodies. Signals were detected on an Odyssey scanner (LiCor) or ChemiDoc (BioRad) or iBright FL-1000 (Thermo Fisher). The following antibodies were used: anti-SQLE (Proteintech, 12544-1-AP, Dilution 1:1000), anti-ZEB1 (Novus Biologicals, NBP-05987, Dilution 1:2000), anti-LC3 A/B (CST, #8457, Dilution 1:2000), anti-PARP1 (Abcam, ab32064, Dilution 1:1000), anti-b-actin (CST, #8457, Dilution 1:1000), and anti-vinculin (Sigma, V9131, Dilution 1:15.000). Secondary antibodies were from Dako: Polyclonal Swine Anti-Rabbit ig-HRP (Ref P0399) and Polyconal Rabbit Anti-Mouse ig-HRP (Ref P0260) both used in a dilution of 1:1000.

**Crystal violet staining**. After incubation, cells were washed twice with PBS and fixed for 20 min in 3.7% paraformaldehyde (PFA). Cells were washed again and stained with 0.5% crystal violet solution for 1 h. Plates were rinsed, dried, and extracted using 10% acetic acid. Absorbance was measured at 550 mm in triplicate and relative staining was calculated in comparison with untreated controls.

**Transwell migration assay**. Assays were performed using 12-well transwell plates with 12 μm pore inserts (Corning). Inserts were coated with 10 μg/cm² collagen I (Sigma) and cells were seeded in serum-free medium in a density of $5 \times 10^4$ (PC3) or $1 \times 10^5$ (C4-2) cells per well. Medium with 10% serum was used as a chemoattractant. Eighteen hours after seeding, migrated cells were stained with crystal violet and quantified.

**Stable isotope labelling and analysis of cholesterol and cholesterol esters**. Cell were incubated in RPMI medium without glucose and glutamine supplemented with 4 mM [U¹³C]-glutamine, 25 mM [U¹³C]-glucose, and 1 mM [U¹³C]-acetate (all Sigma), as indicated. After the indicated incubation time, cells were washed with cold ammonium acetate (154 mM), snap-frozen in liquid nitrogen, and scraped off in 0.4 mL pf ice-cold MeOH/H2O (80/20 v/v). The suspension was transferred to a reaction vial, mixed vigorously, and centrifuged for 2 min at 16,000 g. Supernatants were extracted twice with 0.4 mL of hexane. The combined extracts were collected and taken to dryness under a stream of nitrogen gas at 42 °C. Residues were dissolved in 150 μL of hexane and transferred to Strata® SI-1 columns (Phenomenex). Cholesterol esters were eluted with 750 μL of hexane. Columns were washed with 500 μL of hexane/acetic acid ethyl ether (18/1, v/v) and 500 μL of hexane/acetic acid ethyl ester (9/1, v/v). Cholesterol was eluted by applying 500 μL of hexane/acetic ethyl ester (9/1, v/v). Cholesterol and cholesterol ester fractions were taken to dryness under a stream of nitrogen gas at 42 °C. The resulting residues were dissolved in 50 μL of isopropanol and taken for LC/MS analysis. For hydrolysis of cholesterol esters, fraction 1 was resuspended in 500 μL of 0.3 M KOH in MeOH/H2O (9/1, v/v), mixed vigorously, and incubated at 80 °C overnight. After cooling down to room temperature, samples were extracted twice using 400 μL of hexane. Combined supernatants were taken to dryness under a stream of nitrogen at 42 °C and resuspended in 150 μL of hexane. Cholesterol was then isolated as described above.

For LC/MS analysis of cholesterol and cholesterol esters, 5 μL of sample was applied to a C8 column (Acclaim RSLC, 2.2-μm particles, 50 × 2.1 mm, Thermo Scientific). Cholesterol-containing fractions were separated with mobile-phase buffer A consisting of 0.1% formic acid in CH3CN/H2O (10/90, v/v) and solvent B consisting of 0.1% formic acid in CH3CN/H2O (90/10, v/v). After sample application at 40 °C, the LC gradient programme was 20% solvent B for 2 min, followed by a linear increase to 100% solvent B within 5 min, then maintaining 100% solvent B for 12 min. The flow rate was maintained at 350 μL/min. The eluent was directed to the ESI source of the MS instrument from 7.0 min to 13.0 min after sample injection. MS analysis was performed on a Q-Exactive mass spectrometer (Thermo Scientific) applying the following scan and HESI source parameters: scan ranges: for cholesterol 365–400 m/z (positive mode); resolution: 70,000; AGC-Target 1E6; maximum injection time: 200 ms; sheath gas: 30; auxiliary gas: 12; sweep gas: 3; aux gas heater temperature: 120 °C; spray voltage: 2.5 kV; capillary temperature: 320 °C; S-lens RF level: 55.0. Signal determination and quantitation were performed using TraceFinder™ Software Version 4.1 (Thermo Fisher).

**Annexin V/PI-FACS**. For Annexin V/PI-FACS, the supernatant of the cells was collected and combined with cells harvested by trypsinisation. The pellet was washed with cold PBS and resuspended in 100 μL of 1x Annexin V-binding buffer (10 mM HEPES, pH 7.4, 140 mM NaCl, and 2.5 mM CaCl2) with Annexin V/ Pacific Blue dye. After incubation for 15 min at RT in the dark, 400 μL of 1x binding buffer containing PI (54 μM) was added and the samples were stored cold and dark until analysis. Analysis was performed on a BD FACSCanto II flow cytometer using BD FACSDIVA™ Software. The gating strategy used is provided in Supplementary Fig. 5c.

**AR reporter assays**. We used an established sequence comprising six androgen-response elements (ARE) described before[38]. AREs in bold: 5′-GGATCAGGGAG TCTCTCAGGATCAGGGAGTCTCAGTAGAACAGCAAGTGCTGTTAGAACA GCAAGTGCTGTAAGAACAGCAAGTGCTGCTAGAACAGCAAGTGCTG CGC-3′. This sequence was cloned into pGL3-Enhancer luciferase reporter plasmid (Promega) using restriction-free cloning and the following primers for megaprimer creation: ARE_pGL3_MP_fw
5′-TGCTAGCCCCGGGCTCGAGAGGATCAGGGAGTCTCTCAGGA-3 and
ARE_pGL3_MP_rev 5′-CCAACGATACCGGAATGCCAAGCTTGCGCAGC ACTTGCTGTTC-3′. About 60 ng of pGL3 enhancer plasmid was then mixed with 420 ng of gel-purified megaprimer and the following cycler settings were applied: annealing 60 °C, 20 sec, extension 72°, 45 sec, and final extension 72 °C, 5 min, 34 cycles. The PCR product was then digested using DpnI and transformed into competent bacteria. All plasmids were validated by sequencing.

For the dual-luciferase assay, $1 \times 10^5$ cells were seeded in 12-well plates in triplicate and transfected with 500 ng of GL3-Enhancer-6xARE and 100 ng of pRL-TK renilla control (Promega). About 24 h after transfection, cells were treated with the indicated compounds for 48 h, lysed, and assayed with the dual-luciferase assay system (Promega) using a TECAN SPARK luminescence reader. Data of three independent experiments were presented as relative light units (RLU) by dividing the luminescence of firefly luciferase by the luminescence of renilla luciferase.

**PSA detection in medium supernatants**. Conditioned media were collected and analysed using the Access 2 Immunoassay System (Beckman Coulter). Standardisation was achieved by normalising PSA values to the relative cell number as determined by crystal violet staining.

**Orthotopic implantation of PCa cells**. All animal experiments were approved by the local governmental animal care committee (Landesamt für Verbraucherschutz des Saarlandes, approval number 30/2015) and conducted in accordance with the German legislation on protection of animals and the National Institutes of Health Guide for the Care and Use of Laboratory Animals (NIH Publication #85-23 Rev. 1985). The used orthotopic xenograft model has been described previously[51]. Mice were fed a standard chow diet and housed in a temperature-controlled room under a 12-h light–dark cycle under pathogen-free conditions. In brief, 16 male immunodeficient SCID mice (CB17/lcr-$Prkdc^{scid}$/lcrlcoCrl, 8–10 weeks old, Charles River Laboratories, Sulzfeld, Germany) received an orthotopic (left–anterior prostate lobe) injection of $5 \times 10^5$ VCaP cells per animal suspended in a 1:3 matrigel-medium mixture (10 µL per animal in total). After orthotopic tumours reached a size detectable by high-resolution 3D ultrasound, mice were randomised 1:1 into treatment (terbinafine 50 mg/kg in 25 µL diluted DMSO i.p. every 48 h, $n = 7$) and control (25 µL of diluted DMSO i.p. every 48 h, $n = 8$) groups. One mouse failed to develop a tumour and was removed from the study. Followup was carried out with high-resolution 3D ultrasound (tumour volume) every week, serum PSA measurements every two weeks, and micro-CT and 9.4 T MRI scans every three weeks for a total of nine weeks, after which mice were sacrificed. Tumours were harvested and used for IHC and TUNEL staining. Serum sampled at sacrifice was used for the determination of circulating cholesterol and liver transaminases (AST) using standard laboratory assays of the University Hospital Centre Homburg/Saar, Germany.

**Small-animal magnetic resonance imaging (MRI)**. Six weeks after starting treatment, animals were examined in a horizontal-bore 9.4-T MRI animal scanner (Biospec Avance III 94/20, Bruker Biospin GmbH, Ettlingen, Germany), equipped with a BGA12S gradient system (maximum field strength 675 mT m$^{-1}$, linear inductive rise time 130 µs, and maximum slew rate 4673 mT m$^{-1}$ s$^{-1}$) and ParaVision 6.0.1 software. During magnetic resonance experiments, the animals were maintained under general anaesthesia with 1–2.5% isoflurane in 99–97.5% oxygen provided with an oxygen concentrator at a flow rate of 1.25 L per min (Oxygen Concentrator, DeVilbiss Healthcare LLC, Sommerset, PA, USA). Breathing rate, heart rate, and body temperature of the mice were continuously controlled during the examination[58]. MRI was performed with a linear polarised coil for imaging of the mouse abdomen, with an inner diameter of 38 mm (Bruker Biospin GmbH). MR experiments were performed with prospective triggering to animal respiration. Scout images for control of animal placement within the magnet centre and adjustment of slice and voxel geometry were acquired in the x, y, and z planes with a fast low-angle shot (FLASH)-based 3D localiser sequence. B0 inhomogeneities were compensated by employing field map-based approaches implemented in the scanner software. Field maps were computed from two sets of images recorded with a 3D double-echo multi-gradient-echo (MGE) sequence. Artifacts were prevented by selecting the smallest echo time possible and adjusting the second echo time to the Larmor frequencies of both fat and water, to give the same phase conditions as for the first echo. For MRI experiments, computational field map-based shimming was performed on the entire volume of the slice packages selected, followed by adjustment of Larmor frequency and reference RF pulse strength. T2-weighted imaging was performed with a rapid acquisition relaxation-enhanced (RARE) protocol with a RARE factor of 8 and a flipback pulse, while diffusion-weighted imaging (DWI) was established with an echo-planar imaging (EPI) sequence, as a diffusion-trace experiment. Diffusion-gradient strengths of 100, 250, 500, and 1000 mT m$^{-1}$ were applied in x, y, and z directions. DWI datasets were used to calculate apparent diffusion coefficient (ADC) maps as described before[58]. Tumour sizes and ADC values were recorded directly from the Image Display module of the scanner software. Both imaging experiments were performed with fat-saturation mode activated.

**TUNEL staining**. For TUNEL staining, 4-µm tissue sections were heated in citrate buffer (pH 6.0) for 2 min. TUNEL reactions were developed for 1 h (In Situ Cell Death Detection Kit, Sigma) and counterstained with DAPI (Sigma). Sections were covered with Fluoromount (Sigma) and stored at 4 °C in the dark until final analysis with a phase-contrast microscope (Leica). TUNEL-positive cells/mm$^2$ were assessed by counting TUNEL positive cells in 10 randomly selected high-power fields (11.52-mm$^2$ surface) in five DMSO and terbinafine-treated mice each.

**Terbinafine treatment in patients with advanced CRPC**. For the individual clinical interventions using terbinafine (Lamisil), four patients with late-stage, heavily pretreated prostate cancers, who either had exhausted all approved treatments in Germany or were unfit for subsequent scheduled treatments, were offered terbinafine as off-label individual clinical intervention. Terbinafine was given either at the standard dose of 500 mg/day or at a reduced dose of 250 mg/day (in the case of mild liver insufficiency) over the course of two weeks (in the case of three patients) or six weeks (in the case of one patient). Patients were followed up

according to good clinical practice procedures with weekly physical exams and blood cheques, including differential blood count, electrolytes and transaminases. PSA was determined after two weeks and—in case of continued therapy—every other week. All patients provided written and informed consent for the off-label use of terbinafine as well as for the publication of their cases and data in anonymized form. Treatment was carried out in accordance with paragraph 37 of the latest amendment of the declaration of Helsinki according to good clinical practice guidelines. Retrospective data analysis was approved by the institutional review board of the University Hospital, Würzburg (No 20200318/01).

**Statistics and reproducibility**. All experiments were performed with at least three biologically independent replicates, unless otherwise stated. Data display and statistical analyses were performed using either R 3.10 (http://www.r-project.org) or Graphpad Prism 9 (Graphpad Software Inc.). R packages for data visualisation included ggplot2 (https://cran.r-project.org/web/packages/ggplot2/index.html), ggExtra (https://cran.r-project.org/web/packages/ggExtra/index.html), and stats (https://stat.ethz.ch/R-manual/R-devel/library/stats/html/stats-package.html). Pearson's chi-squared test was used to perform intergroup comparison. If two means of normally distributed data had to be compared, two-sided unpaired student's t-test was used. For comparison of paired, nonnormally distributed data, Wilcoxon signed-rank test was used. More than two group means were differentiated by analysis of variance (ANOVA) with post hoc testing (Tukey's test) if significant differences occurred. Significant associations were set as $p \leq 0.05$ (*), if not stated otherwise.

**Reporting summary**. Further information on research design is available in the Nature Research Reporting Summary linked to this article.

## Data availability

All data generated in this study are available in the Supplementary Information/Source Data file. Expression data were extracted from TCGA (PRAD primary, PanCancer Atlas and PRAD metastasis, SU2C Dream Team) using cBioPortal (http://cbioportal.org). Source data are provided with this paper.

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

## Acknowledgements

The authors thank Beatrice Dankworth, Katrin Borschert, Barbara Dexler, and Vera Schwartz for excellent technical assistance. The authors further thank Marianna Kruithof-de Julio and Eugenio Zoni for providing assistance with tissue microarrays. AS, MTS, MTR, and FR acknowledge funding by the German Research Foundation (DFG, FOR2314 and SCHU 2670/1). CK is grateful for funding through a fellowship by the Ferdinand Eisenberger Foundation of the German Association of Urology (KaC1/FE-17) and from the Interdisciplinary Centre for Clinical Research Würzburg (Z-4/133).

## Author contributions

C.K., B.K., and A.S. designed the study; C.K., P.S., M.T.S., H.M., T.F., and G.B. performed cell culture experiments and data analyses; W.S. and M.T.S. performed LC–MS analyses; F.R. helped with cloning and data analysis; J.L., M.Sa., K.J., and A.M. performed and analysed in vivo experiments; E.H. and M.T.R. performed pathological analyses; H.K., A.K.S., M.K., S.W., and M.E. provided essential material and expertise; M.P., M.Sp., and M.Eck provided patients samples and tissue microarrays; C.K. and A.S. wrote the paper with input from all authors.

## Competing interests

The authors declare no competing interests.
