## [Peer Review File · Nature Communications]

REVIEWER COMMENTS

Reviewer #1 (Remarks to the Author):

This interesting study investigated the upregulation of SQLE in high-grade PCa and uncovered a new mode of regulation by the microRNA miR-205, which is frequently downregulated in PCa. While previous reports have drawn indirect links between SQLE and PCa in patient cohorts, Kalogirou et al. provide a detailed mechanistic study of how the enzyme and its regulation by miR-205 contributes to PCa cell growth and AR transactivation. They show that miR-205 downregulation results in SQLE overexpression and increased cholesterol synthesis, providing a precursor to androgens that drive AR activity and cell proliferation. Importantly, SQLE inhibition induces apoptosis in PCa tumours in vivo, highlighting a potential new therapeutic intervention.

The experiments follow a logical sequence and the authors use an array of complimentary approaches to support their conclusions. Approaches include analysis of publicly available and in house prostate cancer cohorts and tissue samples, luciferase reporter assays, in cellulo metabolic labelling, and an orthotopic mouse model treated with terbinafine. While this in vivo study doesn't directly address involvement of miR-205, it does establish the utility of targeting SQLE in PCa, in line with previous publications in other cancer settings [ref 24,32].

A high level of detail is provided in the experimental methods, and the statistical analyses are appropriate. This well-written and well-presented paper therefore represents a substantial advance in the PCa and cholesterol metabolism fields and is likely to be of wide appeal to the readership of the journal. Nevertheless, the model proposed by the study would be better supported by more efficient inhibition of SQLE, as described below, and further clarity is needed in some aspects of the text.

Comments

1. A primary concern is the use of terbinafine in this study. The doses used to inhibit SQLE in cell culture are extremely high ($> 50 \mu\text{M}$), as expected given that this drug is an inhibitor of fungal SQLE and exhibits low potency against human SQLE [Padyana et al. 2019 Nat Comms. doi: 10.1038/s41467-018-07928-x]. However, high doses of terbinafine trigger the lysosomal degradation of SQLE in HCC cell lines [ref 32 in the text], which is distinct from its typical proteasomal degradation and implies the induction of autophagy. Terbinafine may therefore be having off-target effects on PCa cell growth, which could account for the modest effects of cholesterol rescue (e.g. Fig. S3H, Fig. 4F,G) or the drug's effect on the growth of AR-independent and castration-resistant cell lines (Fig. 3E,F). It would be useful to confirm the authors' in vitro findings using NB-598, an inhibitor of human SQLE that is highly potent (effective at $< 1 \mu\text{M}$) and less likely to have off-target effects.
2. Cholesterol regulates SQLE both transcriptionally and post-translationally (recently reviewed in [Chua et al. 2020 Prog Lipid Res. doi: 10.1016/j.plipres.2020.101033]), and so it is important to determine if the upregulation of SQLE when miR-205 is knocked out is independent of cholesterol treatment. i.e. Is SQLE still upregulated to the same extent with miR-205 knockout when cells are treated with the water-soluble cholesterol?
3. Would increasing the dose of water-soluble cholesterol help to improve the rescue of cell growth or AR transactivation following terbinafine treatment?
4. The authors state in their discussion that "transactivation of the AR following SQLE inhibition was restored by R1881, confirming that SQLE blockade interferes with androgen availability in PCa cells". However, there is no data to support this conclusion in the relevant figures.
5. The authors have used a variety of AR-dependent and AR-independent cell lines in their study, which lends strong support to their conclusions. However, the effect of miR-205 or terbinafine on AR transactivation in LNCaP cells is not shown. While it is understandable that the authors would include VCaP data for this experiment given that it forms the basis of their subsequent in vivo model, it would be useful to include supporting data for the widely used LNCaP cell line.

6. Could the authors clarify the meaning of “a conserved binding site for the PCa-related miR-205 in the 3′-UTR of SQLE”? Fig. 2A does not appear to contain data relating to the conservation of this sequence.

7. Regulation of SQLE transcripts by a microRNA (miR-133b) was previously reported in esophageal squamous cell carcinoma [Qin et al. 2017 *Acta Biochim. Biophys. Sin.* doi: 10.1093/abbs/gmw127], and this microRNA appears in the in silico analysis (Fig. 2A). In their discussion, could the authors comment on whether this is likely to be relevant in PCa?

8. The use of “rewiring” in the manuscript title is vague. Replacing this with “downregulation” would be more informative. Also consider replacing the term “therapeutic window” with the term “therapeutic vulnerability” or similar, since the former phrase already has an established meaning and the in vivo models did not test varying doses of terbinafine.

9. It is sometimes unclear whether the text is referring to the SQLE gene or SQLE protein, particularly in the second results section. Please confirm the correct usage of each term and consider italicising SQLE in figures that concern gene expression.

10. In Fig. 3C (PC-3 panel), lane 3 of the immunoblot is noticeably underloaded. Could the authors provide an updated immunoblot to clarify the extent of SQLE downregulation?

11. SQLE immunoblots are annotated with a 38 kDa molecular weight marker (e.g. Fig. 2I); however, the expected molecular weight of SQLE is ~64 kDa. Are you sure that 38 kDa is correct? If so, what is the explanation for this discrepancy?

12. Please correct small text errors, e.g. line 93 (“We confirm, that”), line 158 (“across the entire (tissue collection?) and along the line”), the y-axes of Fig. 4F and Fig. S4C (“LKL3”), and the label in Fig. 5A (“orthotpic”).

13. Details of the antibodies used for immunoblotting are absent (line 606).

Reviewer #2 (Remarks to the Author):

Prostate cancer (PCa) progression is known to involve deregulation of lipid (fatty acid, cholesterol, phospholipids, etc) biosynthesis and metabolism. Early PCa growth is driven by androgens, which are primarily synthesized from cholesterol as a substrate. Abundant evidence exists for miR-205 functioning as a tumor suppressive microRNA and for its downregulation accompanying PCa development and progression. Here, the authors demonstrate that miR-205 directly targets SQLE (squalene epoxidase; also called squalene monooxygenase), the second rate-limiting enzyme in cholesterol biosynthesis. SQLE is found to be overexpressed in advanced and high-risk PCa, which is involved in PCa cell proliferation. Although many previous studies have reported upregulation of SQLE mRNA and protein in PCa associated with disease progression, SQLE regulation directly by miR-205 is novel. Nevertheless, the manuscript mainly used long-term cultured PCa cells in biological studies, suffers many technical issues and confusing results and interpretations, and, overall, lacks significant conceptual advance. Although the demonstrated inhibitory/therapeutic effects of terbinafine, an anti-fungal medication, in PCa models is of potential clinical interest, a recent publication on its use in ~800 PCa patients did not indicate a conclusive benefit (Ji et al., *Int. J. Cancer*, 144:1888-1895, 2019).

Main points:

1. Figure 1/Fig. S1:

1a). Correlation plots shown in Figure 1c and Fig. S1a are unconvincing with very low (0.2 or less) correlation co-efficiency values.

1b). Correlation bar graphs and dot plots in Fig. S1b-c need to be re-presented with better labeling and legibility.

1c). The representative SQLE IHC images in the TMA are of poor quality.

1d). There is no description of SQLE level stratification in Methods or figure legends. Pls provide the cut-off level of SQLE (Figure 1a).

1e). How the score scale of 0- 12 is calculated (Figure 1D)?

2. Figure 2/ Fig. S2: Data in Figure 2e-f clearly demonstrated inverse correlation between miR-205 and SQLE mRNA levels in primary tumors vs. metastases. Nevertheless, overall this part of the studies has many technical concerns.

2a). The identification of miR-205 as a SQLE-targeting miRNA seems to be haphazard rather than systemic. How many other miRNAs might target the 3'-UTR of SQLE?

2b). In DOX-induced miR-205 overexpression in PCa cells (Figure 2c), the induction often exceeded tens of thousands of folds (of non-induced cells). At such high non-physiological levels, miR-205 will target many more genes than just SQLE. Indeed, similar induction of miR-205 only led to a maximum of 2-3 fold increase in SQLE. There was no systemic studies to demonstrate the target specificity of overexpressed miR-205, a pleiotropic microRNA that has numerous molecular targets including AR.

2c). Likewise, there are also concerns with the miR-205 KO. The Hsa-miR-205 is derived from the MIR205HG, a complicated genomic locus. Recent studies show that the MIR205HG itself functions as a lncRNA that is transcribed at much higher levels than miR-205 and plays important roles in, e.g., regulating prostate basal cell differentiation (Profumo V et al., Nat Commun., 10: 307, 2019) and pituitary hormone production (Du Q, et al., Dev. Cell 49:618-631, 2019). Authors need to demonstrate that their miR-205 KO gRNAs did not impact the MIR205HG integrity and transcription. Were the LNCaP and PC3 miR-205 KO bulk cells or selected clones? Proper characterizations of the KO cells need to be conducted to reveal the precise deleted genomic region and the disrupted sequences surrounding the gRNA cutting sites.

3. Figure 3/ Fig. S3: It's counter-intuitive that the phenotype of SQLE (the enzyme) knockdown can be 'rescued' by cell-permeable cholesterol (the substrate) - can authors offer an explanation? If 'cholesterol biosynthesis is rate-limiting for androgen production and viability in AR-dependent PCa cells.... (page 7)', how would authors reconcile with '.... SQLE has also been identified as a selective vulnerability in neuroendocrine tumors..... (page 6, bottom).....' which are largely AR-negative? Again DOX-induced miR-205 expression may likely be non-physiological unless authors demonstrate that induction of physiological levels of miR-205 also inhibits SQLE and cell proliferation. Significantly, did authors observe any phenotypes of SQLE knockdown in AR-negative PCa cell lines such as PC3 and DU145? IF no phenotype, then how would miR-205-KO in PC3 cells lead to increased SQLE, EMT targets (ZEB1, VIM) and migration (Figure 2h-i/ Fig. S2d-e)?

Frequently, SQLE was detected as two bands (Figure 3a, c, e; Fig. S3a, c, e) - are both bands SQLE? The two bands often show discordant changes upon DOX-induced knockdown or overexpression and quantitative data from several repeat experiments should be presented.

4. Figure 4. It was stated that SQLE is essential for AR transactivation in PCa cells. However, the downregulation of SQLE by miR-205 would not sufficiently support this statement because of reciprocal relationship between miR-205 and AR (signaling). Authors should also provide a set of data by using QSLE knockdown (or KO) cells.

5. Figure 5: Since Terbinafine treatment did not affect the serum levels of cholesterol in tumor-bearing animals, authored concluded that '..... terbinafine functioning by blocking tumor-intrinsic SQLE activity,.....' but there is no data buttressing this statement. Other issues:

5a). How many animals were used in each group?

5b). Line 311- "androgen-independent PCa orthografts". But authors utilized androgen-sensitive VCaP xenograft model. Also, in line 388 the author mentioned "validated orthotopic model of CRPC"; however, based on the schema of study design, they didn't perform castration.

5c). Figure 5E- the author should explain why there is no significant difference of Ki67, in contrast to what they observed in vitro.

4d). Regarding systemic toxicities in terbinafine-treated mice: Author should also provide data on parameters of kidney functions and immune/inflammatory cytokines. Also, the changes in animal body weight?

6. Other points:

5a). The study is generally descriptive and lacks in-depth mechanistic study that elucidates how miR-205/ SQLE axis impacts the AR signaling by regulating cholesterol.

5b). The title is vague and the use of "therapeutic window" seems improper.

Reviewer #3 (Remarks to the Author):

General comments:

The authors have presented an interesting report linking miR-205 with de novo tumoral androgen biosynthesis via SQLE and the activation of AR in prostate carcinogenesis. There is a general lack of clarity whether 'aggressive prostate cancer' in the title of the report refers to treatment resistance (namely castration resistance) and/or invasive cancer (local invasion, regional and distant metastasis). The impact of the report is diminished by the switching the focus from to another with the data flow.

The authors employed a number human prostate cancer cell models, it is sometimes unclear whether the effects shown in the manuscript are SQLE specific. Indeed, from Figure 3 onwards the authors mainly use indirect models modulating expression of miR-205 or with terbinafine treatment. For effects following terbinafine treatment, data on SQLE expression was lacking.

In the report, the miR205-SQLE-AR axis is emphasised as a source of intra-tumoral cholesterol to fuel cancer progression. Cholesterol uptake into the tumor is nonetheless a possibility as reported in the literature. A quick search revealed some relevant papers for the authors to consider: bioRxiv preprint doi: <https://doi.org/10.1101/2020.02.24.963454>; Carcinogenesis. 2017 Aug 1;38(8):806-811. doi: 10.1093/carcin/bgx058; PMID: 29540470 PMID: PMC5887544 DOI: 10.15252/emmm.201708347; J Biol Chem. 2020 Jun 12;295(24):8252-8261. doi: 10.1074/jbc.RA120.013694; Cancer Res. 2019 Jul 1;79(13):3320-3331. doi: 10.1158/0008-5472.CAN-18-2529; Mol Cancer Res. 2019 May;17(5):1166-1179. doi: 10.1158/1541-7786.MCR-18-1147. The authors may wish to discuss cholesterol uptake in the discussions. (Also refer to specific comment on Figure 4 below)

Specific comments:

TMA cohorts consisted of treatment naive tumours. Although considered to have high risk disease, almost half of them were low grade (Gleason 6) and presumably considered high risk because of high PSA (>20 ng/ml). If possible, it would be good to include analysis of clinical samples of castration resistant tumours. While comparison between high and low grade disease within individuals was significant, do tumours with higher grade as a group have higher levels of SQLE immunoreactivity when compared to tumours with lower grade disease? Did SQLE expression correlate to the presence or absence of nodal cancer.

The effects of miR205 modulation, terbinafine treatment and shSQLE are mainly correlative in Figure 2-3-4. How much of the miR205 effects are actually due to SQLE expression?, i.e. how much of terbinafine (or shSQLE) mediated effects are specific to SQLE. The paper would benefit from repeating some key experiments (EMT/migration - cholesterol tracing) using their miR205 KO with terbinafine (or shSQLE) to show that these effects are specific to SQLE.

What is the basal expression of SQLE in RWPE-1 or normal tissues compared to tumoral lesions?

Figure 2F – higher SQLE expression in metastasis is interesting. Do tumours with higher levels of SQLE also have higher levels of AR target gene expression?

In Figure 3b and 3d, what is the effect of cholesterol/DHT supplementation on WT cells? This is an important control that is missing to assess whether the supplementation rescues the phenotype or whether both curves are shifted (and the effect of the silencing remains the same). Do cholesterol or DHT treatment affect SQLE expression?

One main finding of this paper is that SQLE-driven cholesterol synthesis leads to AR activation in AR+ cells (and shSQLE effects can be rescued by DHT). However according to figure S3d, 3e (and suggested in figure 2h, i and S2d, e) SQLE depletion also affects PC3 cells phenotype in a

cholesterol-dependent manner. Obviously in that case, cholesterol is not used for steroid production for AR mediated functions. So what do AR-ve cells use cholesterol for? This should be at least discussed by the authors.

Terbinafine experiments were carried out using relatively high doses in vitro. As terbinafine is known to control other targets including cell cycle progression. It would be good to have additional data on selected known targets as well as its effects on androgen biosynthesis intermediaries (Figure 3F...)

In general, the western blots for SQLE depletion are not very convincing in figures 3c and S3a, c. Maybe some PCR validation might help.

Loss of SQLE affects de novo cholesterol biosynthesis, which accounts for 10-15% of the total cholesterol pool. This suggests that the main source for cholesterol is via external uptake (from the serum). Therefore it would be good to assess the effect of SQLE inhibition on cholesterol uptake.

Reduced cholesteryl ester synthesis (Figure 4B) is interesting. The extent of reduction is higher than that of total cholesterol synthesis, perhaps simply reflecting that the esters fraction being smaller. Is there evidence that the acyl CoA:cholesterol acyltransferase enzyme level is altered?

Given AR is a target of miR205 and that the promoter report assay data, it would be good to have data on AR expression following terbinafine treatment? Also to be confident that the effects is AR mediated, the same experiment in AR negative cells would be useful.

Similar to Figure 3, what is the effect of cholesterol supplementation on WT AR activity (KLK3 level) in figure 4?

Data from in vivo experiment suggests that even in androgen proficient mice, de novo synthesis of androgens is important and rate limiting for in vivo growth. Data so far in the report would implicate the importance of SQLE particularly in castration resistant disease. The in vivo experiment has not addressed this issue, and the in vivo experiment would have been more appropriate to be performed in castrated mice. It would also be useful to have IHC data on endpoint tumors to consider both the level of AR expression as well as whether AR present is active and has a nuclear localisation.

The serum cholesterol level is not altered by terbinafine treatment (Figure 5J). This is surprising – in other words, the authors are suggesting that the treatment has differential effects on tumour and the liver, and that in hormone proficient host mice, hormone dependent treatment naïve tumors are already utilising de novo androgen synthesis as a growth strategy. The authors should clarify this point. In addition, if that is the case, then why are there relatively few cases shown to involve this pathway in the clinical TCGA cohort? The title of the report focuses on aggressive disease, so it would be useful to have data on metastasis or treatment resistance in the in vivo assessment.

The authors suggest that terbinafine mainly affects AR signalling via SQLE status. Does terbinafine treatment simply mimic treatment with AR inhibitors or would it synergize with androgen receptor pathway inhibitors?

Minor comments:

Out of curiosity, why is there 2 bands for SQLE WB?

The use of "overexpression of shRNAs" in figure legends does not sound right

PCR graphs on figure 4f and S4c, S4d are labelled LKL3 instead of KLK3

In the text, S4C is cited after S4D

Would the acute co-treatment of terbinafine and AR inhibitors be beneficial in vitro?

Some of the experiments performed on C4-2 SQLE OE cells (migration, cholesterol labelling, AR activity or KLK3 expression) in supplementary would benefit from being included in the main figures.

Fig 1 showed evidence of alteration involving SQLE in clinical prostate cancer cohorts. Associations

with other components of the cholesterol pathway in Figure S1 are nice but some of the labels for Figure S1 are too small.
Space before the unit e.g. mM was missing in places throughout the report.
 μl and μL were used in different part of the report.

Answer to reviewers' comments:

We would like to thank the reviewers for their insightful evaluation of our manuscript and the helpful suggestions. We have extensively revised the manuscript by performing new experiments according to the reviewers' suggestions. A point-by-point response to the questions raised is provided below.

We would also like to draw the reviewers' attention to the supportive information provided in addition to the revised manuscript. We obtained data from four late-stage prostate cancer patients receiving a two-week course of terbinafine as individual intervention. All patients had exhausted standard-of-care treatments and showed exponential progression of the disease. Terbinafine, given either at the standard dose of 500 mg/day or a reduced dose of 250 mg/day, delayed chemical progression in three of the four patients, evidenced by a reduction in serum PSA levels, without greater adverse side-effects. We also provide a more detailed analysis on two of the four patients. We would like to include these encouraging data in the current manuscript, pending editorial approval. These data are currently provided as accompanying information.

Reviewer #1 (Remarks to the Author)

This interesting study investigated the upregulation of SQLE in high-grade PCa and uncovered a new mode of regulation by the microRNA miR-205, which is frequently downregulated in PCa. While previous reports have drawn indirect links between SQLE and PCa in patient cohorts, Kalogirou et al. provide a detailed mechanistic study of how the enzyme and its regulation by miR-205 contributes to PCa cell growth and AR transactivation. They show that miR-205 downregulation results in SQLE overexpression and increased cholesterol synthesis, providing a precursor to androgens that drive AR activity and cell proliferation. Importantly, SQLE inhibition induces apoptosis in PCa tumours in vivo, highlighting a potential new therapeutic intervention.

The experiments follow a logical sequence and the authors use an array of complimentary approaches to support their conclusions. Approaches include analysis of publicly available and in house prostate cancer cohorts and tissue samples, luciferase reporter assays, in cellulo metabolic labelling, and an orthotopic mouse model treated with terbinafine. While this in vivo study doesn't directly address involvement of miR-205, it does establish the utility of targeting SQLE in PCa, in line with previous publications in other cancer settings [ref 24,32].

A high level of detail is provided in the experimental methods, and the statistical analyses are appropriate. This well-written and well-presented paper therefore represents a substantial advance in the PCa and cholesterol metabolism fields and is likely to be of wide appeal to the readership of the journal. Nevertheless, the model proposed by the study would be better supported by more efficient inhibition of SQLE, as described below, and further clarity is needed in some aspects of the text.

We wish to thank the reviewer for their insightful evaluation of our study.

Comments

1. A primary concern is the use of terbinafine in this study. The doses used to inhibit SQLE in cell culture are extremely high ($> 50 \mu\text{M}$), as expected given that this drug is an inhibitor of fungal SQLE and exhibits low potency against human SQLE [Padyana et al. 2019 Nat Comms. doi:10.1038/s41467-018-07928-x]. However, high doses of terbinafine trigger the lysosomal degradation of SQLE in HCC cell lines [ref 32 in the text], which is distinct from its typical

proteasomal degradation and implies the induction of autophagy. Terbinafine may therefore be having off-target effects on PCa cell growth, which could account for the modest effects of cholesterol rescue (e.g. Fig. S3H, Fig. 4F,G) or the drug's effect on the growth of AR-independent and castration-resistant cell lines (Fig. 3E,F). It would be useful to confirm the authors' *in vitro* findings using NB-598, an inhibitor of human SQLE that is highly potent (effective at < 1 μ M) and less likely to have off-target effects.

We thank the reviewer for raising this important point and we have conducted a series of experiments to address this question. We first compared the effect of NB-598 and terbinafine on *de novo* cholesterol biosynthesis by performing 13 C-glucose labelling. These experiments clearly showed that both terbinafine and NB-598 efficiently block *de novo* cholesterol synthesis. However, NB-598, which has a much higher affinity for the catalytic centre of the SQLE enzyme¹, already completely abolished cholesterol synthesis at a concentration of 1 μ M (new Fig. 3e-g). This concentration was already effective in reducing cell viability in the LNCaP-Abl series of PCa cell lines (new Fig. 5g), which are highly sensitive to ablation of AR signalling². However, NB-598 was less effective in reducing viability of the other AR-dependent PCa cell lines (new Fig. 5b).

We therefore considered whether loss of cell viability in response to terbinafine could involve additional pathways, including the induction of autophagy, as suggested by previous studies in hepatocellular carcinoma (HCC)³. Interestingly, terbinafine, but not NB-598, resulted in a substantial accumulation of the autophagy regulator LC3 in LNCaP cells (new Fig. 5c). However, in contrast to the observations made in HCC³, this was not accompanied by a decrease in SQLE protein. Instead, we observed that both terbinafine and NB-598 increased SQLE protein expression (Reviewer Fig. 1).

Reviewer Figure 1: Terbinafine and NB-598 increase SQLE expression. C4-2 cells were treated with the indicated doses of terbinafine, abiraterone (AB) or NB-598 for 48 hours. Levels of SQLE protein were determined by immunoblotting. Vinculin is shown as loading control.

We also analysed tumour tissue from terbinafine-treated mice for evidence of autophagy by quantifying characteristic LC3 puncta. Interestingly, we observed that terbinafine caused a substantial increase in LC3 staining and the number of LC3 puncta per cell (new Fig. 7g and S7b). This suggests that induction of autophagy could contribute to the efficient inhibition of prostate tumour growth observed in our study. This interesting hypothesis is now also discussed in the manuscript.

2. Cholesterol regulates SQLE both transcriptionally and post-translationally (recently reviewed in [Chua et al. 2020 Prog Lipid Res. doi: 10.1016/j.plipres.2020.101033]), and so it is important to determine if the upregulation of SQLE when miR-205 is knocked out is independent of cholesterol treatment. i.e. Is SQLE still upregulated to the same extent with miR-205 knockout when cells are treated with the water-soluble cholesterol?

We thank the reviewer for raising this important point. We fully agree that the inhibition of SQLE protein stability by cholesterol represents an important regulatory feedback mechanism within the cholesterol biosynthesis axis⁴ and it is likely that the observed induction of SQLE protein following terbinafine or NB-598 treatment (Reviewer Figure 1) is caused by this mechanism. However, we would like to note that the experiments demonstrating the regulation of SQLE expression by miR-205 (Fig. 2i and j) were performed in cholesterol-containing

medium (i.e. in 10% FCS). This suggests that loss of miR-205 can induce SQLE expression, both on mRNA and protein level, under conditions where exogenous cholesterol is readily available. We anticipate that SQLE induction following miR-205 k/o would be even stronger in cholesterol-deplete conditions. However, as this mechanism has already been elucidated, we decided to not follow this further but now discuss the finding of the Gill et al. paper in the text.

3. Would increasing the dose of water-soluble cholesterol help to improve the rescue of cell growth or AR transactivation following terbinafine treatment?

We have tried to increase the concentration of water-soluble cholesterol used in our assays but this has a negative effect on cell viability, potentially caused by the methyl- β -cyclodextrin used to complex free cholesterol at a ratio of 25:1 (w/w) in this formulation.

4. The authors state in their discussion that “transactivation of the AR following SQLE inhibition was restored by R1881, confirming that SQLE blockade interferes with androgen availability in PCa cells”. However, there is no data to support this conclusion in the relevant figures.

The reviewer is correct and we apologise for this mistake. This statement has been corrected to: “Transactivation of the AR and expression of AR target genes following SQLE inhibition was restored by addition of cell-permeable cholesterol, confirming that SQLE blockade interferes with AR activity by reducing cholesterol availability in PCa cells.”

5. The authors have used a variety of AR-dependent and AR-independent cell lines in their study, which lends strong support to their conclusions. However, the effect of miR-205 or terbinafine on AR transactivation in LNCaP cells is not shown. While it is understandable that the authors would include VCaP data for this experiment given that it forms the basis of their subsequent *in vivo* model, it would be useful to include supporting data for the widely used LNCaP cell line.

Our focus was indeed to confirm the robustness of our findings on the regulation of AR transactivation by miR-205 and SQLE by using a cell line suitable for orthotopic implantation experiments. VCaP cells are widely used to investigate the role of AR in PCa cells. While both LNCaP and VCaP are derived from metastatic disease, VCaP show higher levels of AR expression with predominantly nuclear localisation⁵ making this a highly relevant cell line for our study. Nevertheless, we now include new data showing that terbinafine and NB-598 also reduce the expression of the AR target gene *KLK3* in LNCaP cells (new Fig. S6c).

6. Could the authors clarify the meaning of “a conserved binding site for the PCa-related miR-205 in the 3'-UTR of SQLE”? Fig. 2A does not appear to contain data relating to the conservation of this sequence.

We thank the reviewer to raise this important point. Indeed, a high degree of conservation of mature miR-205 sequence has been documented across multiple species of eutheria⁶. Moreover, we have investigated conservation of the miRNA seed region in SQLE across mouse, human and chimpanzee and find good conservation. Figures 2a and b have been revised to reflect this better. We also revised the relevant text section to: “PCa progression is associated with alterations in micro-RNA expression⁷. We therefore used *in silico* alignment tools to identify a conserved sequence in the 3'-UTR of SQLE that matches the seed region for the PCa-related miR-205 (Fig. 2a and b).”

7. Regulation of SQLE transcripts by a microRNA (miR-133b) was previously reported in esophageal squamous cell carcinoma [Qin et al. 2017 Acta Biochim. Biophys. Sin. doi: 10.1093/abbs/gmw127], and this microRNA appears in the *in silico* analysis (Fig. 2A). In their discussion, could the authors comment on whether this is likely to be relevant in PCa?

This is an interesting aspect. The data presented by Qin et al. demonstrate a role for SQLE in promoting cell migration in esophageal squamous cell carcinoma and thus provide independent support for the importance of SQLE in cancer⁸. However, as our analysis of the expression of micro-RNAs predicted to regulate SQLE showed a stronger downregulation for miR-205 (Log2FC -3.39) compared to miR-133b (log2FC -1.66), we decided to follow up on this. We do not wish to suggest that miR-205 is the only micro-RNA contributing to SQLE expression in PCa.

8. The use of “rewiring” in the manuscript title is vague. Replacing this with “downregulation” would be more informative. Also consider replacing the term “therapeutic window” with the term “therapeutic vulnerability” or similar, since the former phrase already has an established meaning and the *in vivo* models did not test varying doses of terbinafine.

We thank the reviewer for this excellent suggestion and have amended the title accordingly.

9. It is sometimes unclear whether the text is referring to the SQLE gene or SQLE protein, particularly in the second results section. Please confirm the correct usage of each term and consider italicising SQLE in figures that concern gene expression.

The text has been revised accordingly.

10. In Fig. 3C (PC-3 panel), lane 3 of the immunoblot is noticeably underloaded. Could the authors provide an updated immunoblot to clarify the extent of SQLE downregulation?

The experiment has been repeated and the figure revised. Also see response to next point.

11. SQLE immunoblots are annotated with a 38 kDa molecular weight marker (e.g. Fig. 2I); however, the expected molecular weight of SQLE is ~64 kDa. Are you sure that 38 kDa is correct? If so, what is the explanation for this discrepancy?

We thank the reviewer for noticing this serious mistake. We used a commercially available SQLE antibody (Origene, cat#: TA346342). The data sheet provided by the company stated that this antibody detects SQLE protein at a size of 38 kDa. We now realise that this was a mislabelling by the provider and we have repeated all immunoblot analyses for SQLE protein using a different primary antibody (Proteintech, cat#: 12544-1-AP). Careful size alignment was applied to demonstrate specificity of the band at 64 kDa with an unspecific band with higher mobility. Respective figures were revised and labelled accordingly.

12. Please correct small text errors, e.g. line 93 (“We confirm, that”), line 158 (“across the entire (tissue collection?) and along the line”), the y-axes of Fig. 4F and Fig. S4C (“LKL3”), and the label in Fig. 5A (“orthotpic”).

We thank the reviewer for pointing out those mistakes. They have been corrected.

13. Details of the antibodies used for immunoblotting are absent (line 606).

We apologise for this oversight. This information has been added.

Reviewer #2 (Remarks to the Author):

Prostate cancer (PCa) progression is known to involve deregulation of lipid (fatty acid, cholesterol, phospholipids, etc) biosynthesis and metabolism. Early PCa growth is driven by androgens, which are primarily synthesized from cholesterol as a substrate. Abundant evidence exists for miR-205 functioning as a tumor suppressive microRNA and for its

downregulation accompanying PCa development and progression. Here, the authors demonstrate that miR-205 directly targets SQLE (squalene epoxidase; also called squalene monooxygenase), the second rate-limiting enzyme in cholesterol biosynthesis. SQLE is found to be overexpressed in advanced and high-risk PCa, which is involved in PCa cell proliferation. Although many previous studies have reported upregulation of SQLE mRNA and protein in PCa associated with disease progression, SQLE regulation directly by miR-205 is novel.

Nevertheless, the manuscript mainly used long-term cultured PCa cells in biological studies, suffers many technical issues and confusing results and interpretations, and, overall, lacks significant conceptual advance. Although the demonstrated inhibitory/therapeutic effects of terbinafine, an anti-fungal medication, in PCa models is of potential clinical interest, a recent publication on its use in ~800 PCa patients did not indicate a conclusive benefit (Ji et al., *Int. J. Cancer*, 144:1888-1895, 2019).

We thank the reviewer for their expert opinion. While we acknowledge the limitations of our study, we respectfully disagree with their interpretation of the data shown in the study by Ji et al.⁹. This study compared mortality in patients with prostate cancer that received either oral or topical treatment with terbinafine after being diagnosed with the disease. The authors found that oral treatment with terbinafine in fact decreased the risk of death from prostate cancer (risk reduction of 47% for disease specific death and 36% overall death). In contrast, no benefit could be demonstrated for patients receiving topical application of the drug. We are encouraged by the authors conclusion that terbinafine should be further explored as a potential low-cost alternative to targeted treatment. We would also like to point out that we now have data from four prostate cancer patients receiving terbinafine treatment as individual treatment (see accompanying information). These results are indicative of an effect of terbinafine treatment on biochemical progression in advanced PCa.

Main points:

1. Figure 1/Fig. S1:

1a). Correlation plots shown in Figure 1c and Fig. S1a are unconvincing with very low (0.2 or less) correlation co-efficiency values.

We agree that these correlation plots do not contribute much novel information and we have removed them from our manuscript.

1b). Correlation bar graphs and dot plots in Fig. S1b-c need to be re-presented with better labeling and legibility.

We have decided to simplify this and only show representative data for *HMGCS1* and *FDFT1* with clearer labelling (new Fig. 1b).

1c). The representative SQLE IHC images in the TMA are of poor quality.

The images have been replaced with higher quality micrographs.

1d). There is no description of SQLE level stratification in Methods or figure legends. Pls provide the cut-off level of SQLE (Figure 1a).

We thank the reviewer for this remark. The *SQLE* expression dichotomisation (cut-off at 1024.544 RPKM) was determined by receiver-operating characteristics (ROC) using the R package pROC and the "closest-to-top-left" algorithm. This information has been added to the manuscript.

1e). How the score scale of 0-12 is calculated (Figure 1D)?

We have now added the following text to the methods section to describe our SQLE scoring system. "IHC of SQLE in the used tissue microarray samples was independently scored by several researchers (M.R., C.K. and/or A.S.) blinded to clinicopathological and outcome data. Potentially discordant scores were resolved by joint review. The percentage of cell immunoreactivity (0% to 100%, divided into 4 groups: 0-25%; 25-50%; 50-75%; 75-100%) and the staining intensity (score 0 to 3, with 0 indicating negative; 1, weak; 2, moderate; and 3, strong) were recorded for each core. Histoscores for SQLE expression (0-12) were calculated as the product of the percentage immunoreactivity in the staining intensity, resulting in an SQLE expression score between 0 and 12."

2. Figure 2/ Fig. S2: Data in Figure 2e-f clearly demonstrated inverse correlation between miR-205 and SQLE mRNA levels in primary tumors vs. metastases. Nevertheless, overall this part of the studies has many technical concerns.

2a). The identification of miR-205 as a SQLE-targeting miRNA seems to be haphazard rather than systemic. How many other miRNAs might target the 3'-UTR of SQLE?

We now provide additional details regarding the identification of miR-205 and its alignment with the SQLE mRNA (new Fig. 2a and b).

2b). In DOX-induced miR-205 overexpression in PCa cells (Figure 2c), the induction often exceeded tens of thousands of folds (of non-induced cells). At such high non-physiological levels, miR-205 will target many more genes than just SQLE. Indeed, similar induction of miR-205 only led to a maximum of 2-3 fold increase in SQLE. There was no systemic studies to demonstrate the target specificity of overexpressed miR-205, a pleiotropic microRNA that has numerous molecular targets including AR.

We apologise for this lack of clarity. The difference in expression obtained by the overexpression construct is in the same range as that observed between benign and metastatic PCa tissue (Fig. 2f, $\Delta\Delta Ct$ of 10 equates to a fold change of 1024). We therefore changed the display of the graph in Figure 2d to better represent this similarity. We do not wish to conclude that SQLE is the only target of miR-205 in PCa cells. Indeed, we already show evidence that deletion of miR-205 increased expression of ZEB1, which has previously been identified as a miR-205 target¹⁰.

2c). Likewise, there are also concerns with the miR-205 KO. The Hsa-miR-205 is derived from the MIR205HG, a complicated genomic locus. Recent studies show that the MIR205HG itself functions as a lncRNA that is transcribed at much higher levels than miR-205 and plays important roles in, e.g., regulating prostate basal cell differentiation (Profumo V et al., Nat Commun., 10: 307, 2019) and pituitary hormone production (Du Q, et al., Dev. Cell 49:618-631, 2019). Authors need to demonstrate that their miR-205 KO gRNAs did not impact the MIR205HG integrity and transcription. Were the LNCaP and PC3 miR-205 KO bulk cells or selected clones? Proper characterizations of the KO cells need to be conducted to reveal the precise deleted genomic region and the disrupted sequences surrounding the gRNA cutting sites.

We thank the reviewer for raising this important point. The KO strategy used here is based on the co-expression of two gRNAs targeting the same locus. This strategy leads to highly efficient genomic deletion and allows the production of cell pools with a large proportion of knockout cells. While the polyclonal nature prevents precise characterisation of the genomic deletion, efficiency is routinely confirmed using the T7 endonuclease kit (Biolabs, M0302). In addition, experiments were performed using cell pools generated using two different gRNAs (see Fig. 2h) with equivalent results.

We now provide data showing that the expression of the *MIR205HG* lncRNA is not altered by our knock out strategy in PC-3 cells using TaqMan analysis (Fig. S2b). Levels of *MIR205HG* lncRNA were below detection levels in LNCaP cells (data not shown).

3. Figure 3/ Fig. S3: It's counter-intuitive that the phenotype of SQLE (the enzyme) knockdown can be 'rescued' by cell-permeable cholesterol (the substrate) - can authors offer an explanation? If 'cholesterol biosynthesis is rate-limiting for androgen production and viability in AR-dependent PCa cells.... (page 7)', how would authors reconcile with '.... SQLE has also been identified as a selective vulnerability in neuroendocrine tumors..... (page 6, bottom).....' which are largely AR-negative?

We apologise for the potential lack of clarity that led to this misunderstanding. The SQLE enzyme converts squalene, a 30-carbon triterpene produced by the mevalonate pathway, into (S)-2,3-epoxysqualene, which is then further metabolised to generate cholesterol. Providing water-soluble cholesterol thus supplies the product of the metabolic pathway that SQLE is involved in and provides the substrate for androgen synthesis.

The study by Mahoney et al. showed that inhibition of SQLE resulted in the toxic accumulation of squalene, the substrate of the SQLE reaction ¹. To investigate whether squalene accumulation is responsible for the toxicity observed in PCa cells, we have treated C4-2 and LNCaP cells with YM53601, an inhibitor of FDFT1, the enzyme converting farnesyl diphosphate into squalene. While Mahoney et al. demonstrate that inhibition of squalene synthesis prevents toxicity caused by SQLE inhibition, we found that YM53601 also reduced cell viability (Fig. 5e). This crucial difference indicates that the response to SQLE inhibition is distinct in different cancer entities. We believe that our manuscript provides substantial evidence indicating that SQLE modulation targets androgen signalling in PCa. This is evidenced by i) the higher sensitivity of AR dependent PCa cell lines towards SQLE inhibition (Fig. 5a and b), ii) the effect of SQLE inhibition or silencing on AR transactivation and target gene expression (Fig. 6e-h) and iii) reduced plasma PSA and AR nuclear localisation in our orthotopic mouse model (Fig. 7b and i).

Again DOX-induced miR-205 expression may likely be non-physiological unless authors demonstrate that induction of physiological levels of miR-205 also inhibits SQLE and cell proliferation.

See our response to point 2b. We believe that the levels of induction of miR-205 achieved by our expression strategy lies within the physiological range observed in PCa tissues.

Significantly, did authors observe any phenotypes of SQLE knockdown in AR-negative PCa cell lines such as PC3 and DU145? IF no phenotype, then how would miR-205-KO in PC3 cells lead to increased SQLE, EMT targets (ZEB1, VIM) and migration (Figure 2h-i/ Fig. S2d-e)?

We apologise for the lack of clarity that could have caused this misunderstanding. We used miR-205 KO in PC-3 and LNCaP cells to demonstrate that SQLE is indeed a target gene of miR-205. As ZEB1 has previously been identified as miR-205 target ¹⁰, it served as a positive control for our experiments. We also confirmed regulation of EMT targets and increased cell migration (Fig. S2e and f) as a control for the efficiency miR-205 deletion rather than to determine any connection between SQLE and modulation of EMT. Interestingly, a recent report has linked SQLE to EMT in esophageal squamous cancer ⁸. However, our study only focused on the role of SQLE in supporting cell viability and tumour growth in PCa. While we cannot exclude that SQLE also plays a role in EMT in PCa, we believe that this lies outside of the scope of our current study.

Frequently, SQLE was detected as two bands (Figure 3a, c, e; Fig. S3a, c, e) – are both bands SQLE? The two bands often show discordant changes upon DOX-induced knockdown or overexpression and quantitative data from several repeat experiments should be presented.

We now have repeated all immunoblot analyses using a different SQLE antibody. This antibody detects a single band at 68 kDa that is specific for the SQLE protein. Furthermore, an unspecific band with higher mobility is also detected. This is now clearly marked in all figures. We also provide new data confirming efficient SQLE silencing by qPCR for all experiments (Fig. S4a).

4. Figure 4. It was stated that SQLE is essential for AR transactivation in PCa cells. However, the downregulation of SQLE by miR-205 would not sufficiently support this statement because of reciprocal relationship between miR-205 and AR (signaling). Authors should also provide a set of data by using SQLE knockdown (or KO) cells.

We agree that this point needed further clarification. We now provide additional data showing the SQLE silencing reduces the expression of the AR-luc reporter (new Fig. 6c) and the AR target gene *KLK3* (new Fig. 6e) in C4-2 cells. We also show reduced expression of the AR target *KLK3* following SQLE inhibition by terbinafine or TB-598 (new Fig. 6g). We believe that these results support our conclusion.

5. Figure 5: Since Terbinafine treatment did not affect the serum levels of cholesterol in tumor-bearing animals, authored concluded that ‘..... terbinafine functioning by blocking tumor-intrinsic SQLE activity,.....’ but there is no data buttressing this statement.

We have removed this statement from our manuscript.

Other issues:

5a). How many animals were used in each group?

We used a total of 16 mice that were randomised 1:1 into the treatment and control groups, as stated in the methods. We now also include this information in the figure legend.

5b). Line 311- “androgen-independent PCa orthografts”. But authors utilized androgen-sensitive VCaP xenograft model. Also, in line 388 the author mentioned “validated orthotopic model of CRPC”; however, based on the schema of study design, they didn’t perform castration.

We agree that these statements are misleading and need further clarification. VCaP cells were isolated from a metastatic lesion from a patient with castration resistant PCa (CRPC), thus representing a model of this stage of the disease. VCaP cells produce high levels of PSA and express wild type AR as well as the AR-V7 splice variant⁵. While VCaP cells exhibit androgen-independent growth, they still respond to androgens by altering expression of multiple growth-relevant AR target genes¹¹. To avoid any misunderstanding, we rephrased all sections of the manuscript referring to this cell line.

5c). Figure 5E- the author should explain why there is no significant difference of Ki67, in contrast to what they observed in vitro.

We did not want to give the impression that terbinafine treatment results in cell cycle arrest *in vitro*. We therefore investigated induction of apoptosis using Annexin V staining following terbinafine treatment in PCa cells. This analysis clearly showed that terbinafine induces apoptosis, evidenced by Annexin V staining and PARP cleavage, in the AR

dependent/sensitive PCa cell lines (LNCaP, C4-2 and VCaP) but not in AR-independent DU-145 and PC-3 cells (new Fig. S5a and b).

5d). Regarding systemic toxicities in terbinafine-treated mice: Author should also provide data on parameters of kidney functions and immune/inflammatory cytokines. Also, the changes in animal body weight?

According to FDA guideline, one of the most serious potential side-effects associated with the use of terbinafine (Lamisil®) is liver damage. We therefore took care to monitor for liver damage, by determining serum AST levels (Fig. S7e) and morphological integrity of liver tissue (Fig. S7f). We now also provide images of H&E-stained kidney tissue from control and terbinafine treated mice (new Fig. S7g). We also did not observe any reduction in body weight >20%, as defined by ethical guidelines. As terbinafine has an established safety profile, we believe that this is sufficient to exclude major adverse side effects.

6. Other points:

6a). The study is generally descriptive and lacks in-depth mechanistic study that elucidates how miR-205/ SQLE axis impacts the AR signaling by regulating cholesterol.

We respectfully disagree with this assessment. While the exact mechanism by which SQLE controls AR signalling requires further investigation, our study provides important insight into this regulatory axis. This study shows for the first time that SQLE is a target of micro-RNA associated with aggressive disease in PCa. Moreover, we show that SQLE modulation, either genetically or through chemical inhibition with two distinct compounds, abolishes PCa cell growth *in vitro* and in an orthotopic mouse model *in vivo*. We also provide evidence for modulation of AR transactivation downstream of the miR-205/SQLE axis. To our knowledge, this is the first study describing this connection and we believe that it may encourage further evaluation of terbinafine as an anti-cancer therapy.

5b). The title is vague and the use of “therapeutic window” seems improper.

The title has been revised.

Reviewer #3 (Remarks to the Author):

General comments:

The authors have presented an interesting report linking miR-205 with de novo tumoral androgen biosynthesis via SQLE and the activation of AR in prostate carcinogenesis. There is a general lack of clarity whether ‘aggressive prostate cancer’ in the title of the report refers to treatment resistance (namely castration resistance) and/or invasive cancer (local invasion, regional and distant metastasis). The impact of the report is diminished by the switching the focus from to another with the data flow.

The authors employed a number human prostate cancer cell models, it is sometimes unclear whether the effects shown in the manuscript are SQLE specific. Indeed, from Figure 3 onwards the authors mainly use indirect models modulating expression of miR-205 or with terbinafine treatment. For effects following terbinafine treatment, data on SQLE expression was lacking.

In the report, the miR205-SQLE-AR axis is emphasised as a source of intra-tumoral cholesterol to fuel cancer progression. Cholesterol uptake into the tumor is nonetheless a possibility as reported in the literature. A quick search revealed some relevant papers for the authors to consider:

bioRxiv preprint doi: <https://doi.org/10.1101/2020.02.24.963454>;
Carcinogenesis. 2017 Aug 1;38(8):806-811. doi: 10.1093/carcin/bgx058;
PMID: 29540470 PMCID: PMC5887544 DOI: 10.15252/emmm.201708347;
J Biol Chem. 2020 Jun 12;295(24):8252-8261. doi: 10.1074/jbc.RA120.013694;
Cancer Res. 2019 Jul 1;79(13):3320-3331. doi: 10.1158/0008-5472.
CAN-18-2529; Mol Cancer Res. 2019 May;17(5):1166-1179. doi: 10.1158/1541-7786.MCR-18-1147.

The authors may wish to discuss cholesterol uptake in the discussions. (Also refer to specific comment on Figure 4 below)

We thank the reviewer for raising this point. However, we believe that the additional data provided in the revised version improves clarity and supports our conclusion that the miR-205/SQLE axis is important for cell and tumour growth in PCa. Indeed, there is substantial evidence that cholesterol uptake can be a source for steroid hormone synthesis in PCa cells. However, due to the intricate feedback regulation of the cholesterol biosynthesis pathway through SREBP2, inhibition of cholesterol uptake, for example by silencing of SR-B1, leads to increased expression of mevalonate pathway genes and compensation by *de novo* synthesis¹². It is therefore likely that both *de novo* synthesis and uptake of cholesterol needs to be blocked to achieve efficient inhibition of androgen synthesis on PCa cells. We have addressed cholesterol uptake using stable isotope labelled cholesterol (discussed in detail below). Based on our new data and as suggested by the reviewer, we now discuss cholesterol uptake in the manuscript.

Specific comments:

1) TMA cohorts consisted of treatment naïve tumours. Although considered to have high risk disease, almost half of them were low grade (Gleason 6) and presumably considered high risk because of high PSA (>20 ng/ml). If possible, it would be good to include analysis of clinical samples of castration resistant tumours. While comparison between high and low grade disease within individuals was significant, do tumours with higher grade as a group have higher levels of SQLE immunoreactivity when compared to tumours with lower grade disease? Did SQLE expression correlate to the presence or absence of nodal cancer.

We thank the reviewer for raising this important point. We analysed the data from our high-risk PCa TMA also for correlation of SQLE expression with nodal status. This analysis did not yield a significant association (new Fig. S1d), most likely due to the stratified nature of the cohort. We therefore also included data from a second TMA comprising 453 unselected PCa patients. In this cohort, a significant association between SQLE staining and Gleason score, tumour stage and nodal status was observed (new Fig. 1d). Moreover, we found a significant correlation between SQLE and pre-operative PSA (new Fig. 1e). With this new data, we believe that we can conclude that enhanced SQLE expression is associated with high-risk disease in PCa.

2) The effects of miR205 modulation, terbinafine treatment and shSQLE are mainly correlative in Figure 2-3-4. How much of the miR205 effects are actually due to SQLE expression?, i.e. how much of terbinafine (or shSQLE) mediated effects are specific to SQLE. The paper would benefit from repeating some key experiments (EMT/migration - cholesterol tracing) using their miR205 KO with terbinafine (or shSQLE) to show that these effects are specific to SQLE.

We now provide additional data showing that depletion of SQLE using two specific shRNA sequences also reduces the activity of the AR-responsive reporter and lowers expression of the *KLK3* mRNA (new Fig. 6c and e). Moreover, we also observed that SQLE inhibition with terbinafine or NB-598 dramatically reduced expression of AR target gene *KLK3* (new Fig. 6g). Importantly, the effect of SQLE inhibition by terbinafine treatment on androgen signalling could also be observed in the orthotopic mouse model, as we observed a reduction in AR nuclear

localisation (new Fig. 7i). Together with our previous data showing a clear effect of SQLE overexpression on AR transactivation, our data clearly show that SQLE modulates cholesterol synthesis and androgen signalling in PCa.

3) What is the basal expression of SQLE in RWPE-1 or normal tissues compared to tumoral lesions?

We have already provided data on SQLE expression in RWPE non-transformed prostate epithelial cells (Fig. 2e), where the cells served as normalisation control. This analysis showed that basal expression of SQLE in RWPE cells is lower compared to the PCa cell lines.

4) Figure 2F – higher SQLE expression in metastasis is interesting. Do tumours with higher levels of SQLE also have higher levels of AR target gene expression?

This is an excellent suggestion. We now also provide results showing the expression of KLK3 (PSA) in the same samples (revised Fig. 2g). This clearly shows that PSA is also higher expressed in metastasis compared to paired primary tumour samples.

5) In Figure 3b and 3d, what is the effect of cholesterol/DHT supplementation on WT cells? This is an important control that is missing to assess whether the supplementation rescues the phenotype or whether both curves are shifted (and the effect of the silencing remains the same). Do cholesterol or DHT treatment affect SQLE expression?

We agree that this is an important control. The solvent controls in Fig. 5e already show that the same concentrations of cell-permeable cholesterol and R1881 do not alter viability of PCa cells in the absence of SQLE inhibition. Due to the well-established negative feedback loop controlling expression of cholesterol biosynthesis genes through regulated intramembrane processing of SREBP2¹³ and the direct regulation of SQLE protein stability⁴, it is highly likely that cholesterol addition will reduce SQLE expression. In contrast, DHT treatment could increase SQLE expression as the SQLE gene has been shown to contain an AR binding site¹⁴. Nevertheless, R1881 also restored cell viability after abiraterone treatment (Fig. 5e), making it unlikely that the effect of androgen is only caused by induction of SQLE expression. Further analysis of the regulation of SQLE expression by AR is outside the scope of the current study.

6) One main finding of this paper is that SQLE-driven cholesterol synthesis leads to AR activation in AR+ cells (and shSQLE effects can be rescued by DHT). However according to figure S3d, 3e (and suggested in figure 2h, i and S2d, e) SQLE depletion also affects PC3 cells phenotype in a cholesterol-dependent manner. Obviously in that case, cholesterol is not used for steroid production for AR mediated functions. So what do AR-ve cells use cholesterol for? This should be at least discussed by the authors.

We agree with the reviewer that this is an important point that needs to be discussed. As cholesterol is also an essential component of cellular membranes, we assume that AR independent cells require cholesterol synthesis (or uptake) to fulfil their requirements for this molecule to sustain proliferation. We have included this notion in the manuscript.

7) Terbinafine experiments were carried out using relatively high doses in vitro. As terbinafine is known to control other targets including cell cycle progression. It would be good to have additional data on selected known targets as well as its effects on androgen biosynthesis intermediaries (Figure 3F...)

We carefully evaluated the effect of terbinafine on cholesterol synthesis and found a good correlation between the effect of terbinafine on cholesterol synthesis and its effect on cell viability at the concentrations used (new Fig. 3g and 5a). Moreover, the reduction in cell

viability caused by 50 μ M of terbinafine could be ameliorated by cell-permeable cholesterol addition, showing the specificity of the inhibitor (Fig. 5e). We also include additional data on a second compound, NB-598, that inhibits SQLE activity at lower concentrations. This compound was much more effective in blocking cholesterol synthesis (new Fig. 3e and g) and also reduced viability in androgen dependent/sensitive PCa cells (new Fig. 5b) and resistant derivatives of the LNCaP cell line (new Fig. 5g).

Nevertheless, based on a previous study linking terbinafine to the induction of autophagy in liver cancer cells³, we have also investigated whether this drug might affect autophagy in PCa cells. We found that terbinafine, but not NB-598, increased the expression of the autophagy related protein LC3 in the androgen-sensitive LNCaP cells (new Fig. 5c). Interestingly, LNCaP cells showed the highest sensitivity towards terbinafine treatment (Fig. 5a), suggesting that the drug could have additional effects in these cells.

We also attempted to detect intermediaries of the androgen synthesis pathway using a stable isotope labelling strategy. However, this proved to be technically challenging due to the very low abundance of these compounds and the large number of carbon atoms (19 in the case of testosterone), resulting in multiple isotopologue peaks which further lowers the detection limit. We currently lack the analytical capability to explore this further.

8) In general, the western blots for SQLE depletion are not very convincing in figures 3c and S3a, c. Maybe some PCR validation might help.

We have confirmed the efficiency of SQLE silencing by qPCR (new Fig. S4a).

9) Loss of SQLE affects *de novo* cholesterol biosynthesis, which accounts for 10-15% of the total cholesterol pool. This suggests that the main source for cholesterol is via external uptake (from the serum). Therefore, it would be good to assess the effect of SQLE inhibition on cholesterol uptake.

To address this question, we have evaluated cholesterol uptake using stable isotope labelled cholesterol. In contrast to the water-soluble cholesterol used in our rescue experiments, cholesterol requires cholesterol transporters to enter cells. We treated C4-2 and LNCaP cells with cholesterol-2,3,4-¹³C₃ (cholesterol-¹³C₃) after induction of miR-205 or treatment with terbinafine, and determined the amount of total and cholesterol-¹³C₃ in the cells. This analysis showed that inhibition of SQLE resulted in a reduction of both the labelled cholesterol (resulting from uptake) and the unlabelled cholesterol (resulting from *de novo* synthesis). This result shows that cholesterol uptake cannot compensate for the reduction in *de novo* synthesis caused by SQLE inhibition (new Fig. 3b and S3c).

10) Reduced cholesteryl ester synthesis (Figure 4B) is interesting. The extent of reduction is higher than that of total cholesterol synthesis, perhaps simply reflecting that the esters fraction being smaller. Is there evidence that the acyl CoA:cholesterol acyltransferase enzyme level is altered?

This is an interesting point that we also noticed. As suggested by the reviewer, we have evaluated the expression of *SOAT1* in terbinafine treated cells by qRT-PCR. Indeed, terbinafine resulted in a substantial downregulation of *SOAT1* mRNA (new Fig. 3d), potentially explaining the strong effect on cholesterol esters. This is interesting as specific inhibitors of *SOAT1* are successfully applied in the treatment of adrenocortical carcinoma¹⁵. We are currently investigating whether *SOAT* inhibitors might synergise with SQLE inhibition in blocking androgen signalling in PCa.

11) Given AR is a target of miR205 and that the promoter report assay data, it would be good to have data on AR expression following terbinafine treatment? Also to be confident that the effects is AR mediated, the same experiment in AR negative cells would be useful.

We agree with the reviewer that a direct effect of miR-205 could also contribute to the reduction in AR reporter activity observed in our experiments. However, we did not find any evidence for reduced expression of *AR* mRNA following induction of miR-205 in LNCaP cells (new Fig. S6b). We now also include additional data showing that silencing of *SQLE* also reduces AR-reporter activity and lowers expression of *KLK3* mRNA (new Fig. 6b and e). Moreover, we found that *SQLE* inhibition by terbinafine or NB-598 reduces *KLK3* expression (new Fig. 6g and S6c). Together, these new data further support our conclusion that modulation of *SQLE* activity affects androgen signalling in PCa cells.

12) Similar to Figure 3, what is the effect of cholesterol supplementation on WT AR activity (*KLK3* level) in figure 4?

We have added this control in the new figure S6d.

13) Data from in vivo experiment suggests that even in androgen proficient mice, de novo synthesis of androgens is important and rate limiting for in vivo growth. Data so far in the report would implicate the importance of *SQLE* particularly in castration resistant disease. The in vivo experiment has not addressed this issue, and the in vivo experiment would have been more appropriate to be performed in castrated mice. It would also be useful to have IHC data on endpoint tumors to consider both the level of AR expression as well as whether AR present is active and has a nuclear localisation.

We thank the reviewer for this excellent suggestion. Careful evaluation of histological sections of tumour tissue from DMSO or terbinafine treated mice revealed significant differences in AR localisation. While AR is mainly nuclear in tumours from the DMSO treated cohort, we observed a significant reduction in AR nuclear localisation in terbinafine treated tumours (Fig. 7i). This suggests that blocking *SQLE* activity inhibits AR transactivation without causing a reduction in serum cholesterol. We believe that this finding provides additional support for our conclusions. Nevertheless, we acknowledge that multiple processes can lead to the development of CRPC and that further studies are required to evaluate whether *SQLE* inhibition by terbinafine also affects tumour growth in castrated mice.

14) The serum cholesterol level is not altered by terbinafine treatment (Figure 5J). This is surprising – in other words, the authors are suggesting that the treatment has differential effects on tumour and the liver, and that in hormone proficient host mice, hormone dependent treatment naïve tumors are already utilising de novo androgen synthesis as a growth strategy. The authors should clarify this point.

In addition, if that is the case, then why are there relatively few cases shown to involve this pathway in the clinical TCGA cohort?

The reviewer is right in concluding that this is a surprising result. Plasma cholesterol levels are controlled by dietary cholesterol and by *de novo* cholesterol synthesis, mainly by the liver. A previous study by Liu et al. observed a small reduction in plasma cholesterol levels in mice after terbinafine treatment at a daily oral dose of 80 mg/kg. However, this study used *Sqle* transgenic mice treated with DEN plus high-fat/high-cholesterol diet to induce liver cancers. In this setting, terbinafine treatment also reduced liver tumours, making it difficult to differentiate between systemic and tumour-specific effects³. We applied a dose of 50 mg/kg every 48 hours by i.p. injection and our mice were fed a standard chow diet (containing 400 mg/kg of cholesterol). It is possible that any effects of terbinafine on systemic cholesterol levels were compensated by dietary cholesterol. This raises the intriguing possibility that circulating cholesterol may not be readily available within the tumour microenvironment of the prostate. This would favour the growth of cancer cells that are capable of *de novo* cholesterol synthesis and renders them sensitive to *SQLE* inhibition. We made a similar observation previously demonstrating that inhibition of stearoyl-CoA desaturase (*SCD*) blocks orthotopic growth of

PCa cells in mice kept on a normal diet¹⁶. However, further analysis outside of the scope of this study would be needed to prove this hypothesis.

It should also be noted that the VCaP cell line used in the implantation experiment was originally isolated from a patient with castration resistant disease and thus cannot be considered treatment naïve. In contrast, the TCGA cohort is based on prostatectomy tissue when most tumours have not yet developed into CRPC. Based on these considerations, we believe that our findings offer insight into the metabolic requirements of PCa tumours and provide additional evidence for the importance of intra-tumoural cholesterol synthesis.

15) The title of the report focuses on aggressive disease, so it would be useful to have data on metastasis or treatment resistance in the in vivo assessment.

We agree with the reviewer that the title is somewhat misleading. We do not wish to suggest a role of SQLE is driving metastatic spread in PCa. Our conclusion regarding aggressive disease was based on the clinical parameters of the patient cohorts used for the analysis of miR-205 and SQLE analysis (Fig. 1 and 2, Fig. S2). A specific evaluation of the potential contribution of SQLE in metastasis formation would be highly interesting but is beyond the scope of the current study.

We now also include data describing the effect of terbinafine treatment in four late-stage prostate cancer patients. Two weeks of terbinafine treatment resulted in a reduction of serum PSA levels in three of the four patients without greater adverse side-effects. These data are currently provided as accompanying information.

16) The authors suggest that terbinafine mainly affects AR signalling via SQLE status. Does terbinafine treatment simply mimic treatment with AR inhibitors or would it synergize with androgen receptor pathway inhibitors?

This is a very interesting suggestion. We are currently following up on this line of investigation in a larger study. Nevertheless, we believe that adding these experiments would go beyond the scope of the current manuscript.

Minor comments:

Out of curiosity, why is there 2 bands for SQLE WB?

We have repeated all immunoblots for SQLE using a different antibody. Careful size alignment was applied to demonstrate specificity of the band at 64 kDa with an unspecific band with higher mobility. Respective figures were revised and labelled accordingly.

The use of "overexpression of shRNAs" in figure legends does not sound right

This has been corrected.

PCR graphs on figure 4f and S4c, S4d are labelled LKL3 instead of KLK3

This has been corrected.

In the text, S4C is cited after S4D

This has been corrected.

Would the acute co-treatment of terbinafine and AR inhibitors be beneficial in vitro?

As mentioned above, we believe that this is an interesting experiment that we are currently evaluating as part of a follow-up study.

Some of the experiments performed on C4-2 SQLE OE cells (migration, cholesterol labelling, AR activity or KLK3 expression) in supplementary would benefit from being included in the main figures.

We thank the reviewer for this suggestion. We have moved the experiments showing induction of AR activity and induction of migration after SQLE overexpression to the main figures.

Fig 1 showed evidence of alteration involving SQLE in clinical prostate cancer cohorts. Associations with other components of the cholesterol pathway in Figure S1 are nice but some of the labels for Figure S1 are too small.

We have revised the figures accordingly. We now only show selected results for improved clarity.

Space before the unit e.g. mM was missing in places throughout the report. μ l and μ L were used in different part of the report.

This has been corrected.

Additional references:

1. Mahoney CE, *et al.* A chemical biology screen identifies a vulnerability of neuroendocrine cancer cells to SQLE inhibition. *Nat Commun* **10**, 96 (2019).
2. Puhr M, *et al.* The Glucocorticoid Receptor Is a Key Player for Prostate Cancer Cell Survival and a Target for Improved Antiandrogen Therapy. *Clin Cancer Res* **24**, 927-938 (2018).
3. Liu D, *et al.* Squalene epoxidase drives NAFLD-induced hepatocellular carcinoma and is a pharmaceutical target. *Sci Transl Med* **10**, (2018).
4. Gill S, Stevenson J, Kristiana I, Brown AJ. Cholesterol-dependent degradation of squalene monooxygenase, a control point in cholesterol synthesis beyond HMG-CoA reductase. *Cell metabolism* **13**, 260-273 (2011).
5. van Bokhoven A, *et al.* Molecular characterization of human prostate carcinoma cell lines. *The Prostate* **57**, 205-225 (2003).
6. Du Q, *et al.* MIR205HG Is a Long Noncoding RNA that Regulates Growth Hormone and Prolactin Production in the Anterior Pituitary. *Dev Cell* **49**, 618-631 e615 (2019).
7. Ambs S, *et al.* Genomic profiling of microRNA and messenger RNA reveals deregulated microRNA expression in prostate cancer. *Cancer Res* **68**, 6162-6170 (2008).
8. Qin Y, Zhang Y, Tang Q, Jin L, Chen Y. SQLE induces epithelial-to-mesenchymal transition by regulating of miR-133b in esophageal squamous cell carcinoma. *Acta Biochim Biophys Sin (Shanghai)* **49**, 138-148 (2017).
9. Ji J, Sundquist J, Sundquist K. Use of terbinafine and risk of death in patients with prostate cancer: A population-based cohort study. *Int J Cancer* **144**, 1888-1895 (2019).

10. Gregory PA, *et al.* The miR-200 family and miR-205 regulate epithelial to mesenchymal transition by targeting ZEB1 and SIP1. *Nat Cell Biol* **10**, 593-601 (2008).
11. Toropainen S, Niskanen EA, Malinen M, Sutinen P, Kaikkonen MU, Palvimo JJ. Global analysis of transcription in castration-resistant prostate cancer cells uncovers active enhancers and direct androgen receptor targets. *Sci Rep* **6**, 33510 (2016).
12. Gordon JA, *et al.* Upregulation of Scavenger Receptor B1 Is Required for Steroidogenic and Nonsteroidogenic Cholesterol Metabolism in Prostate Cancer. *Cancer Res* **79**, 3320-3331 (2019).
13. Brown MS, Goldstein JL. The SREBP pathway: regulation of cholesterol metabolism by proteolysis of a membrane-bound transcription factor. *Cell* **89**, 331-340 (1997).
14. Wilson S, Qi J, Filipp FV. Refinement of the androgen response element based on ChIP-Seq in androgen-insensitive and androgen-responsive prostate cancer cell lines. *Sci Rep* **6**, 32611 (2016).
15. Sbiera S, *et al.* Mitotane inhibits Sterol-O-Acyl Transferase 1 triggering lipid-mediated endoplasmic reticulum stress and apoptosis in adrenocortical carcinoma cells. *Endocrinology*, en20151367 (2015).
16. Peck B, *et al.* Inhibition of fatty acid desaturation is detrimental to cancer cell survival in metabolically compromised environments. *Cancer Metab* **4**, 6 (2016).

REVIEWER COMMENTS

Reviewer #1 (Remarks to the Author):

Overall, the authors have satisfied my concerns, and the manuscript has been strengthened.

The manuscript would benefit from double-checking wrt spelling
e.g. Line 111 correlated rather than corelated; Line 439 SR-B1 rather than SE-B1

Reviewer #2 (Remarks to the Author):

miR-205, a microRNA normally expressed in prostatic basal cells, has been shown by many studies to possess 'tumor-suppressive' functions although its genetic deletion actually compromises the Pten-null-induced prostate cancer (Mol Carcinogenesis, 58:2241-53, 2019). Increased cholesterol synthesis and intra-tumoral steroidogenesis in PCa have been well reported and SQLE expression has been correlated with lethal PCa (Cancer Res, 76:4785-90, 2016). The present study establishes a mechanistic link among miR-205, SQLE, cholesterol production and PCa aggressiveness. In the revised manuscript, authors have made conscientious efforts in addressing most of the concerns raised by previous reviewers. There are, however, still several issues that should be addressed in association with the current manuscript.

1. How is miR-205 downregulated in PCa? Epigenetic mechanisms (e.g., promoter hypermethylation) or genomic loss?
2. Is SQLE genomically amplified in metastatic PCa?
3. What about miR-205-SQLE-cholesterol-AR axis in AR-negative PCa or DNPC that do not express AR nor require AR signaling? Data in Fig. 5 seems to suggest that this signaling axis is less involved or less important than in AR+ PCa models. This important point was also raised by Rev#3 but was not properly addressed by the authors mainly because the cell models they utilized were not isogenic and therefore may be difficult to compare. Some of the key recent publications on AR-negative PCa (cells) should be cited (e.g., Cancer Cell 32:474-489, 2017; Nat. Commun. 9(1):3600, 2018; JCI, 130:4492-4505, 2019).
4. Authors kept arguing that VCaP cells, originally isolated from a vertebrate metastasis, should not be considered 'treatment-naïve' cells but instead should be thought of as 'castration-resistant' or treatment-refractory cells. The truth is that most PCa cells were derived from the treatment-failed mCRPC patients but once they are derived as cell lines, such as VCaP cells, they may become exquisitely sensitive to castration (unless they are AR-neg cells such as PC3). At least one castration (CRPC) experiment should be conducted for this project.
5. There are still issues with data presentations. For example, the SQLE mRNA vs. protein still have not been distinguished from each other (e.g., Fig. S2a). Fig. S2b should be about MIR205HG but why the y-axis label was 'SQLE expression'?
6. I think the title should specifically mention miR-205 instead of just 'microRNA-driven', which is rather vague.

Reviewer #3 (Remarks to the Author):

The authors have adequately addressed all of the comments and queries raised in the review, with the provision of new data and additional discussions included.

Reviewer #4 (Remarks to the Author):

I was asked to provide a clinical and methodological review on the revised paper submitted. Thus, the following comments focus on the reported analyses correlating SQLE with patient outcome and the four case reports

The authors provide analyses based on several patient populations to investigate the prognostic

impact of SQLE status /expression on patient survival prognosis. In the current version, there are several issues that need to be addressed: definition of analysis population, reporting of analyses and interpretation of findings. Overall, the current version of the paper lacks adherence to the REporting recommendations for tumour MARKer prognostic studies (REMARK). REMARK is a reporting guideline endorsed by journals of the Nature group.

Specifically, I have the following comments which may help to improve the manuscript:

Definition of populations

Regarding the 179 patients (TMA 2), please provide more information on how data of these patients and for which purpose they were collected. I assume these are all patient being considered high-risk for relapse at diagnosis, but without evidence for metastatic disease. Having a look at this paper by Kneitz et al 2014, the population was diagnosed between 1987 and 2005. Just referring to reference 21 is not sufficient for the reader.

You report analyses on 453 "unselected" patients (TMA 1). It is entirely unclear from where data of these patients were collected. What do you mean by "unselected"? There need be at least some criteria. Why was it exactly 453? During which time were patients diagnosed? What was their treatment? What additional clinical characteristics were available?

There is a difference in the distribution of the Gleason Score levels between the 179 TMA 2 patients and the 453 "unselected" patients; especially regarding Gleason 6 and 7. So the cohort of 453 patients are at higher risk when only looking at the Gleason score distribution. On the other hand, pre-operative PSA levels are on average higher in the 179 TMA 2 group. Do you consider these two cohorts being at different risks? Please comment on this.

You report follow-up time in table 1, how is that defined? Follow-up time for all patients? Only those still being alive at last follow-up? Please specify.

The TCGA population N= 430, how were data from these patients collected? Also, I am a bit puzzled, were these patients with metastatic or localized disease? On page 16, line 547 you state that gene expression data were derived from metastatic patients. However, the KM plot is a bit "too good" for metastatic prostate cancer patients, unless it is a very well selected population.

Please provide more information on how and where patient data were collected in table 1.

Statistical analysis

SQLE expression is a continuous variable, therefore, when analyzing the association between SQLE and patient outcome, the primary analysis should include SQLE as a continuous variable and not dichotomized into high and low. There are several methodological issues as also outlined in the REMARK guidelines. One of them is, that you assume a linear relationship between the predictor and outcome. However, the most important is the risk for false positive findings – an arbitrary cut-off showing a significant result may in fact be wrong. Therefore, I strongly recommend that you conduct analysis including the SQLE variable in its continuous form including investigation on potentially non-linear relationships.

Obviously, SQLE is associated with known high-risk features, so why do you think SQLE is the driving prognostic marker? Have you considered to conduct a multivariable analysis including other known prognostic markers?

Figure 1 g, here you plot the cancer specific survival, how was that defined? Please describe the censoring rules you applied. How was death due to cancer defined and how were data quality checked regarding this endpoint? Just referring to the definition by the National Cancer Institute (NCI) is not sufficient. Additionally, such analysis is not helpful. If you really want to show the risk of death due to cancer you should calculate cumulative incidence functions with non-cancer specific death as competing risk.

In the survival analysis of Cancer Genome Atlas Data, you used the “receiver-operating characteristics” to identify the “optimal” cut-off for SQLE (1024.544). However, in figure 1f and 1g you just dichotomized by the median (same issue in Figure 1S f and g). Why? This shows the inherent problem of your approach cutting a continuous variable into pieces just for the aim to display survival functions in a Kaplan-Meier plot. I do not say that this is wrong, but the way you derive a potential cut-off should mainly be driven by an investigation on the continuous predictor as outlined above.

Discussion

Treatment options for prostate cancer have markedly improved over the past decades. Therefore, it is very important to know during which time periods patients in the different cohorts were treated. In other words, 15 years ago, SQLE may have been of prognostic relevance, but in the meantime, this impact is no longer present, because of improved therapeutic options.

Lastly, you need to address the issue of residual confounding when appraising your analyses regarding the prognostic value of SQLE – even if the effect is still present in multivariable analyses which I strongly recommend conducting.

Case reports

Three of four patients received the drug only 2 weeks. Patient # 3 received it six weeks. Although you observed a dip in PSA, based on these anecdotal case reports, I think it is not appropriate to state that you have really delayed PSA progression. Also, regarding safety, how can you judge safety if the patient only received the drug for 2 weeks? None of the patients really derived a clinical benefit. Thus, I strongly recommend tuning down the wording regarding conclusions on effectiveness and safety

Please report the respective survival time of each patient and, for the time series plots, please provide a mark when the respective patient has died.

Please drop the last column in table 1. There is absolutely no need to provide summary statistics when reporting 4 cases.

For each patient, please provide the time from initiation of terbinafine treatment to death.

Finally, please provide description of all 4 patients not only two.

Response to reviewer' comments:

We would like to thank the reviewers for their time and effort in reviewing our manuscript and we greatly appreciate their insightful evaluation.

Reviewer #1 (Remarks to the Author):

Overall, the authors have satisfied my concerns, and the manuscript has been strengthened. The manuscript would benefit from double-checking wrt spelling e.g. Line 111 correlated rather than corelated; Line 439 SR-B1 rather than SE-B1.

We thank the reviewer for their positive evaluation. The manuscript has been proofread again to eliminate spelling mistakes.

Reviewer #2 (Remarks to the Author):

miR-205, a microRNA normally expressed in prostatic basal cells, has been shown by many studies to possess 'tumor-suppressive' functions although its genetic deletion actually compromises the Pten-null-induced prostate cancer (Mol Carcinogenesis, 58:2241-53, 2019). Increased cholesterol synthesis and intra-tumoral steroidogenesis in PCa have been well reported and SQLE expression has been correlated with lethal PCa (Cancer Res, 76:4785-90, 2016). The present study establishes a mechanistic link among miR-205, SQLE, cholesterol production and PCa aggressiveness. In the revised manuscript, authors have made conscientious efforts in addressing most of the concerns raised by previous reviewers.

There are, however, still several issues that should be addressed in association with the current manuscript.

1. How is miR-205 downregulated in PCa? Epigenetic mechanisms (e.g., promoter hypermethylation) or genomic loss?

We agree that this is an important question and we apologise for not including the appropriate references in the manuscript. Downregulation of miR-205 in prostate cancer (PCa) involved epigenetic regulation via enhanced DNA methylation of several CpG sites within the *miR-205* locus ¹. Furthermore, the expression of miR-205 is controlled by p63 and it was shown that both p63 and miR-205 are lost in metastatic prostate cancer ². Both studies are now mentioned in the revised manuscript (see lines 97 and 423/426).

2. Is SQLE genomically amplified in metastatic PCa?

We also thank the reviewer for raising this important point. Copy number alterations of the *SQLE* locus across multiple cancer types has already been described before, with PCa showing gains in 31% of patients ³. We therefore have included a figure displaying *SQLE* amplification and expression in primary and metastatic PCa (new Figure S1a and S1b). We believe that these data further confirm the importance of *SQLE* in prostate cancer.

3. What about miR-205-SQLE-cholesterol-AR axis in AR-negative PCa or DNPC that do not express AR nor require AR signaling? Data in Fig. 5 seems to suggest that this signaling axis is less involved or less important than in AR+ PCa models. This important point was also raised by Rev#3 but was not properly addressed by the authors mainly because the cell models they utilized were not isogenic and therefore may be difficult to compare. Some of the key recent publications on AR-negative PCa (cells) should be cited (e.g., Cancer Cell 32:474-489, 2017; Nat. Commun. 9(1):3600, 2018; JCI, 130:4492-4505, 2019).

We realise that we did not fully clarify this point. Our study is based on the premise that resistance to anti-androgen therapy (ADT) can, at least in some instances, be achieved by intratumoral testosterone synthesis, as demonstrated by numerous previous studies (e.g. ⁴). Castration resistant prostate cancer (CRPC) frequently maintains

expression of *AR* and *AR*-target genes ⁵, suggesting that *AR* signalling is still relevant in this setting ⁶.

We believe that using isogenic cell lines is not feasible in this context, as genetic manipulation would not recapitulate the selection processes that lead to the emergence of CRPC. Instead, we have used a panel of PCa cell lines in our study: PC3 and DU145 are *AR*-negative while LNCaP cells are *AR*-positive and androgen-dependent. The LNCaP-derivative C4-2 ⁷ and the PDX-derived VCaP cell line ⁸ represent models of CRPC that show reduced requirement for exogenous androgen but maintain *AR* expression and sensitivity towards *AR* inhibition. Our results indicate that *AR*-negative cell lines are less sensitive to SQLE inhibition than the *AR*-positive cell lines (Figure 5a and b). To clarify this point, we have added an additional statistical comparison to the data and also revised our discussion of the results (see lines 255ff).

We would also like to point out that the LNCaP-Abl cells used in our study are derived through long-term culture of LNCaP cells in steroid-free medium ⁹, and thus experienced a similar selective pressure as the one induced by ADT. Despite their independence on exogenous androgen, LNCaP-Abl cells are still dependent on *AR* signalling, as they maintain *AR* expression ⁹ and are sensitive to enzalutamide (Figure 5f). Our results show that LNCaP-Abl cells are highly sensitive to SQLE inhibition by terbinafine or NB-598 (Figure 5f and g), indicating that at least a subset of CRPC could be targeted by anti-SQLE therapy.

In line with the reviewer's comment regarding additional phenotypes of CRPC, we have included a statement on the limitations of our results (see lines 476-478 and have added the mentioned studies to the references.

4. Authors kept arguing that VCaP cells, originally isolated from a vertebrate metastasis, should not be considered 'treatment-naïve' cells but instead should be thought of as 'castration-resistant' or treatment-refractory cells. The truth is that most PCa cells were derived from the treatment-failed mCRPC patients but once they are derived as cell lines, such as VCaP cells, they may become exquisitely sensitive to castration (unless they are *AR*-neg cells such as PC3). At least one castration (CRPC) experiment should be conducted for this project.

We hope that this issue has been at least partially clarified by our response to point 3. Furthermore, although VCaP cells are frequently described as a model of CRPC in the literature, it did not want to imply that they do not respond to castration. Indeed, a previous study observed a transient decline in PSA following castration in mice that had been orthotopically implantation with VCap cells ¹⁰. The same study also demonstrated that castration-resistant VCaP cells achieved high intratumoral androgen concentrations and upregulation of steroidogenic enzymes.

In our study, terbinafine treatment reduced PSA levels and tumour growth at a dose that is similar to the one used to treat fungal infections in humans. Importantly, reduction in tumour growth was not accompanied by a reduction in circulating cholesterol, making it unlikely that the therapeutic effect of terbinafine is mediated by blocking steroid synthesis in the adrenal gland. Given the results of Knuutila et al., ¹⁰ we would expect that inhibition of SQLE could be even more effective in castration-resistant VCaP tumours. We have already attempted to investigate this point by using the LNCaP-Abl cell line in combination with castration. However, inefficient tumour-take in this model precluded a treatment study with sufficient power. While we hope to address this in future studies, the lengthy timeline of ethical approval for animal experimentation prevents us from including further *in vivo* studies into the current manuscript.

5. There are still issues with data presentations. For example, the SQLE mRNA vs. protein

still have not been distinguished from each other (e.g., Fig. S2a). Fig. S2b should be about MIR205HG but why the y-axis label was 'SQLE expression'?

We thank the reviewer for pointing this out and we have carefully revised the text and figures.

6. I think the title should specifically mention miR-205 instead of just 'microRNA-driven', which is rather vague.

We agree with this suggestion and have revised the title.

Reviewer #3 (Remarks to the Author):

The authors have adequately addressed all of the comments and queries raised in the review, with the provision of new data and additional discussions included.

We thank the reviewer for their positive evaluation.

Reviewer #4 (Remarks to the Author):

I was asked to provide a clinical and methodological review on the revised paper submitted. Thus, the following comments focus on the reported analyses correlating SQLE with patient outcome and the four case reports

The authors provide analyses based on several patient populations to investigate the prognostic impact of SQLE status /expression on patient survival prognosis. In the current version, there are several issues that need to be addressed: definition of analysis population, reporting of analyses and interpretation of findings. Overall, the current version of the paper lacks adherence to the REporting recommendations for tumour MARKer prognostic studies (REMARK). REMARK is a reporting guideline endorsed by journals of the Nature group.

Specifically, I have the following comments which may help to improve the manuscript:

Definition of populations:

Regarding the 179 patients (TMA 2), please provide more information on how data of these patients and for which purpose they were collected. I assume these are all patient being considered high-risk for relapse at diagnosis, but without evidence for metastatic disease. Having a look at this paper by Kneitz et al 2014, the population was diagnosed between 1987 and 2005. Just referring to reference 21 is not sufficient for the reader.

You report analyses on 453 "unselected" patients (TMA 1). It is entirely unclear from where data of these patients were collected. What do you mean by "unselected"? There need be at least some criteria. Why was it exactly 453? During which time were patients diagnosed? What was their treatment? What additional clinical characteristics were available?

We thank the reviewer for pointing out this oversight. We have added a detailed description of the two patient cohorts to the materials and methods section (see lines 563-590).

There is a difference in the distribution of the Gleason Score levels between the 179 TMA 2 patients and the 453 "unselected" patients; especially regarding Gleason 6 and 7. So the cohort of 453 patients are at higher risk when only looking at the Gleason score distribution. On the other hand, pre-operative PSA levels are on average higher in the 179 TMA 2 group. Do you consider these two cohorts being at different risks? Please comment on this.

We agree with the reviewer that this point had to be clarified. We now define the two patient cohorts used for histological analysis in the materials and methods section. The patient cohort represented by TMA2 is classified as high-risk according to the D'Amico risk classification system ¹¹. We also removed the term "high-risk" from the

description of this TMA and just refer to Table 1 for clinicopathological and outcome data for all cohorts.

You report follow-up time in table 1, how is that defined? Follow-up time for all patients? Only those still being alive at last follow-up? Please specify.

Again, this is now described in detail in the materials and methods section.

The TCGA population N=430, how were data from these patients collected? Also, I am a bit puzzled, were these patients with metastatic or localized disease? On page 16, line 547 you state that gene expression data were derived from metastatic patients. However, the KM plot is a bit “too good” for metastatic prostate cancer patients, unless it is a very well selected population.

We agree with the reviewer that this analysis was somewhat unclear. We used TCGA data from both primary and metastatic PCa in our analysis. The survival analysis previously shown in Fig. 1a used a TCGA gene expression dataset (PRAD primary, PanCancer Atlas). Although this dataset is generally considered suitable for high-quality survival analysis¹², we only have access to the clinicopathological patient information held within TCGA. In the light of this limitation, we have decided to remove these data from the revised version of the manuscript as it did not add additional information over the survival analysis of the two TMA cohorts.

Please provide more information on how and where patient data were collected in table 1. **We have now added additional information to Table 1 and also extended the description of patient cohorts in the materials and methods section.**

Statistical analysis:

SQLE expression is a continuous variable, therefore, when analyzing the association between SQLE and patient outcome, the primary analysis should include SQLE as a continuous variable and not dichotomized into high and low. There are several methodological issues as also outlined in the REMARK guidelines. One of them is, that you assume a linear relationship between the predictor and outcome. However, the most important is the risk for false positive findings – an arbitrary cut-off showing a significant result may in fact be wrong. Therefore, I strongly recommend that you conduct analysis including the SQLE variable in its continuous form including investigation on potentially non-linear relationships.

We apologise for this lack of stringency. As overexpression of SQLE in lethal PCa had already been described previously¹³, the primary aim of our study was to elucidate the molecular mechanisms underlying the importance of SQLE in this disease. However, to address this important point, we now also include a COX regression analysis using SQLE expression both as a continuous and as dichotomised vector. Both parameters are significantly associated with recurrence, even if tested against known prognosticators such as Gleason score, in a multivariate model (see supplementary Table 1).

Obviously, SQLE is associated with known high-risk features, so why do you think SQLE is the driving prognostic marker? Have you considered to conduct a multivariable analysis including other known prognostic markers?

As mentioned in the response to the previous point, we do not claim that SQLE is a driving prognostic marker in PCa. Our data confirm previous reports that SQLE is associated with more aggressive tumour features and adverse survival features (recurrence free survival) in two independent TMA tissue cohorts. Nevertheless, the multivariate COX analysis shows that SQLE is an independent marker of recurrence (supplementary Table 1).

Figure 1 g, here you plot the cancer specific survival, how was that defined? Please describe

the censoring rules you applied. How was death due to cancer defined and how were data quality checked regarding this endpoint? Just referring to the definition by the National Cancer Institute (NCI) is not sufficient. Additionally, such analysis is not helpful. If you really want to show the risk of death due to cancer you should calculate cumulative incidence functions with non-cancer specific death as competing risk.

This is an important point that has escaped our attention. We therefore decided to change the analysis and now show recurrence-free survival (RFS) and overall survival (OS) for the two patient cohorts (see revised Fig. 1e/f and S1h/i).

In the survival analysis of Cancer Genome Atlas Data, you used the “receiver-operating characteristics” to identify the “optimal” cut-off for SQLE (1024.544). However, in figure 1f and 1g you just dichotomized by the median (same issue in Figure 1S f and g). Why? This shows the inherent problem of your approach cutting a continuous variable into pieces just for the aim to display survival functions in a Kaplan-Meier plot. I do not say that this is wrong, but the way you derive a potential cut-off should mainly be driven by an investigation on the continuous predictor as outlined above.

We agree with this comment and have now removed this analysis from our manuscript.

Discussion

Treatment options for prostate cancer have markedly improved over the past decades. Therefore, it is very important to know during which time periods patients in the different cohorts were treated. In other words, 15 years ago, SQLE may have been of prognostic relevance, but in the meantime, this impact is no longer present, because of improved therapeutic options.

The main intention of our study was to uncover the molecular mechanism underlying the function of SQLE in PCa. We believe that our study, analysing multiple patient cohorts spanning a considerable time period (up to 2012) and using advanced cell lines models and orthotopic xenografts, provides substantial insight into the role of SQLE in driving AR signalling and tumour growth. The accompanying case series also suggests that targeting SQLE could be a therapeutic opportunity in advanced PCa. We agree, however, that therapeutic options have changed substantially over the past two decades. To address this problem, we now changed our survival analysis to recurrence-free survival (RFS), as the time to progression after prostatectomy is independent of the advanced treatment options available today.

Lastly, you need to address the issue of residual confounding when appraising your analyses regarding the prognostic value of SQLE – even if the effect is still present in multivariable analyses which I strongly recommend conducting.

We have conducted multivariable analysis (see Table S1). We also added a statement on the limitations of our study (see lines 142-144).

Case reports

Three of four patients received the drug only 2 weeks. Patient # 3 received it six weeks. Although you observed a dip in PSA, based on these anecdotal case reports, I think it is not appropriate to state that you have really delayed PSA progression. Also, regarding safety, how can you judge safety if the patient only received the drug for 2 weeks? None of the patients really derived a clinical benefit. Thus, I strongly recommend tuning down the wording regarding conclusions on effectiveness and safety.

We have revised the wording of the case report to reflect those limitations. Nevertheless, we would also like to point out a study using high-dose of terbinafine for the treatment of cutaneous or lymphocutaneous sporotrichosis without major adverse events¹⁴. Further assessment of drug safety lies beyond the scope of our study.

Please report the respective survival time of each patient and, for the time series plots, please provide a mark when the respective patient has died.

We have revised the case report accordingly.

Please drop the last column in table 1. There is absolutely no need to provide summary statistics when reporting 4 cases.

We agree and have removed this column from Table 1.

For each patient, please provide the time from initiation of terbinafine treatment to death.

This information is now provided in a table.

Finally, please provide description of all 4 patients not only two.

We now provide a detailed description of all four patients in the text of the case series.

Additional references:

1. Hulf T, *et al.* Epigenetic-induced repression of microRNA-205 is associated with MED1 activation and a poorer prognosis in localized prostate cancer. *Oncogene* **32**, 2891-2899 (2013).
2. Tucci P, *et al.* Loss of p63 and its microRNA-205 target results in enhanced cell migration and metastasis in prostate cancer. *Proc Natl Acad Sci U S A* **109**, 15312-15317 (2012).
3. Brown DN, *et al.* Squalene epoxidase is a bona fide oncogene by amplification with clinical relevance in breast cancer. *Sci Rep* **6**, 19435 (2016).
4. Montgomery RB, *et al.* Maintenance of intratumoral androgens in metastatic prostate cancer: a mechanism for castration-resistant tumor growth. *Cancer Res* **68**, 4447-4454 (2008).
5. Labrecque MP, *et al.* Molecular profiling stratifies diverse phenotypes of treatment-refractory metastatic castration-resistant prostate cancer. *J Clin Invest* **129**, 4492-4505 (2019).
6. Watson PA, Arora VK, Sawyers CL. Emerging mechanisms of resistance to androgen receptor inhibitors in prostate cancer. *Nat Rev Cancer* **15**, 701-711 (2015).
7. Wu TT, *et al.* Establishing human prostate cancer cell xenografts in bone: induction of osteoblastic reaction by prostate-specific antigen-producing tumors in athymic and SCID/bg mice using LNCaP and lineage-derived metastatic sublines. *Int J Cancer* **77**, 887-894 (1998).
8. Korenchuk S, *et al.* VCaP, a cell-based model system of human prostate cancer. *In Vivo* **15**, 163-168 (2001).

9. Puhr M, *et al.* The Glucocorticoid Receptor Is a Key Player for Prostate Cancer Cell Survival and a Target for Improved Antiandrogen Therapy. *Clin Cancer Res* **24**, 927-938 (2018).
10. Knuutila M, *et al.* Castration induces up-regulation of intratumoral androgen biosynthesis and androgen receptor expression in an orthotopic VCaP human prostate cancer xenograft model. *Am J Pathol* **184**, 2163-2173 (2014).
11. D'Amico AV. Risk-based management of prostate cancer. *N Engl J Med* **365**, 169-171 (2011).
12. Liu J, *et al.* An Integrated TCGA Pan-Cancer Clinical Data Resource to Drive High-Quality Survival Outcome Analytics. *Cell* **173**, 400-416 e411 (2018).
13. Stopsack KH, *et al.* Cholesterol Metabolism and Prostate Cancer Lethality. *Cancer Res* **76**, 4785-4790 (2016).
14. Chapman SW, *et al.* Comparative evaluation of the efficacy and safety of two doses of terbinafine (500 and 1000 mg day⁻¹) in the treatment of cutaneous or lymphocutaneous sporotrichosis. *Mycoses* **47**, 62-68 (2004).

REVIEWERS' COMMENTS

Reviewer #2 (Remarks to the Author):

The authors have properly addressed my concerns and comments.

Reviewer #4 (Remarks to the Author):

The authors have adequately addressed the comments with the provision of additional analyses, clarification of populations and points in the discussion.

Response to reviewers' comments:

We thank the reviewers for their insightful assessment of our manuscript.

Reviewer #2 (Remarks to the Author):

The authors have properly addressed my concerns and comments.

Reviewer #4 (Remarks to the Author):

The authors have adequately addressed the comments with the provision of additional analyses, clarification of populations and points in the discussion.